

# Descriptions of five new species of the salamander genus *Chiropterotriton* (Caudata: Plethodontidae) from eastern Mexico and the status of three currently recognized taxa

Gabriela Parra Olea[1], Mirna G. Garcia-Castillo[1,2], Sean M. Rovito[3], Jessica A. Maisano[4], James Hanken[5] and David B. Wake[6]

[1] Zoology, Instituto de Biología, Universidad Nacional Autonoma de México, Mexico city, México
[2] Posgrado en Ciencias Biológicas, Universidad Nacional Autónoma de México, Ciudad de México, México
[3] Unidad de Genómica Avanzada (Langebio), CINVESTAV, Irapuato, Guanajuato, México
[4] Jackson School of Geosciences, University of Texas at Austin, Austin, TX, USA
[5] Department of Organismic and Evolutionary Biology and Museum of Comparative Zoology, Harvard University, Cambridge, MA, USA
[6] Department of Integrative Biology and Museum of Vertebrate Zoology, University of California, Berkeley, CA, USA

Corresponding author
Gabriela Parra Olea,
gparra@ib.unam.mx

## ABSTRACT

The genus *Chiropterotriton* is endemic to Mexico with a geographical distribution along the Sierra Madre Oriental, the Trans Mexican Volcanic Belt and the Sierra de Juárez. The recent use of molecular tools has shown that Mexico's amphibian diversity is highly underestimated, including a large number of cryptic, unnamed species. *Chiropterotriton* has 18 described species including terrestrial, arboreal and cave-dwelling species. In previous molecular studies, the presence of multiple undescribed species was evident. We present a phylogenetic hypothesis based on mitochondrial data, which includes all described species and six undescribed taxa. Based on the morphological analyses and, when available, combined with molecular data, we describe five new species of the genus; *Chiropterotriton casasi* sp. nov., *C. ceronorum* sp. nov., *C. melipona* sp. nov., *C. perotensis* sp. nov. and *C. totonacus* sp. nov. In addition, we redescribe two others: *Chiropterotriton chiropterus* and *C. orculus*, and provide a comparable account of one additional sympatric congener. This increases the number of species in the genus to 23, which represent a considerable component of Mexican plethodontid richness.

## INTRODUCTION

The genus *Chiropterotriton Taylor (1944)* has proven to be one of the taxonomically most difficult of all genera of neotropical salamanders. These salamanders vary widely in morphology and ecology from relatively large troglodytic forms to gracile arboreal species. Most species, however, are small to medium sized with a fairly generalized external

morphology, representing minor variations on a conserved body plan (*Darda & Wake, 2015*). This external morphological similarity has complicated recognition of new species and the relationships between them, particularly based on morphological data alone.

When *Taylor (1944)* described the genus, he initially included a number of other Central American salamanders from Nuclear Central America and Costa Rica. These species, which are all relatively small and slender, were recognized as a distinct unit within the genus (*Chiropterotriton* Beta; *Wake & Lynch, 1976*) and eventually described as several distinct genera (*Cryptotriton, Dendrotriton* and *Nototriton*), leaving *Chiropterotriton* endemic to the highlands of Mexico and west of the Isthmus of Tehuantepec. Despite their external similarity, the divergence between each of these genera and *Chiropterotriton* spans the basal node in the Bolitoglossini clade (*Rovito et al., 2015a*). Taxonomy of the Mexican *Chiropterotriton* was complicated not only by their small size and generalized morphology, but also by the fact that two of the earliest species descriptions for the group, *C. chiropterus* (*Cope, 1863*) and *C. orculus* (*Cope, 1865*) are very brief and provide imprecise localities, and because the holotype of each species has been lost.

*Rabb (1958)* made a major advance in our understanding of the taxonomy and morphology of the northern species in the group. By examining both topotypic specimens and material from additional localities, he showed that unappreciated diversity existed even within the subset of species from this region, based on external morphology and tooth counts. Rabb's foundational morphological and taxonomic work on the genus was followed by a long period of taxonomic stasis. Following his discovery and description of *Chiropterotriton magnipes* (*Rabb, 1965*), the most morphologically distinct species in the genus, no additional species were described for nearly 50 years. Despite the lapse in species descriptions, molecular data made it clear that much diversity lay hidden within already known populations. *Darda (1994)* derived an allozyme dataset that showed that many populations likely represented distinct species, and his results were largely corroborated by mtDNA sequence data (*Parra-Olea, 2003*) although there were some discrepancies between the results from the two data sets. Collection of new material from previously known populations for molecular analysis, as well as the discovery of new populations, led to the description of six new species since 2014 (*Campbell et al., 2014*; *Rovito & Parra-Olea, 2015*; *García-Castillo et al., 2017*, *2018*). Despite these recent descriptions, many populations from central Mexico have defied assignment to known species and are best recognized as distinct species.

The *Chiropterotriton chiropterus* complex has suffered from taxonomic rearrangements, mostly due to imprecise type localities and the lack of adequate samples from those localities. Based on external morphology, *Wake & Lynch (1976)* defined the *chiropterus* group to include *C. chiropterus, C. chondrostega, C. dimidiatus* and *C. lavae*. Later, on the basis of immunological data, *Maxson & Wake (1981)* redefined the *chiropterus* group to include only *C. chiropterus* and *C. lavae*. Based on allozyme data, *Darda (1994)* recognized a group of populations found along the Trans-Mexican Volcanic Belt, which he called the *chiropterus* complex. This group was formed by *C. chiropterus* from La Joya, Veracruz, *C. orculus* from Zacualtipan, Hidalgo, and nine additional undescribed species.

However, *Parra-Olea (2003)* concluded that *C. chiropterus* applies exclusively to the low-elevation populations located in or near the city of Huatusco, Veracruz.

The *Chiropterotriton orculus* complex is represented by a relatively widespread species of the genus. Based on morphological characters, *Cope (1865)* described *C. orculus* as *Spelerpes orculus* from Mexican Table Land, but 4 years later he placed this species in synonymy with *C. chiropterus* (*Cope, 1869*). *Darda's (1994)* allozyme data recognized *C. orculus* as a distinct species, restricting it to two populations. *Parra-Olea (2003)* added one more population to *C. orculus* and emphasized the discordance between allozymes and mtDNA between some populations. Currently, *C. orculus* includes several morphologically uniform populations in the central Trans Mexican Volcanic Belt around Mexico City.

We focus on populations of *Chiropterotriton* from the eastern Trans-Mexican Volcanic Belt and nearby regions of Veracruz and Puebla (Fig. 1). While some of these populations have already been included in allozyme and/or mtDNA analyses, data for others are presented here for the first time. Using a combination of linear morphological measurements, osteological data derived from micro-computed tomography (μCT) scans, and previously published mtDNA and allozyme data we examine the taxonomic status of these populations. We present a phylogenetic hypothesis based on mtDNA which includes all 18 described species plus six undescribed taxa, including populations identified in previous studies as new species within complexes. Based on the molecular data and morphological analyses, we describe five new species. These increase the number of described species from 18 to 23 and still recognize one candidate species not yet described. We redescribe *C. orculus* and *C. chiropterus*, designating neotypes for each, in order to clarify the taxonomic status of nearby populations that resemble one or both of these species in external morphology. Finally, in order to make full comparisons with sympatric taxa for the newly described species, we provide a fuller description of *C. lavae* based on examination of the type series and additional specimens collected subsequently.

## MATERIALS AND METHODS

### Sampling

Animal use was approved by the University of California, Berkeley, IACUC protocol #R093-0205 to DBW. Collection permits were provided by the Secretaría del Medio Ambiente y Recursos Naturales (SEMARNAT): SGPA/DGVS/00947/16, SGPA/DGVS/03038/17 and FAUT-0303, issued to Gabriela Parra-Olea.

### Amplification and sequencing

Whole genomic DNA was extracted from liver, intestine or tail tissue using DNeasy tissue Kit (Qiagen, Valencia, CA, USA). Although a comprehensive molecular analysis of the genus *Chiropterotriton* is beyond the scope of the present work, two mitochondrial fragments of each new species (when available) were sequenced in order to allow comparisons to other members of the genus (Table 1). PCR amplification was done using primers LX12SN1 and LX16S1R for mitochondrial fragment L2; it includes partial sequences from the 12S ribosomal subunit, the tRNA and the large subunit 16S
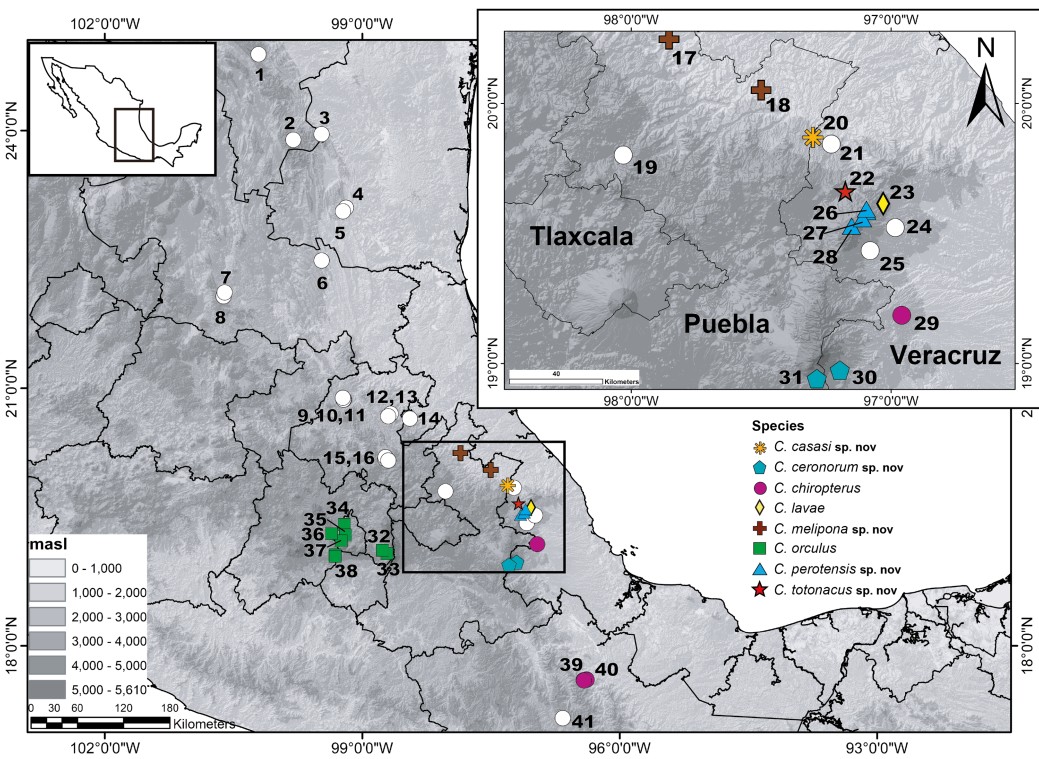

**Figure 1 Geographic distribution of the genus *Chiropterotriton* in Mexico.** Numbers correspond to the following species: (1) *C. priscus*; (2) *C. miquihuanus*; (3) *C. infernalis*; (4) *C. cieloensis*; (5) *C. cracens*; (6) *C. multidentatus* (Cd. Maíz); (7) *C. multidentatus* (Rancho Borbotón); (8) *C. multidentatus* (Sierra de Álvarez); (9) *C. magnipes*; (10) *C. mosaueri*; (11) *C. chondrostega*; (12) *C. terrestris*; (13) *C. arboreus* (Zacualtipán); (14) *C. arboreus* (Zilacatipan); (15) *C. dimidiatus*; (16) *C. chico*; (17) *C. melipona* sp. nov. (Xicotepec); (18) *C. melipona* sp. nov. (Cuetzalan); (19) *Chiropterotriton* sp. G; (20) *C. casasi* sp. nov.; (21) *C. aureus*; (22) *C. totonacus* sp. nov.; (23) *C. lavae*; (24) *C. nubilus* (Tlalnehuayocan); (25) *C. nubilus* (Coxmatla);(26) *C. perotensis* sp. nov. (Las Lajas); (27) *C. perotensis* sp. nov. (Llanillo Redondo); (28) *C. perotensis* sp. nov. (Conejo); (29) *C. chiropterus* (Huatusco); (30) *C. ceronorum* sp. nov. (Xometla); (31) *C. ceronorum* sp. nov. (Texmalaquilla); (32) *C. orculus* (Amecameca); (33) *C. orculus* (Amecameca); (34) *C. orculus* (Ciudad de México); (35) *C. orculus* (Bosque de Tlalpan); (36) *C. orculus* (Desierto de los Leones); (37) *C. orculus* (Ajusco); (38) *C. orculus* (Lagunas de Zempoala); (39) *C. chiropterus* (La Esperanza); (40) *C. chiropterus* (Yolox) and (41) *Chiropterotriton* sp. K.

(*Zhang et al., 2008*). PCR conditions were as follows: 35 cycles at 96 °C (2 min), 55 °C (1 min) and 72 °C (5 min). We also amplified a fragment of the COI gene using primers dgLCO and dgHCO (*Meyer, 2003*). PCR conditions were as follows: 35 cycles at 94 °C (30 s), 50 °C (30 s) and 72 °C (45 s). We cleaned PCR products with ExoSap-IT (USB Corporation, Cleveland, OH, USA) and sequencing reaction with BigDye Terminator v3.1 cycle kit (Applied Biosystems, Foster City, CA, USA). The products were purified using Sephadex G-50 (GE Heathcare, Chicago, IL, USA) and run on an ABI 3730 capillary sequencer at the Instituto de Biología, UNAM.

## Sequence alignment and phylogenetic analyses
Editing and assembly of sequences were performed in Sequencher 5.0.1 (Gene Codes Corporation, Ann Arbor, MI, USA). We used Muscle 3.8 (*Edgar, 2004*) to align L2 and

**Table 1 Voucher information and GenBank numbers.** Voucher information and GenBank numbers for specimens used for phylogenetic analyses from Colección Nacional de Anfibios y Reptiles, Instituto de Biología, UNAM (IBH), Museum of Vertebrate Zoology (MVZ) and Colección de Referencia de Anfibios y Reptiles del Instituto de Ecología, A. C (CARIE). GP, EPR and AMH correspond to field numbers with no voucher available. Numbers in parentheses correspond to geographic location shown in Fig. 1.

| Species | Voucher number | Locality | 16S GenBank | COI GenBank |
|---|---|---|---|---|
| *C. arboreus* | IBH 28191 | Hidalgo: 6.8 km SW (by rd) of Zacualtipán on road to Tianguistengo (13) | MK335386 | MK335232 |
| *C. arboreus* | IBH 22847 | Veracruz: 3.2 km S Zilacatipan (14) | MN914712 | – |
| *C. aureus* | IBH 31042 | Veracruz: 6.5 km (by air) N from Atzalan, ejido de desarrollo urbano Quetzalcoatl (21) | MK335396 | MK335242 |
| *C. aureus* | IBH 31044 | Veracruz: 6.5 km (by air) N from Atzalan, ejido de desarrollo urbano Quetzalcoatl (21) | MK335397 | MK335243 |
| *C. ceronorum* sp. nov. | IBH 30987 | Veracruz: 1.1 km N Xometla (30) | MN914713 | MN920423 |
| *C. ceronorum* sp. nov. | IBH 30988 | Veracruz: 1.1 km N Xometla (30) | MN914714 | MN920424 |
| *C. ceronorum* sp. nov. | MVZ 201387 | Puebla: Santa Cruz de Texmalaquilla (31) | AY522488 | – |
| *C. ceronorum* sp. nov. | MVZ 201389 | Puebla: Santa Cruz de Texmalaquilla (31) | AY522487 | – |
| *C. chico* | MVZ 200679 | Hidalgo: 3.8 km S Mineral del Chico (16) | AY522471 | – |
| *C. chiropterus* | CARIE 0777 | Veracruz: Huatusco (29) | MK335407 | MK335253 |
| *C. chiropterus* | CARIE 0719 | Veracruz: Huatusco (29) | MK335408 | – |
| *C. chiropterus* | IBH 30099 | Oaxaca: San Bernardo, 4.8 km SW (by rd) of La Esperanza on MX 177 (40) | MK335409 | MK335254 |
| *C. chiropterus* | IBH 22736 | Oaxaca: San Bernardo, ca. 5 km SW (by rd) of La Esperanza on MX 175 (40) | MN914715 | – |
| *C. chiropterus* | IBH 30088 | Oaxaca: ca. 400 m from MX 175 on road to San Isidro Yolox (40) | MN914716 | – |
| *C. chiropterus* | IBH 22550 | Oaxaca: La Galera, 11.0 km SW (by rd) of La Esperanza on MX175 (39) | MN914717 | – |
| *C. chiropterus* | GP 088 | Oaxaca: 67 km N Guelatao, trail to San Isidro, La Esperanza (39) | AY522490 | – |
| *C. chondrostega* | IBH 28195 | Hidalgo: 1.0 km S (by rd) of La Encarnación on road to MX 85, Parque Nacional los Marmoles (11) | MN914718 | – |
| *C. chondrostega* | IBH 30098 | Hidalgo: 1.0 km S (by rd) of La Encarnación on road to MX 85, Parque Nacional los Marmoles (11) | MK335383 | MK335229 |
| *C. cieloensis* | IBH 28181 | Tamaulipas: 0.2 km E (by air) of Rancho El Cielo, 6.9 km NNW (by air) of center of Gómez Farías, Reserva de la Biosfera El Cielo (4) | MK335385 | MK335231 |
| *C. cieloensis* | IBH 28190 | Tamaulipas: 0.2 km E (by air) of Rancho El Cielo, 6.9 km NNW (by air) of center of Gómez Farías, Reserva de la Biosfera El Cielo (4) | MN914719 | – |
| *C. cracens* | IBH 28192 | Tamaulipas: Road from Alta Cima to San Jose, 1.3 km NE (by air) of San Jose, Reserva de la Biosfera El Cielo (5) | MK335384 | MK335230 |
| *C. dimidiatus* | IBH 22344 | Hidalgo: 4.3 km N Hwy 105 at Mineral del Monte (15) | MN914720 | – |
| *C. dimidiatus* | IBH 28196 | Hidalgo: 4.1 km S (by rd) of Mineral del Chico on road to Pachuca, Parque Nacional El Chico (15) | MK335390 | MK335236 |
| *C. infernalis* | MVZ 269665 | Tamaulipas: Cueva del Brinco, Conrado Castillo, ca. 43.5 km SW (by rd) of Ejido Guayabas (3) | MK335382 | MK335228 |
| *C. infernalis* | IBH 29575 | Tamaulipas: Conrado Castillo, ca. 43.5 km SW (by rd) of Ejido Guayabas (3) | MN914721 | MN920425 |
| *C. lavae* | IBH 22349 | Veracruz: 200 m N Hwy 140 at La Joya (23) | MN914724 | – |
| *C. lavae* | IBH 22351 | Veracruz: 200 m N Hwy 140 at La Joya (23) | MN914723 | – |
| *C. lavae* | IBH 22360 | Veracruz: 200 m N Hwy 140 at La Joya (23) | MN914722 | – |
| *C. lavae* | IBH 22369 | Veracruz: 200 m N Hwy 140 at La Joya (23) | MK335393 | MK335239 |
| *C. magnipes* | IBH 28176 | Hidalgo: "El Coní", 900 m SSE of center of Durango, Municipio Zimapan, Parque Nacional los Marmoles (9) | MK335387 | MK335233 |

| Species | Voucher number | Locality | 16S GenBank | COI GenBank |
|---|---|---|---|---|
| *C. magnipes* | IBH 30093 | Hidalgo: "El Coní", 900 m SSE of center of Durango, Municipio Zimapan, Parque Nacional los Marmoles (9) | MN914725 | – |
| *C. melipona* sp. nov. | IBH 30112 | Puebla: 7.1 km N (by rd) of center of Cuetzalan on road to Yohualichán (18) | MK335410 | MK335255 |
| *C. melipona* sp. nov. | MVZ 178706 | Puebla: 3.9 km S Xicotepec de Juárez (17) | AY522477 | – |
| *C. melipona* sp. nov. | MVZ 200723 | Puebla: Xicotepec de Juárez (17) | AY522478 | – |
| *C. melipona* sp. nov. | MVZ 178707 | Puebla: Xicotepec de Juárez (17) | AY522479 | – |
| *C. miquihuanus* | IBH 30329 | Nuevo León: 1.8 km S (by rd) of La Encantada on road from La Bolsa to Zaragoza (2) | MK335381 | MK335227 |
| *C. miquihuanus* | IBH 30330 | Nuevo León: 22.6 km N (by rd) of La Bolsa on road to Zaragoza (2) | MN914726 | – |
| *C. mosaueri* | IBH 28179 | Hidalgo: "El Coní", 900 m SSE of center of Durango, Municipio Zimapan, Parque Nacional los Marmoles (10) | MK335388 | MK335234 |
| *C. multidentatus* | IBH 28177 | San Luis Potosí: Cueva el Madroño, 900 m NW (by air) of entrance to Valle de los Fantasmas on MX 70, Sierra de Alvarez (8) | MK335416 | – |
| *C. multidentatus* | IBH 30102 | San Luis Potosí: Cueva el Madroño, 900 m NW (by air) of entrance to Valle de los Fantasmas on MX 70, Sierra de Alvarez (8) | MK335417 | – |
| *C. multidentatus* | IBH 28193 | San Luis Potosí: 26.2 km E (by rd) of center of Ciudad del Maíz on MX 80, at turnoff to RMO Las Antenas San Luis Potosí (6) | MK335412 | – |
| *C. multidentatus* | IBH 30104 | San Luis Potosí: 26.2 km E (by rd) of center of Ciudad del Maíz on MX 80, at turnoff to RMO Las Antenas San Luis Potosí (6) | MK335414 | – |
| *C. multidentatus* | IBH 28194 | San Luis Potosí: 26.2 km E (by rd) of center of Ciudad del Maíz on MX 80, at turnoff to RMO Las Antenas San Luis Potosí (6) | MK335413 | – |
| *C. multidentatus* | IBH 23111 | San Luis Potosí: Rancho Borbortón (7) | MK335415 | – |
| *C. nubilus* | IBH 31048 | Veracruz: 8.2 km W from Xico, Coxmatla (25) | MK335402 | MK335248 |
| *C. nubilus* | CARIE 0740 | Veracruz: Bosque Rancho Viejo, Tlalnehuayocan (24) | MK335406 | MK335252 |
| *C. orculus* | IBH 30765 | Estado de México: Amecameca, road to Popocatepetl volcano (33) | MK335391 | MK335237 |
| *C. orculus* | IBH 30746 | Estado de México: Amecameca, road to Popocatepetl volcano (32) | MK335392 | MK335238 |
| *C. orculus* | IBH 30943 | Estado de México: Amecameca, road to Popocatepetl volcano (33) | MN914727 | – |
| *C. orculus* | IBH 22866 | Estado de México: Amecameca, road to Popocatepetl volcano (32) | MN914728 | – |
| *C. orculus* | IBH 22210 | Ciudad de Mexico: Colonia Prolongación Miguel Hidalgo (34) | MN914729 | – |
| *C. orculus* | AMH 300 | Ciudad de Mexico: Desierto de los Leones (36) | MN914730 | – |
| *C. orculus* | EPR | Ciudad de Mexico: Bosque de Tlalpan (35) | MN914731 | – |
| *C. orculus* | IBH 29851 | Morelos: Parque Nacional Lagunas de Zempoala (38) | MN914732 | – |
| *C. orculus* | IBH 31023 | Morelos: Parque Nacional Lagunas de Zempoala (38) | MN914733 | – |
| *C. orculus* | IBH 26478 | Ciudad de Mexico: El Ajusco, km 29.4 from Picacho-Ajusco road (37) | MN914734 | – |
| *C. orculus* | MVZ 138672 | Ciudad de Mexico: Desierto de Los Leones National Park, 8.8 km [rd.] SW La Venta by Mexico Hwy. 15 (36) | AY522442 | – |
| *C. perotensis* sp. nov. | IBH 22395 | Veracruz: 15.9 km on microondas road, Las Vigas (26) | MN914735 | – |
| *C. perotensis* sp. nov. | IBH 22568 | Veracruz: Microondas las Lajas (26) | KP886893 | – |
| *C. perotensis* sp. nov. | IBH 23066 | Veracruz: 15.9 km on microondas road, Las Vigas (26) | MN914736 | – |
| *C. perotensis* sp. nov. | IBH 31032 | Veracruz: Conejo, road to the peak of Cofre de Perote (28) | MN914743 | – |
| *C. perotensis* sp. nov. | IBH 31033 | Veracruz: Conejo, road to the peak of Cofre de Perote (28) | MN914744 | – |
| *C. perotensis* sp. nov. | IBH 31034 | Veracruz: Conejo, road to the peak of Cofre de Perote (28) | MN914737 | – |
| *C. perotensis* sp. nov. | IBH 31035 | Veracruz: Conejo, road to the peak of Cofre de Perote (28) | MN914738 | MN920426 |
| *C. perotensis* sp. nov. | IBH 31036 | Veracruz: Conejo, road to the peak of Cofre de Perote (28) | MN914739 | – |

| Species | Voucher number | Locality | 16S GenBank | COI GenBank |
|---|---|---|---|---|
| *C. perotensis* sp. nov. | IBH 31037 | Veracruz: 2 km (by air) al NE de Llanillo redondo camino a Valle Alegre (27) | MN914740 | – |
| *C. perotensis* sp. nov. | IBH 31038 | Veracruz: 2 km (by air) al NE de Llanillo redondo camino a Valle Alegre (27) | MN914741 | – |
| *C. perotensis* sp. nov. | IBH 31039 | Veracruz: 2 km (by air) al NE de Llanillo redondo camino a Valle Alegre (27) | MN914742 | MN920427 |
| *C. priscus* | IBH 22367 | Nuevo León: 19.4 km W 18 de Marzo, Cerro Potosí (1) | MK335380 | MK335226 |
| *C. terrestris* | GP 215 | Hidalgo: 5.3 km N Hwy 105 at Zacualtipan (12) | MK335389 | MK335235 |
| *C. totonacus* sp. nov. | IBH 31030 | Veracruz: El Polvorín, 5 km SW of Villa Aldama (22) | MN914745 | MN920428 |
| *C. totonacus* sp. nov. | IBH 31031 | Veracruz: El Polvorín, 5 km SW of Villa Aldama (22) | MN914746 | MN920429 |
| *Chiropterotriton* sp. G | MVZ 178700 | Puebla: 4 km S Chignahuapan (19) | AY522480 | – |
| *Chiropterotriton* sp. G | MVZ 178703 | Puebla: 4 km S Chignahuapan (19) | AY522481 | – |
| *Chiropterotriton* sp. K | MVZ 173231 | Oaxaca: Cerro San Felipe (41) | AY522493 | – |
| *Aquiloeurycea cephalica* | IBH 30253 | Hidalgo: 1.0 km S (by rd) of La Encarnación on road to MX 85, Parque Nacional los Mármoles | MK335378 | – |
| *Thorius* sp. | IBH 30942 | Oaxaca: Santa María Chilchotla, Sierra Mazateca | MN914747 | – |

COI sequences. The alignment for the L2 fragment included 35 *Chiropterotriton* samples sequenced in this study, 40 sequences available on GenBank from previous studies (*Parra-Olea, 2003*; *Rovito et al., 2015a*; *García-Castillo et al., 2018*) and two additional sequences from *Aquiloeurycea cephalica* and *Thorius* sp. as outgroups. The alignment for COI included seven sequences from this study and 21 from GenBank (*García-Castillo et al., 2018*). All sequence information is shown in Table 1. We used Mesquite v3.40 (*Maddison & Maddison, 2018*) to concatenate and review the data matrix. We used PartitionFinder v1.0 (*Lanfear et al., 2012*) to select substitution model and a partitioning scheme using the Bayesian Information Criterion (BIC). We ran Maximum Likelihood and Bayesian inference through the CIPRES data portal (*Miller, Pfeiffer & Schwartz, 2010*) for phylogenetic analyses; RAxML v8.2 (*Stamatakis, 2014*) to generate a Maximum Likelihood tree, with 1,000 bootstrap replicates as nodal support; and MrBayes v3.2 (*Huelsenbeck & Ronquist, 2001*) for Bayesian inference, with 20 million generations, sampling every 1,000 generations, with four chains to obtain a majority consensus tree. Finally, we used Tracer v.1.7 (*Rambaut et al., 2018*) to review the convergence and stability of the chains.

## Morphological analyses and species descriptions

Species descriptions largely follow the format used by *Lynch & Wake (1989)* for species of Neotropical plethodontids and include many of the same basic characters and measurements, including coloration and external measurements. We used an electronic vernier calipers to measure 11 characters: snout-vent length (SVL), tail length (TL), axilla-groin distance (AX), forelimb length (FLL), hind limb length (HLL), snout-to-gular-fold distance (head length, HL), head width at angle of jaw (HW), head depth (HD), shoulder width (SW), internarial distance (IN) and right foot width (FW). In order to obtain an index for nostril shape, we used an ocular micrometer to measure the longest and

shortest nostril dimensions (nostril length, NL; nostril width, NW) and we calculated a ratio of nostril dimensions (ND = NL/NW). We also counted ankylosed premaxillary (PMT), maxillary (MT) and vomerine teeth (VT). We present counts for PMT and MT together because of the difficulty in distinguishing them in some specimens. We also measured limb interval (LI) as the number of costal folds between adpressed limbs. Positive values equal the number of folds visible between adpressed limbs that don't meet or overlap; negative values denote overlap between limbs. We treat males and females separately to evaluate the extent of sexual dimorphism (Table 2). Finally, 12 additional measurements were obtained for each holotype: anterior rim of orbit to snout, eyelid length, eyelid width, horizontal orbital diameter, interorbital distance, length of third (longest) toe, length of fifth toe, projection of snout beyond mandible, snout to anterior angle of vent, snout to forelimb, tail depth at base, and tail width at base.

In addition, μCT scans were used to prepare osteological accounts based primarily on the cranial characters and character states defined by *Darda & Wake (2015*; Table 3; Fig. 2). Scans made at the University of Texas High-Resolution X-ray CT Facility are archived in a digital repository and may be viewed online via the Internet links provided below. The complete scans include the ossified forelimb skeleton as well as the bony skull, but only skulls are illustrated here.

We examined 123 individuals from the eight species of principal interest and used published data for comparisons to other species of *Chiropterotriton*. The latter species were chosen for comparison based on either geographic or phylogenetic closeness. All material, including holotypes or neotypes designated below, is deposited at the National Museum of Natural History, Smithsonian Institution, Washington, DC, USA (USNM) and the Museum of Vertebrate Zoology, University of California Berkeley, USA (MVZ) collections (Appendix I).

The electronic version of this article in Portable Document Format (PDF) will represent a published work according to the International Commission on Zoological Nomenclature (ICZN), and hence the new names contained in the electronic version are effectively published under that Code from the electronic edition alone. This published work and the nomenclatural acts it contains have been registered in ZooBank, the online registration system for the ICZN. The ZooBank LSIDs (Life Science Identifiers) can be resolved and the associated information viewed through any standard web browser by appending the LSID to the prefix http://zoobank.org/. The LSID for this publication is: [9B4B9DFF-E12B-430D-A541-BA0EBB9B90E6]. The online version of this work is archived and available from the following digital repositories: PeerJ, PubMed Central and CLOCKSS.

## RESULTS

Our phylogenetic reconstruction was based on two mitochondrial fragments, with a final matrix of 2,143 bp (gaps included) from 75 individuals that includes all described species of *Chiropterotriton*. Both ML and Bayesian analyses show two main clades in the genus (Fig. 3). The first main clade, with rather low support (BS = 54, not recovered in Bayesian tree), includes 12 species that correspond to the north-central distributions: *C. cracens, C, cieloensis, C. arboreus, C. multidentatus, C. infernalis, C. mosaueri,*

**Table 2 Mean ± standard deviation (above) and range (below) of morphometric variables.** Mean ± standard deviation (above) and range (below) of morphometric variables from males and females of *C. aureus*, *C. nubilus*, *C. ceronorum*, *C. perotensis*, *C. totonacus*, C. *melipona*, *C. casasi*, *C. chiropterus*, *C. orculus* and *C. lavae*. Measurements are given in millimeters (mm), except TL/SLV (proportional value), LI (limb interval), and tooth counts.

| Males | C. aureus N = 1 | C. nubilus N = 1 | C. ceronorum sp. nov. N = 10 | C. perotensis sp. nov. N = 12 | C. totonacus sp. nov. N = 10 | C. melipona sp. nov. N = 4 | C. casasi sp. nov. N = 4 | C. chiropterus N = 8 | C. orculus N = 10 | C. lavae N = 10 |
|---|---|---|---|---|---|---|---|---|---|---|
| SVL | 28.5 | 29.4 | 33.9 ± 1.54 (30.6–36.2) | 29.7 ± 1.92 (26.5–32.8) | 35.7 ± 1.96 (32.0–38.6) | 29.2 ± 2.25 (26.4–31.4) | 37.8 ± 3.10 (34.5–42.0) | 37.5 ± 0.98 (36.1–38.8) | 35.9 ± 1.36 (33.6–38.9) | 32.4 ± 0.92 (31.0–33.8) |
| TL | 36.5 | 40.2 | 33.9 ± 1.99 (30.4–37.7) | 30.9 ± 3.06 (26.0–35.2) N = 8 | 41.1 ± 3.20 (34.3–44.9) N = 9 | 33.9 ± 3.37 (31.0–38.2) | 39.1 ± 3.29 (36.8–42.9) N = 3 | 47.3 ± 3.24 (42.6–52.3) N = 7 | 36.6 ± 2.87 (33.3–41.0) N = 9 | 38.5 ± 2.11 (36.2–42.3) |
| TL/ SVL | 1.28 | 1.37 | 1.00 ± 0.06 (0.89–1.12) | 1.03 ± 0.08 (0.92–1.16) N = 8 | 1.16 ± 0.10 (0.92–1.24) N = 9 | 1.16 ± 0.05 (1.10–1.22) | 1.04 ± 0.13 (0.90–1.15) N = 3 | 1.25 ± 0.08 (1.13–1.38) N = 7 | 1.02 ± 0.08 (0.86–1.15) N = 9 | 1.19 ± 0.06 (1.11–1.27) |
| AX | 15.5 | 15.9 | 16.9 ± 0.70 (15.5–17.9) | 15.5 ± 0.93 (14.2–17.0) | 18.3 ± 1.30 (16.7–20.4) | 15.7 ± 1.30 (14.0–17.0) | 19.8 ± 0.46 (19.4–20.4) | 19.6 ± 0.59 (18.7–20.8) | 18.6 ± 1.04 (17.1–20.5) | 16.2 ± 0.87 (14.7–17.4) |
| FLL | 5.9 | 6.4 | 8.9 ± 0.69 (7.2–10.0) | 6.8 ± 0.59 (5.5–7.8) | 10.0 ± 0.72 (8.9–10.9) | 6.3 ± 0.86 (5.1–7.0) | 9.9 ± 0.59 (9.4–10.7) | 9.1 ± 0.44 (8.2–9.5) | 8.9 ± 0.65 (7.4–9.6) | 9.3 ± 0.59 (8.4–10.2) |
| HLL | 7.5 | 7.1 | 9.4 ± 0.83 (7.5–10.3) | 7.2 ± 0.61 (6.1–8.2) | 11.0 ± 1.00 (9.4–12.2) | 7.2 ± 0.83 (6.1–7.9) | 11.5 ± 0.74 (11.1–12.6) | 10.3 ± 0.47 (9.5–10.8) | 9.3 ± 0.64 (8.2–10.4) N = 9 | 9.9 ± 0.72 (8.5–11.0) |
| HL | 6.4 | 6.6 | 7.5 ± 0.55 (6.3–8.2) | 6.6 ± 0.33 (6.1–7.1) | 8.5 ± 0.64 (7.7–9.5) | 6.3 ± 0.52 (5.5–6.6) | 8.3 ± 0.60 (7.5–8.8) | 8.1 ± 0.41 (7.7–8.9) | 7.4 ± 0.47 (6.7–8.1) | 7.5 ± 0.33 (7.2–8.1) |
| HW | 4.0 | 4.0 | 5.1 ± 0.35 (4.3–5.5) | 4.2 ± 0.18 (3.9–4.5) | 5.2 ± 0.29 (4.8–5.7) | 4.3 ± 0.33 (3.9–4.6) | 5.8 ± 0.45 (5.3–6.3) | 5.6 ± 0.22 (5.4–6.0) | 5.0 ± 0.35 (4.5–5.5) | 4.9 ± 0.31 (4.5–5.6) |
| HD | 1.8 | 2.0 | 2.5 ± 0.17 (2.1–2.7) | 2.0 ± 0.18 (1.7–2.3) | 2.4 ± 0.34 (2.1–3.3) | 2.3 ± 0.22 (2.1–2.6) | 2.5 ± 0.28 (2.2–2.8) | 2.7 ± 0.07 (2.6–2.8) | 2.4 ± 0.13 (2.2–2.7) | 2.5 ± 0.19 (2.3–2.9) |
| SW | 3.4 | 3.4 | 3.6 ± 0.29 (3.0–3.9) | 2.7 ± 0.28 (2.3–3.4) | 3.6 ± 0.28 (3.2–4.0) | 3.3 ± 0.26 (3.1–3.7) | 3.5 ± 0.37 (3.1–3.8) | 4.0 ± 0.35 (3.2–4.4) | 3.4 ± 0.30 (3.1–4.0) | 3.1 ± 0.30 (2.6–3.5) |
| IN | 1.0 | 1.2 | 2.3 ± 0.18 (2.0–2.6) | 1.7 ± 0.26 (1.1–2.0) | 2.4 ± 0.23 (1.9–2.7) | 1.4 ± 0.13 (1.3–1.6) | 2.1 ± 0.30 (1.7–2.4) | 1.9 ± 0.13 (1.7–2.1) | 2.2 ± 0.19 (1.9–2.5) | 2.3 ± 0.20 (1.9–2.5) |
| FW | 2.4 | 2.6 | 3.8 ± 0.44 (2.9–4.6) | 2.6 ± 0.33 (2.1–3.1) | 4.2 ± 0.45 (3.5–4.9) | 2.4 ± 0.27 (2.2–2.8) | 3.7 ± 0.19 (3.6–4.0) | 3.7 ± 0.33 (3.3–4.4) | 3.2 ± 0.22 (2.8–3.5) | 3.7 ± 0.39 (3.1–4.2) |
| LI | 2.0 | 2.0 | 0.0 ± 0.41 (−0.5 to 1.0) | 2.5 ± 0.67 (1.0–3.0) | −0.6 ± 0.70 (−1.0 to 1.0) | 2.3 ± 0.29 (2.0–2.5) | 0.8 ± 0.50 (0.0–1.0) | 0.3 ± 0.53 (−0.5 to 1.0) | 1.9 ± 0.88 (0.0–3.0) | −0.6 ± 0.52 (−1.0 to 0.0) |
| PMT | 4.0 | 7.0 | 3.4 ± 0.97 (3.0–6.0) | 2.8 ± 0.97 (0.0–4.0) | 4.8 ± 0.63 (4.0–6.0) | 2.3 ± 1.50 (1.0–4.0) | 3.5 ± 1.29 (2.0–5.0) | 3.6 ± 1.30 (2.0–5.0) | 2.7 ± 0.82 (2.0–4.0) | 3.3 ± 2.00 (0.0–6.0) |
| MT | 10.0 | 13.0 | 11.0 ± 3.30 (7.0–18.0) | 7.2 ± 4.73 (2.0–17.0) | 32.9 ± 7.80 (18.0–48.0) | 9.5 ± 2.38 (7.0–12.0) | 9.0 ± 2.94 (6.0–13.0) | 12.6 ± 3.46 (9.0–17.0) | 8.2 ± 2.25 (5.0–11.0) | 7.0 ± 2.71 (1.0–10.0) |
| VT | 15.0 | 10.0 | 13.0 ± 2.05 (11.0–17.0) | 9.0 ± 1.65 (7.0–12.0) | 11.6 ± 1.90 (10.0–15.0) | 11.0 ± 2.94 (8.0–15.0) | 9.0 ± 1.41 (8.0–11.0) | 10.6 ± 1.06 (9.0–12.0) | 8.6 ± 1.90 (5.0–11.0) | 8.9 ± 1.10 (7.0–10.0) |

| Females | C. aureus N = 3 | C. nubilus N = 2 | C. ceronorum sp. nov. N = 10 | C. perotensis sp. nov. N = 8 | C. totonacus sp. nov. N = 10 | C. melipona sp. nov. N = 3 | C. casasi sp. nov. N = 1 | C. chiropterus N = 4 | C. orculus N = 10 | C. lavae N = 9 |
|---|---|---|---|---|---|---|---|---|---|---|
| SVL | 26.8 ± 0.86 (26.0–27.7) | 30.5 ± 3.89 (27.7–33.2) | 34.9 ± 1.53 (33.3–38.4) | 31.7 ± 2.19 (27.4–34.3) | 35.5 ± 1.90 (31.8–38.3) | 28.5 ± 1.36 (27.1–29.8) | 40.9 | 33.5 ± 2.55 (30.7–36.7) | 39.0 ± 2.70 (34.9–43.0) | 31.6 ± 2.46 (27.9–34.9) |
| TL | 31.1 ± 1.41 (30.1–32.1) | 34.3 ± 5.16 (30.6–37.9) | 33.9 ± 2.82 (28.5–38.2) | 31.5 ± 3.31 (27.0–37.3) N = 7 | 42.6 ± 5.08 (36.3–49.2) N = 6 | 32.3 ± 2.26 (30.7–33.9) N = 2 | 34.0 br | 39.5 ± 2.35 (37.0–42.6) | 39.2 ± 3.64 (34.7–44.7) N = 9 | 32.5 ± 4.89 (25.7–40.1) |

*(Continued)*

| Females | C. aureus N = 3 | C. nubilus N = 2 | C. ceronorum sp. nov. N = 10 | C. perotensis sp. nov. N = 8 | C. totonacus sp. nov. N = 10 | C. melipona sp. nov. N = 3 | C. casasi sp. nov. N = 1 | C. chiropterus N = 4 | C. orculus N = 10 | C. lavae N = 9 |
|---|---|---|---|---|---|---|---|---|---|---|
| TL/SVL | 1.16 ± 0.00 (1.16–1.16) | 1.12 ± 0.03 (1.10–1.14) | 0.97 ± 0.07 (0.85–1.07) | 1.00 ± 0.11 (0.79–1.11) N = 7 | 1.20 ± 0.13 (1.06–1.38) N = 6 | 1.11 ± 0.11 (1.03–1.18) N = 2 | – | 1.19 ± 0.12 (1.01–1.26) | 1.02 ± 0.08 (0.87–1.12) N = 9 | 1.02 ± 0.10 (0.85–1.15) |
| AX | 15.0 ± 0.49 (14.7–15.6) | 16.4 ± 2.69 (14.5–18.3) | 18.5 ± 0.95 (17.1–20.0) | 16.6 ± 1.58 (13.6–19.2) | 18.7 ± 0.95 (17.3–20.1) | 15.8 ± 0.59 (15.4–16.5) | 20.3 | 18.5 ± 2.27 (15.4–20.7) | 21.2 ± 1.58 (18.6–23.2) | 16.3 ± 1.68 (13.9–18.5) |
| FLL | 5.3 ± 0.42 (4.8–5.6) | 6.5 ± 0.28 (6.3–6.7) | 8.6 ± 0.38 (8.1–9.3) | 6.7 ± 0.61 (5.9–7.5) | 9.7 ± 0.85 (8.7–11.3) | 6.5 ± 0.72 (6.0–7.3) | 10.6 | 7.8 ± 0.48 (7.1–8.2) | 8.9 ± 0.63 (7.6–10.0) | 8.2 ± 0.72 (7.1–9.5) |
| HLL | 6.7 ± 0.35 (6.4–7.1) | 7.2 ± 0.14 (7.1–7.3) | 8.9 ± 0.70 (7.3–9.9) | 7.1 ± 0.66 (6.1–8.2) | 10.8 ± 0.93 (9.3–12.5) | 7.4 ± 0.58 (7.1–8.1) | 12.0 | 8.9 ± 0.31 (8.4–9.1) | 9.5 ± 0.57 (8.6–10.4) | 8.8 ± 0.73 (7.5–9.8) |
| HL | 6.0 ± 0.31 (5.7–6.3) | 7.4 ± 0.99 (6.7–8.1) | 7.1 ± 0.29 (6.6–7.6) | 6.7 ± 0.31 (6.2–7.2) | 7.6 ± 0.38 (7.0–8.1) | 6.4 ± 0.60 (5.8–7.0) | 8.6 | 7.3 ± 0.56 (6.5–7.8) | 8.0 ± 0.52 (7.4–8.9) | 7.0 ± 0.42 (6.3–7.6) |
| HW | 3.6 ± 0.10 (3.5–3.7) | 4.4 ± 0.14 (4.3–4.5) | 5.1 ± 0.21 (4.7–5.3) | 4.4 ± 0.21 (4.1–4.6) | 5.2 ± 0.22 (5.0–5.6) | 4.2 ± 0.25 (4.0–4.5) | 5.9 | 4.8 ± 0.21 (4.5–5.0) | 5.2 ± 0.29 (4.7–5.6) | 4.7 ± 0.30 (4.1–5.0) |
| HD | 1.8 ± 0.02 (1.8–1.8) | 2.0 ± 0.07 (1.9–2.0) | 2.4 ± 0.12 (2.3–2.6) | 2.2 ± 0.17 (2.0–2.5) | 2.3 ± 0.17 (2.0–2.6) | 2.4 ± 0.12 (2.3–2.5) | 2.6 | 2.5 ± 0.14 (2.3–2.6) | 2.6 ± 0.32 (2.3–3.4) | 2.3 ± 0.18 (2.1–2.7) |
| SW | 3.1 ± 0.17 (3.0–3.3) | 3.3 ± 0.28 (3.1–3.5) | 3.7 ± 0.24 (3.3–4.1) | 3.1 ± 0.22 (2.6–3.3) | 3.6 ± 0.17 (3.4–3.9) | 3.2 ± 0.15 (3.1–3.4) | 3.3 | 3.6 ± 0.38 (3.3–4.1) | 3.9 ± 0.46 (3.4–4.8) | 3.3 ± 0.33 (2.8–3.8) |
| IN | 1.1 ± 0.06 (1.0–1.1) | 1.2 ± 0.02 (1.2–1.2) | 1.9 ± 0.15 (1.5–2.1) | 1.8 ± 0.14 (1.6–2.0) | 2.2 ± 0.19 (2.0–2.5) | 1.4 ± 0.06 (1.4–1.5) | 2.3 | 1.7 ± 0.38 (1.4–2.1) | 2.1 ± 0.25 (1.7–2.5) | 1.8 ± 0.13 (1.6–2.0) |
| FW | 1.8 ± 0.21 (1.6–2.0) | 2.3 ± 0.57 (1.9–2.7) | 3.5 ± 0.40 (2.8–3.9) | 2.6 ± 0.24 (2.2–3.0) | 4.0 ± 0.52 (3.3–4.8) | 2.6 ± 0.38 (2.3–3.0) | 3.7 | 3.1 ± 0.37 (2.6–3.5) | 3.4 ± 0.37 (2.6–3.9) | 3.3 ± 0.27 (3.0–3.7) |
| LI | 2.3 ± 0.58 (2.0–3.0) | 1.5 ± 0.71 (1.0–2.0) | 1.5 ± 0.41 (1.0–2.0) | 3.3 ± 0.71 (2.0–4.0) | 0.0 ± 0.67 (−1.0 to 1.0) | 1.8 ± 0.76 (1.0–2.5) | 1.0 | 2.0 ± 0.41 (1.5–2.5) | 2.9 ± 0.32 (2.0–3.0) | 0.6 ± 0.73 (0.0–2.0) |
| PMT | 6.3 ± 0.58 (6.0–7.0) | 6.5 ± 0.71 (6.0–7.0) | 7.4 ± 0.97 (6.0–9.0) | 6.1 ± 2.17 (4.0–11.0) | 7.0 ± 1.05 (6.0–9.0) | 7.0 ± 1.73 (6.0–9.0) | 6.0 | 6.3 ± 1.26 (5.0–8.0) | 7.1 ± 0.88 (6.0–8.0) | 7.2 ± 1.99 (4.0–10.0) |
| MT | 38.3 ± 1.53 (37.0–40.0) | 41.5 ± 2.12 (40.0–43.0) | 47.7 ± 7.26 (36.0–56.0) | 27.9 ± 5.03 (19.0–36.0) | 52.6 ± 4.50 (45.0–60.0) | 31.0 ± 5.20 (25.0–34.0) | 30.0 | 48.0 ± 7.94 (42.0–57.0) N = 3 | 28.8 ± 4.05 (23.0–35.0) | 20.8 ± 6.69 (13.0–36.0) |
| VT | 12.3 ± 1.53 (11.0–14.0) | 13.5 ± 0.71 (13.0–14.0) | 15.9 ± 2.69 (13.0–22.0) | 11.1 ± 1.13 (10.0–13.0) | 13.7 ± 2.11 (9.0–17.0) | 13.0 ± 5.29 (9.0–19.0) | 13.0 | 12.5 ± 2.38 (10.0–15.0) | 12.0 ± 1.94 (9.0–15.0) | 11.4 ± 2.30 (8.0–15.0) |

*C. chondrostega, C. magnipes, C. priscus, C. miquihuanus, C. terrestris* and *C. chico*. The second main clade with strong support (Bootstrap, BS = 100 and Posterior Probability, PP = 1.0) also includes 12 species, but with central-southern distributions: *C. dimidiatus, C. totonacus* sp. nov., *C. ceronorum* sp. nov., *C. lavae, C. perotensis* sp. nov., *C.* sp. K, *C.* sp. G, *C. orculus, C. melipona* sp. nov., *C. aureus, C. nubilus* and *C. chiropterus.* The major clade is the main subject of the following species descriptions and includes four of the five new species that were initially proposed by *Darda (1994)* as *Chiropterotriton* sp. E, *C.* sp. F, *C.* sp. H and *C.* sp. I. This clade also contains the two redescribed species, *C. orculus* and *C. chiropterus*, as well as *C. lavae.* One of the species we describe below, *C. casasi* sp. nov., has not been found since the collection of the type series in 1969 and no tissue has been available for molecular analyses. Each species is diagnosed by

**Table 3 Cranial osteological variation among *Chiropterotriton* species based on characters and character states defined by Darda & Wake (2015).** Each species is represented by a single μCT-scanned specimen except *C. chiropterus*, for which there are an additional four cleared-and-stained (c&s) specimens. States that are not observed in these specimens are omitted; for example, character 6, state c. All specimens show the same state for characters 11 (squamosal process absent) and 12 (vomer preorbital process present). Each species name is followed by the specimen's museum catalog number, sex (F, female; M, male) and snout-vent length. Instances in which two states are listed for a given character (*) represent right-left asymmetry in that specimen.

| Species | 1. Septomaxilla development (a) absent | (b) present | 2. Nasal-premaxilla articulation (a) separate | (b) abut | (c) overlap | 3. Nasal-maxilla articulation (a) separate | (b) abut | (c) overlap | 4. Nasal-prefrontal articulation (a) separate | (b) abut | (c) overlap | 5. Nasal-frontal articulation (a) separate | (b) overlap | 6. Frontal-frontal articulation (a) separate | (b) abut | (d) interdigitate | 7. Parietal-parietal articulation (a) separate | (b) abut | (d) interdigitate | 8. Frontoparietal fontanel (a) extensive | (b) reduced | (c) absent | 9. Parietal process (a) absent | (b) present | 10. Otic process (a) absent | (b) one | (c) two |
|---|---|---|---|---|---|---|---|---|---|---|---|---|---|---|---|---|---|---|---|---|---|---|---|---|---|---|---|
| *C. ceronorum,* sp. nov. USNM 224212, M, 36.2 mm | | X | | X | | | | X | | X | | | X | | X* | X* | X | X | | | X | | X | | | X | |
| *C. perotensis,* sp. nov. MVZ 200693, F, 31.1 mm | X | | X | | | | X | | X* | X* | | | X | | X* | X* | X* | X* | | X | | | X | | X | | |
| *C. totonacus,* sp. nov. MVZ 163945, F, 35.8 mm | X | | X | | | X | | | X | | | X | | | X | | X | | | | X | | X | | | X | |
| *C. melipona,* sp. nov. MVZ 178706, M, 28.5 mm | | X | X | | | | X | | | X | | X | | X | | | X | | | X | | | X | | | | X |
| *C. casasi,* sp. nov. MVZ 92874, M, 42.0 mm | | X | | | X | | | X | | X | | | X | | | X | | | X | | | X | | X | | | X |
| *C. chiropterus* MVZ 85602, M, 38.9 mm | | X | | | X | | | X | | | X | | X | | | X | | | X | | | X | X | | | | X |
| *C. chiropterus,* c&s MVZ 85596, M, 40.0 mm | | X | | X | | | | X | | X | | | X | | | X | | | X | | | X | X | | | | X |
| *C. chiropterus,* c&s MVZ 85632, F, 34 mm | | | | | | | | | | | | | | | | X | X* | | X* | | X | | X | | | | X |
| *C. chiropterus,* c&s MVZ 85594, M, 36 mm | | X | | | X | | | X | | | X | | X | | | X | | | X | | | X | X | | | | X |

(Continued)

| Species | Character | | | | | | | | | | | | | | | | | | | | | | | | | | | | |
|---|---|---|---|---|---|---|---|---|---|---|---|---|---|---|---|---|---|---|---|---|---|---|---|---|---|---|---|---|---|
| | 1. Septomaxilla development | | 2. Nasal-premaxilla articulation | | | 3. Nasal-maxilla articulation | | | 4. Nasal-prefrontal articulation | | | 5. Nasal-frontal articulation | | 6. Frontal-frontal articulation | | | 7. Parietal-parietal articulation | | | 8. Frontoparietal fontanel | | | 9. Parietal process | | 10. Otic process | | |
| | (a) ab-sent | (b) pre-sent | (a) sepa-rate | (b) abut | (c) over-lap | (a) sepa-rate | (b) abut | (c) over-lap | (a) sepa-rate | (b) abut | (c) over-lap | (a) sepa-rate | (b) over-lap | (a) sepa-rate | (b) abut | (d) inter-digitate | (a) sepa-rate | (b) abut | (d) inter-digitate | (a) exten-sive | (b) re-duced | (c) ab-sent | (a) ab-sent | (b) pre-sent | (a) ab-sent | (b) one | (c) two |
| *C. chiropterus*, c&s | X | | | X | | | | X | | | X | | X | | | X | | | X | | | X | X | | | | X |
| MVZ 85613, M, 37.7 mm | | | | | | | | | | | | | | | | | | | | | | | | | | | |
| *C. orculus* | | X | | X | | | X | | X | | | | X | X* | X* | | X* | X* | | | X | | X | | X | | |
| MVZ 138783, M, 38.9 mm | | | | | | | | | | | | | | | | | | | | | | | | | | | |
| *C. lavae* | X | | | X | | | | X | X | | | | X | X* | | X* | | | X | | X | X | X | | | | X |
| MVZ 163912, M, 33.8 mm | | | | | | | | | | | | | | | | | | | | | | | | | | | |

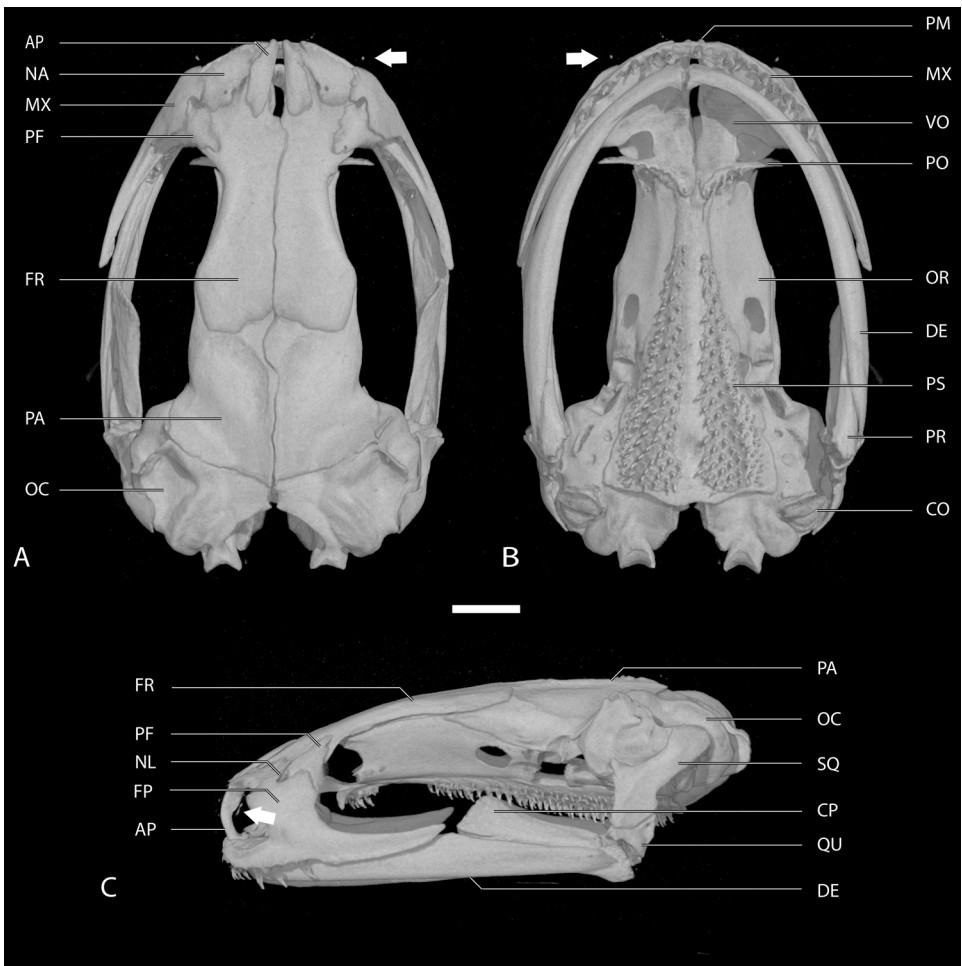

**Figure 2 Skull of the holotype of *Chiropterotriton casasi* sp. nov. seen in (A) dorsal, (B) ventral and (C) lateral views.** Images are derived from a µCT scan of MVZ 92874, an adult male. Arrows point to the septomaxillary bone. Abbreviations: AP, ascending process of the premaxilla; CO, columella; CP, coronoid process of the prearticular; DE, dentary; FP, facial process of the maxilla; FR, frontal; MX, maxilla; NA, nasal; NL, foramen of the nasolacrimal duct; OC, otic capsule; OR, orbitosphenoid; PA, parietal; PF, prefrontal; PM, premaxilla; PO, preorbital process of the vomer; PR, prearticular; PS, parasphenoid; QU, quadrate; SQ, squamosal; VO, vomer. Scale bar, 1 mm.

morphological characters through morphometric and osteological comparisons (Tables 2 and 3).

### *Chiropterotriton ceronorum* sp. nov.
Ceron Family Salamander, Salamandra de los Cerón
Figures 4A–4C, 5A, 6B, 7B, 8B.

Chresonymy
*Chiropterotriton chiropterus* (part)—*Gadow, 1905*.
*Chiropterotriton* sp. I.—*Darda, 1994* (population 22); *Parra-Olea, 2003*; *Raffaëlli, 2007*; *Raffaëlli, 2013*; *Rovito & Parra-Olea, 2015*; *García-Castillo et al., 2017*; *García-Castillo et al., 2018*.

**Holotype**: USNM 224212, an adult male from ca. 1 km NE Santa Cruz Texmalaquilla (4.7 mi by road NE of Atzitzintla), on south slope of Pico de Orizaba, Puebla, Mexico, 3,110 masl, 18.9484° N, 97.2802° W. Collected 3 September 1975 by R.W. McDiarmid.

**Paratypes**: Twenty specimens, all from Puebla, Mexico. Ten males: MVZ 201393, Santa Cruz Texmalaquilla, S side of Mt. Orizaba; USNM 224202, 224207–08, 224211, 224218–20, 224230 and 224236, same data as holotype. Ten females: USNM 224240–41, 224247, 224250, 224252–53, 224257, 224259 and 224275–76, same data as holotype.

**Referred specimens**: Two hundred thirty-two specimens, all from Mexico. Santa Cruz Texmalaquilla, Puebla: MVZ 201387–92; USNM 224193–201, 224203–06, 224209–10, 224213–17, 224221–29, 224231–35, 224237–39, 224242–46, 224248–49, 224251, 224254–56, 224258 and 224260–74. Xometla, Veracruz: CAS 98934–36, 98939, 98953, 98957; KU 106641–65; IBH 30987–88; LACM 117161–230; MVZ 114378–82, 138759, 138761–63, 143910–17, 163583–97, 163601–06, 163612, 184830, 195827–30, 198914–17, 198919, 198921, 231345–47, 233032–34; and USNM 492145–47.

**Diagnosis:** This medium-sized species of plethodontid salamander is phylogenetically close to *Chiropterotriton perotensis, C. totonacus* and *C. lavae*; mean SVL 33.9 mm in ten adult males (range 30.6–36.2) and 34.9 mm in ten adult females (range 33.3–38.4). The head is moderately wide; HW averages 15% of SVL in both males and females (range 14–16%). In males, the snout is broad and truncated. Jaw muscles are pronounced and visible as a bulging mass immediately behind the eyes. Eyes are moderately protuberant and extend laterally beyond the jaw margin in ventral view. There are few maxillary teeth in males (mean MT 11.0, range 7–18) but they are more numerous in females (mean MT 47.7, range 36–56). There are few vomerine teeth in males (mean VT 13.0, range 11–17) and females (mean VT 15.9, range 13–22), and they are arranged in a curved line that does not extend past the outer margin of the internal choanae. The tail is moderately long; mean TL equals 1.0 of SVL in males (range 0.89–1.12) and 0.97 of SVL in females (range 0.85–1.07). Limbs are moderately long; FLL + HLL averages 54% of SVL in males (range 48–57%) and 50% in females (range 45–54%). Adpressed limbs approach closely or overlap slightly in males (mean LI 0.0, range −0.5 to 1) but they are separated by as many as two costal folds in females (mean LI 1.5, range 1–2). Digits are slender and expanded distally, with distinct subterminal pads and moderate webbing at the base. All digits are discrete, including the first, which extends beyond the margins of the webbing. The outermost toes are particularly well developed. The smallest male with a mental gland is 30.6 mm SVL. The mental gland is prominent and oval (nearly round) to round. Parotoid glands are not evident.

**Comparisons:** *Chiropterotriton ceronorum* differs from *C. perotensis* by its larger adult body size (mean SVL 33.9 mm in male and 34.9 mm in female *C. ceronorum* vs. 29.7 mm in male and 31.7 mm in female *C. perotensis*), longer limbs (mean LI 0.0 in male and 1.5 in female *C. ceronorum* vs. 2.5 in male and 3.3 in female *C. perotensis*), longer head (mean HL 7.5 mm in male and 7.1 mm in female *C. ceronorum* vs. 6.6 mm in male

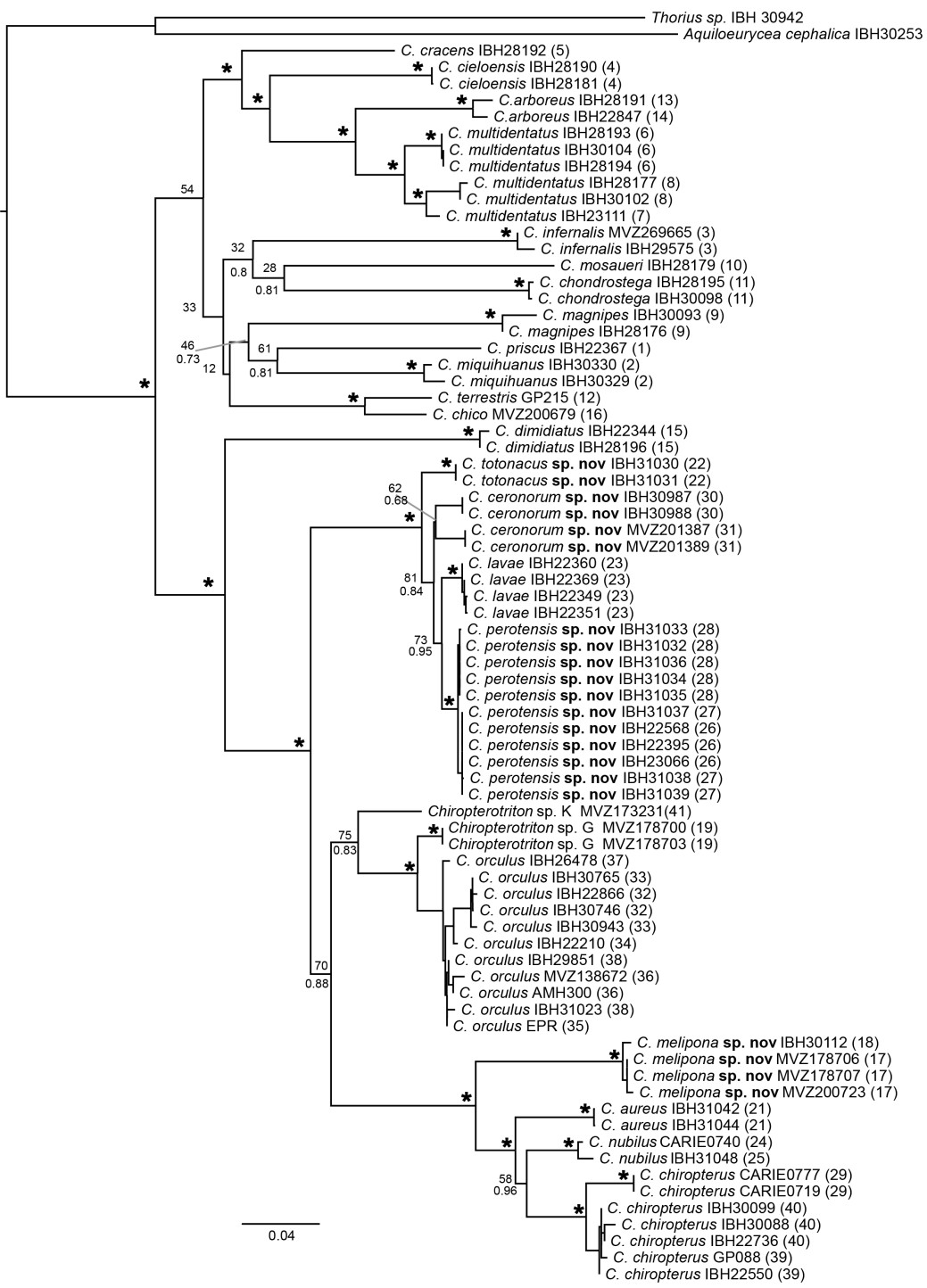

**Figure 3 Maximum likelihood (ML) phylogeny of the genus *Chiropterotriton* based on two mitochondrial markers.** Both ML and Bayesian measures of nodal support are indicated by bootstrap proportions (BS; above) and posterior probabilities (PP; below), respectively. Asterisks indicate statistically significant support in both analyses (PP > 0.95, BS > 70). Numbers in parentheses refer to localities from Fig. 1.

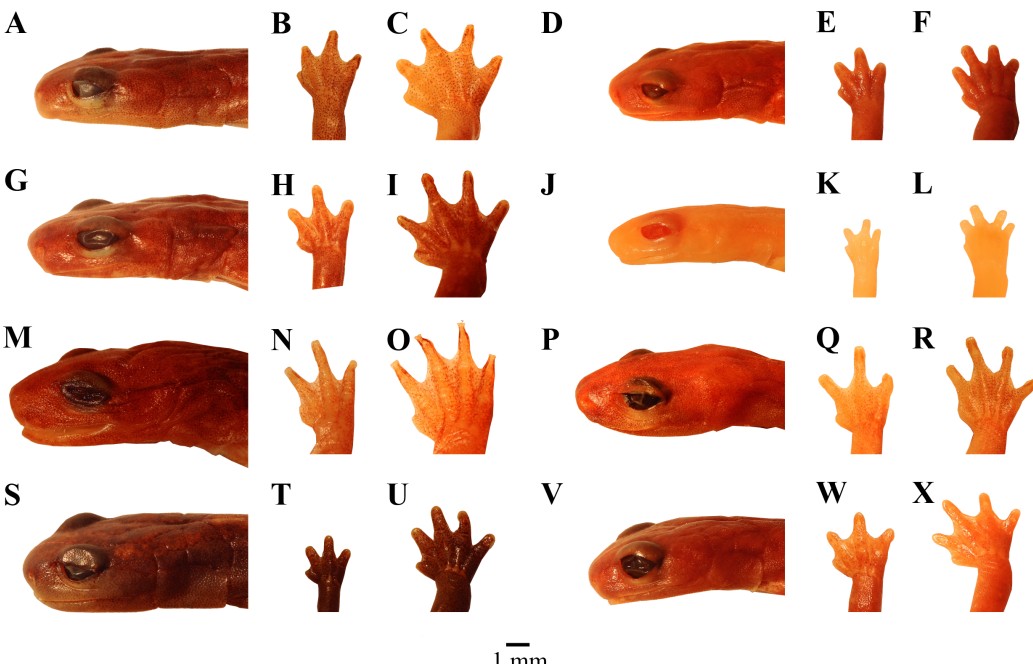

**Figure 4 Photographs of heads, hands and feet of preserved specimens of eight species of** *Chiropterotriton.* (A–C) *C. ceronorum,* holotype, USNM 224212; (D–F) *C. perotensis,* paratype, MVZ 186711; (G–I) *C. totonacus,* holotype, MVZ 163945; (J–L) *C. melipona,* paratype, MVZ 178706; (M–O) *C. casasi,* holotype, MVZ 92874; (P–R) *C. chiropterus,* neotype, MVZ 85590; (S–U) *C. orculus,* MVZ 138776; (V–X) *C. lavae,* MVZ 106436. Right hands and feet are seen in dorsal view.

and 6.7 mm in female *C. perotensis*), broader head (mean HW 5.1 mm in both male and female *C. ceronorum* vs. 4.2 mm in male and 4.4 mm in female *C. perotensis*), broader feet (mean FW 3.8 mm in male and 3.5 mm in female *C. ceronorum* vs. 2.6 mm in both male and female *C. perotensis*), more maxillary teeth (mean MT 11.0 in male and 47.7 in female *C. ceronorum* vs. 7.2 in male and 27.9 in female *C. perotensis*) and more vomerine teeth (mean VT 13.0 in male and 15.9 in female *C. ceronorum* vs. 9.0 in male and 11.1 in female *C. perotensis*).

*Chiropterotriton ceronorum* differs from *C. totonacus* in its slightly smaller adult body size (mean SVL 33.9 mm in male and 34.9 mm in female *C. ceronorum* vs. 35.7 mm in male and 35.5 mm in female *C. totonacus*), shorter tail (mean TL/SVL 1.0 in male and 0.97 in female *C. ceronorum* vs. 1.16 in male and 1.20 in female *C. totonacus*), shorter limbs (mean LI 0.0 in male and 1.5 in female *C. ceronorum* vs. −0.6 in male and 0.0 in female *C. totonacus*) and fewer maxillary teeth (mean MT 11.0 in male and 47.7 in female *C. ceronorum* vs. 32.9 in male and 52.6 in female *C. totonacus*).

*Chiropterotriton ceronorum* differs from *C. melipona* by its larger adult body size (mean SVL 33.9 mm in male and 34.9 mm in female *C. ceronorum* vs. 29.2 mm in male and 28.5 mm in female *C. melipona*), longer limbs in males (mean LI 0.0 in *C. ceronorum* vs. 2.3 in *C. melipona*), longer head (mean HL 7.5 mm in male and 7.1 mm in female *C. ceronorum* vs. 6.3 mm in male and 6.4 mm in female *C. melipona*), broader head

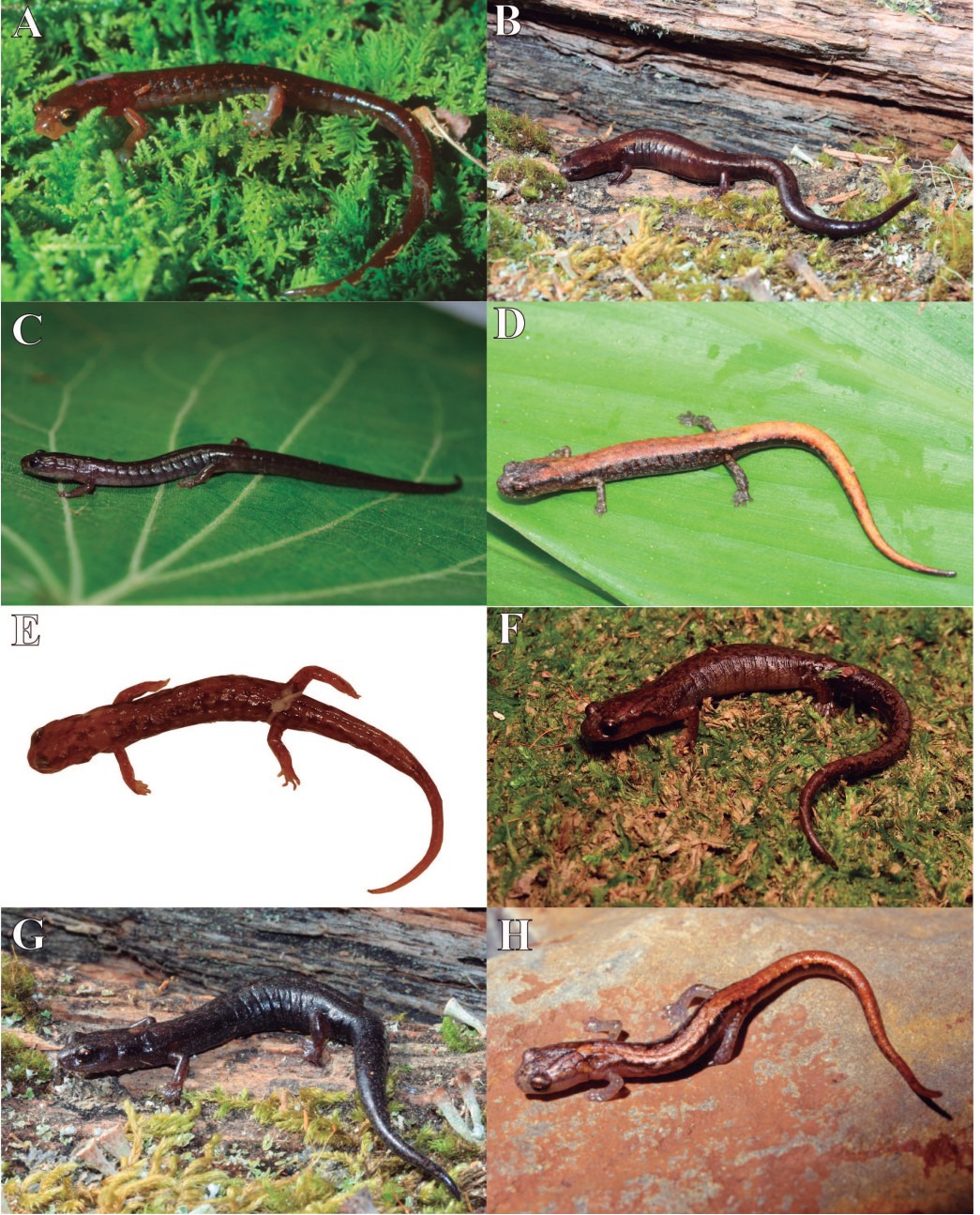

**Figure 5 Photographs of live and preserved specimens of eight species of *Chiropterotriton*.** (A) *C. ceronorum* sp. nov., IBH 30988; (B) *C. perotensis* sp. nov., IBH 30745; (C) *C. totonacus* sp. nov., IBH 31031; (D) *C. melipona* sp. nov., IBH 30112; (E) *C. casasi* sp. nov., paratype, MVZ 92876; (F) *C. chiropterus*, CARIE 0719; (G) *C. orculus*, IBH 30997; (H) *C. lavae*, IBH 22365.

(mean HW 5.1 mm in both male and female *C. ceronorum* vs. 4.3 mm in male and 4.2 mm in female *C. melipona*), broader feet (mean FW 3.8 mm in male and 3.5 mm in female *C. ceronorum* vs. 2.4 mm in male and 2.6 mm in female *C. melipona*), more maxillary teeth (mean MT 11.0 in male and 47.7 in female *C. ceronorum* vs. 9.5 in male and 31.0 in

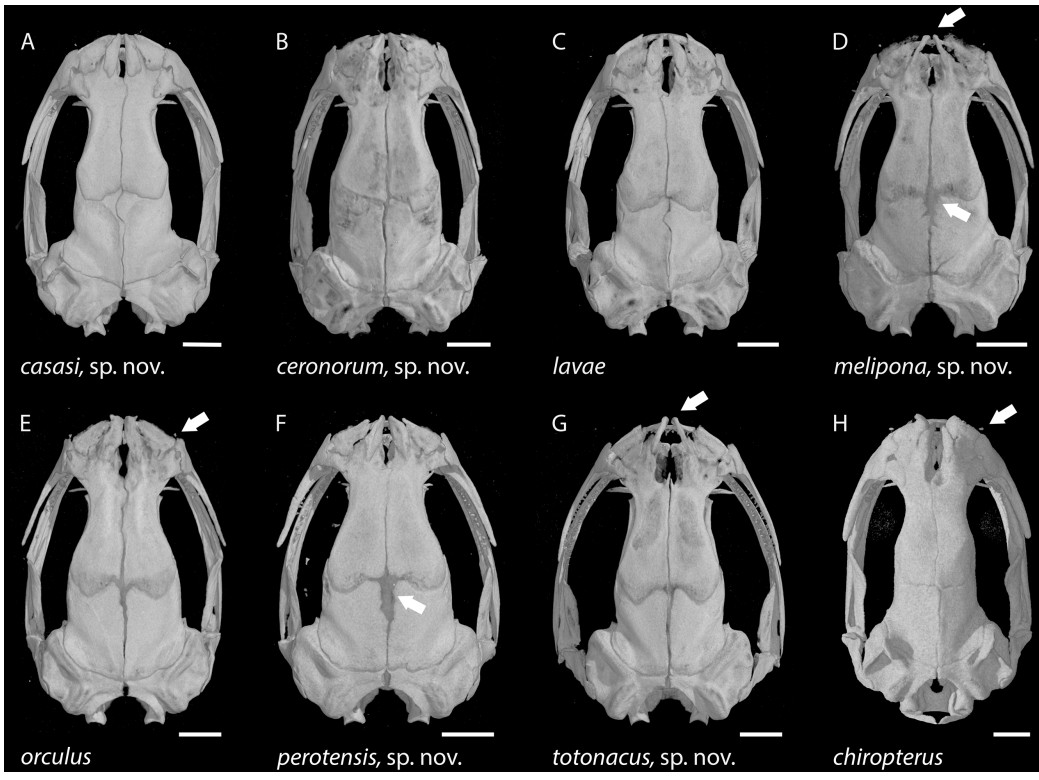

**Figure 6** **Skulls of eight *Chiropterotriton* species seen in dorsal view.** (A) *C. casasi* sp. nov.—holotype, MVZ 92874, an adult male; (B) *C. ceronorum* sp. nov.—holotype, USNM 224212, an adult male; (C) *C. lavae*—neotype, MVZ 163912, an adult male; (D) *C. melipona* sp. nov.—paratype, MVZ 178706, an adult male; (E) *C. orculus*—neotype, MVZ 138783, an adult male; (F) *C. perotensis* sp. nov. —paratype, MVZ 200693, an adult male; (G) *C. totonacus* sp. nov.—holotype, MVZ 163945, an adult female; (H) *C. chiropterus*—MVZ 85602, an adult male. Arrows point to the prominent frontoparietal fontanel in the cranial roof in (D) and (F), to the unusually narrow ascending processes of the premaxillary bone at the rostral end of the skull in (D) and (G), and to the tiny septomaxillary bones adjacent to the external nares in (E) and (H). All skulls are depicted at the same length; scale bar, 1 mm. Anterior is at the top. Images are derived from μCT scans.             

female *C. melipona*) and more vomerine teeth (mean VT 13.0 in male and 15.9 in female *C. ceronorum* vs. 11.0 in male and 13.0 in female *C. melipona*).

*Chiropterotriton ceronorum* differs from *C. casasi* in its smaller adult body size (mean SVL 33.9 mm in male and 34.9 mm in female *C. ceronorum* vs. 37.8 mm in male and 40.9 mm in one female *C. casasi*), shorter head (mean HL 7.5 mm in male and 7.1 mm in female *C. ceronorum* vs. 8.3 mm in male and 8.6 mm in one female *C. casasi*), narrower head (mean HW 5.1 mm in both male and female *C. ceronorum* vs. 5.8 mm in male and 5.9 mm in one female *C. casasi*), longer limbs in males (mean LI 0.0 in *C. ceronorum* vs. 0.8 in *C. casasi*), more maxillary teeth (mean MT 11.0 in male and 47.7 in female *C. ceronorum* vs. mean 9.0 in males and 30 in one female *C. casasi*) and more vomerine teeth (mean VT 13.0 in male and 15.9 in female *C. ceronorum* vs. mean 9.0 in males and 13 in one female *C. casasi*).

*Chiropterotriton ceronorum* differs from *C. chiropterus* in its smaller adult body size in males (mean SVL 33.9 mm in *C. ceronorum* vs. 37.5 mm in *C. chiropterus*), shorter

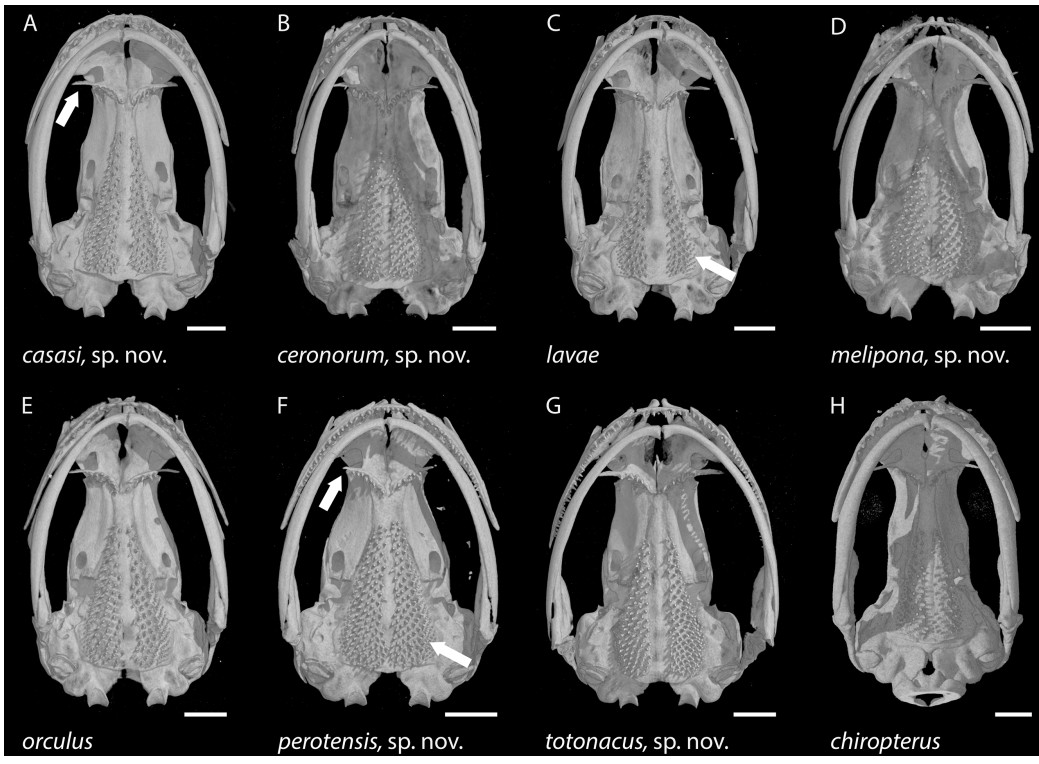

**Figure 7** **Skulls of eight *Chiropterotriton* species seen in ventral view.** (A) *C. casasi* sp. nov.—holotype, MVZ 92874, an adult male; (B) *C. ceronorum* sp. nov.—holotype, USNM 224212, an adult male; (C) *C. lavae*—neotype, MVZ 163912, an adult male; (D) *C. melipona* sp. nov.—paratype, MVZ 178706, an adult male; (E) *C. orculus*—neotype, MVZ 138783, an adult male; (F) *C. perotensis* sp. nov.—paratype, MVZ 200693, an adult male; (G) *C. totonacus* sp. nov.—holotype, MVZ 163945, an adult female; (H) *C. chiropterus*—MVZ 85602, an adult male. Arrows point to the long vs. short preorbital process of the vomer in (A) and (F), respectively; and to the unusually small parasphenoid tooth patch in (C) vs. the much larger patch in (F). All skulls are depicted at the same length; scale bar, 1 mm. Anterior is at the top. Images are derived from µCT scans.               

tail (mean TL/SVL 1.0 in male and 0.97 in female *C. ceronorum* vs. 1.25 in male and 1.19 in female *C. chiropterus*), longer limbs (mean LI 0.0 in male and 1.5 in female *C. ceronorum* vs. 0.3 in male and 2.0 in female *C. chiropterus*) and fewer maxillary teeth (mean MT 11.0 in male and 47.7 in female *C. ceronorum* vs. 12.6 in male and 48.0 in female *C. chiropterus*).

*Chiropterotriton ceronorum* differs from *C. orculus* in its smaller adult body size (mean SVL 33.9 mm in male and 34.9 in female *C. ceronorum* vs. 35.9 mm in male and 39.0 in female *C. orculus*), longer limbs (mean LI 0.0 in male and 1.5 in female *C. ceronorum* vs. 1.9 in male and 2.9 in female *C. orculus*), more maxillary teeth (mean MT 11.0 in male and 47.7 in female *C. ceronorum* vs. 8.2 in male and 28.8 mm in female *C. orculus*) and more vomerine teeth (mean VT 13.0 in male and 15.9 in female *C. ceronorum* vs. 8.6 in male and 12.0 in female *C. orculus*).

*Chiropterotriton ceronorum* differs from *C. lavae* in being slightly larger (mean SVL 33.9 mm in male and 34.9 mm in female *C. ceronorum* vs. 32.4 mm in male and 31.6 mm in female *C. lavae*), a shorter tail (mean TL/SVL 1.0 in male and 0.97 in female

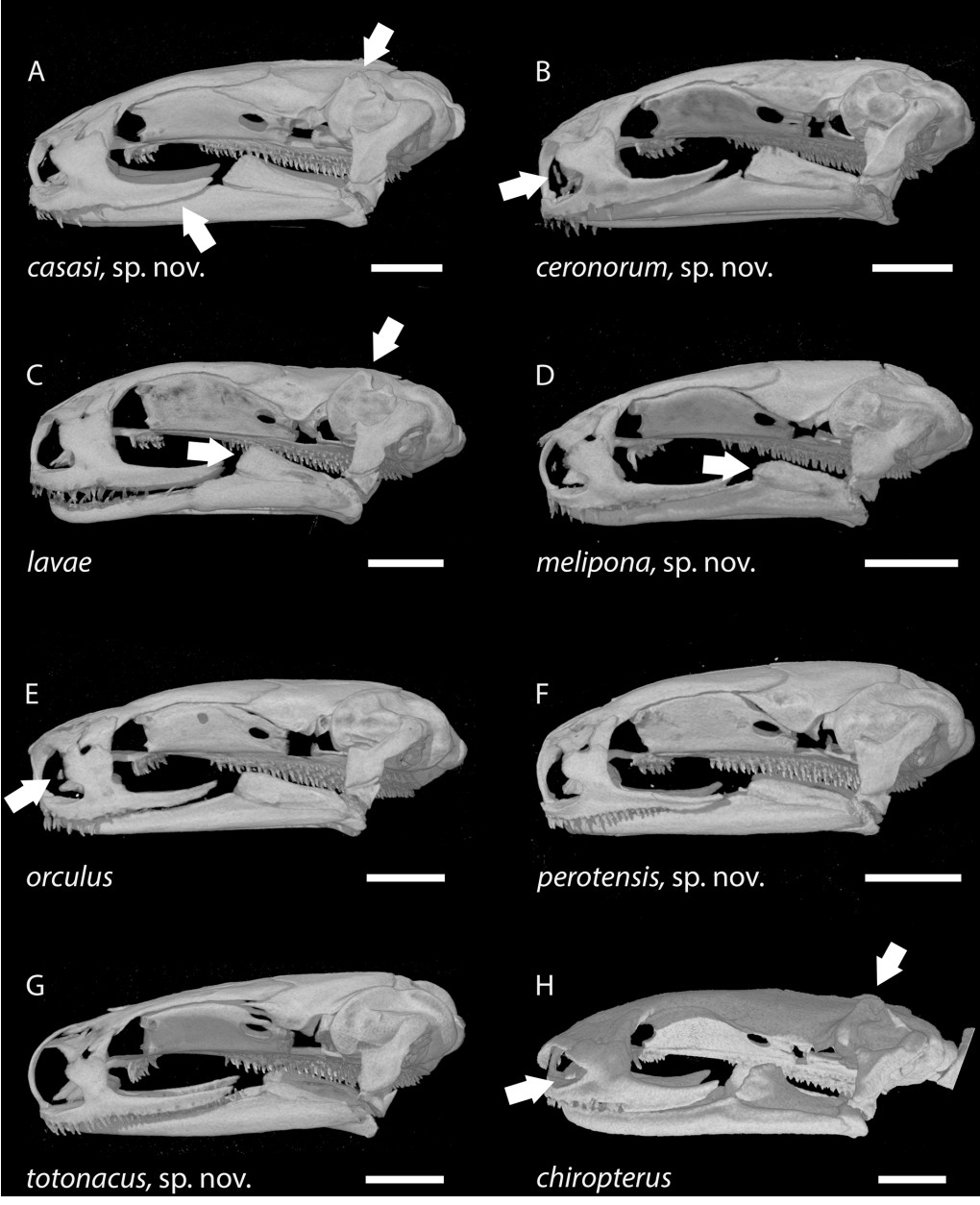

**Figure 8** **Skulls of eight *Chiropterotriton* species seen in lateral view.** (A): *C. casasi* sp. nov.—holotype, MVZ 92874, an adult male; (B): *C. ceronorum* sp. nov.—holotype, USNM 224212, an adult male; (C) *C. lavae*—neotype, MVZ 163912, an adult male; (D) *C. melipona* sp. nov.—paratype, MVZ 178706, an adult male; (E) *C. orculus*—neotype, MVZ 138783, an adult male; (F) *C. perotensis* sp. nov.—paratype, MVZ 200693, an adult male; (G) *C. totonacus* sp. nov.—holotype, MVZ 163945, an adult female; (H) *C. chiropterus*—MVZ 85602, an adult male. Arrows point to prominent dorsal crests on the otic capsule in (A), (C) and (H); to the high vs. low coronoid process on the prearticular bone of the lower jaw in (C) and (D), respectively; to the tiny septomaxillary bones in (B), (E) and (H); and to the posterior portion of the maxillary bone, which typically is dorsoventrally expanded and edentulous in males (A) vs. narrow and toothed in females (G). All skulls are depicted at the same length; scale bar, 1 mm. Anterior is to the left. Images are derived from μCT scans.

*C. ceronorum* vs. 1.19 in male and 1.02 in female *C. lavae*), shorter limbs (mean LI 0.0 in male and 1.5 in female *C. ceronorum* vs. −0.6 in male and 0.6 in female *C. lavae*), more maxillary teeth (mean MT 11.0 in male and 47.7 in female *C. ceronorum* vs. 7.0 in male and 20.8 in female *C. lavae*), and more vomerine teeth (mean VT 13.0 in male and 15.9 in female *C. ceronorum* vs. 8.9 in male and 11.4 in female *C. lavae*).

Chiropterotriton ceronorum differs from *C. aureus* by its larger adult body size (mean SVL 33.9 mm in male and 34.9 mm in female *C. ceronorum* vs. 28.5 mm in one male and 26.8 mm in female *C. aureus*), a shorter tail (mean TL/SVL 1.0 in male and 0.97 in female *C. ceronorum* vs. 1.28 in one male and 1.16 in female *C. aureus*), longer limbs (mean LI 0.0 in male and 1.5 in female *C. ceronorum* vs. 2.0 in one male and 2.3 in female *C. aureus*), longer head (mean HL 7.5 mm in male and 7.1 mm in female *C. ceronorum* vs. 6.4 mm in one male and 6.0 mm in female *C. aureus*), broader head (mean HW 5.1 mm in both male and female *C. ceronorum* vs. 4.0 mm in one male and 3.6 mm in female *C. aureus*), broader feet (mean FW 3.8 mm in male and 3.5 mm in female *C. ceronorum* vs. 2.4 mm in one male and 1.8 mm in female *C. aureus*), and more maxillary teeth (mean MT 11.0 in male and 47.7 in female *C. ceronorum* vs. 10.0 in one male and 38.3 in female *C. aureus*).

Chiropterotriton ceronorum differs from *C. nubilus* by its larger adult body size size (mean SVL 33.9 mm in male and 34.9 mm in female *C. ceronorum* vs. 29.4 mm in one male and 30.5 mm in female *C. nubilus*), a shorter tail (mean TL/SVL 1.0 in male and 0.97 in female *C. ceronorum* vs. 1.37 in one male and 1.12 in female *C. nubilus*), longer limbs in males (mean LI 0.0 in male *C. ceronorum* vs. 2.0 in one male *C. nubilus*), longer head in males (mean HL 7.5 mm in male *C. ceronorum* vs. 6.6 mm in one male *C. nubilus*), broader head (mean HW 5.1 mm in both male and female *C. ceronorum* vs. 4.0 mm in one male and 4.4 mm in female *C. nubilus*), and broader feet (mean FW 3.8 mm in male and 3.5 mm in female *C. ceronorum* vs. 2.6 mm in male and 2.3 mm in female *C. nubilus*).

**Description of holotype.** SVL 36.2 mm, TL 34.3 mm, AX 17.9 mm, SW 3.4 mm, HL 8.1 mm, HW 5.3 mm, HD 2.6 mm, projection of snout beyond mandible 0.8 mm, distance from anterior rim of orbit to snout 2.0 mm, interorbital distance 2.6 mm, eyelid length 1.8 mm, eyelid width 1.3 mm, horizontal orbit diameter 1.6 mm, nostril diameter 0.3 mm, FLL 10.0 mm, HLL 10.3 mm, snout-to-forelimb length 11.5 mm, snout to anterior angle of vent 35.2 mm, tail width at base 2.4 mm, tail depth at base 2.6 mm, FW 4.6 mm, length of fifth toe 0.7 mm, length of third (longest) toe 1.3 mm, mental gland length 2.0 mm, mental gland width 1.7. Numbers of teeth: premaxillary 3, maxillary 5-4 (right-left) and vomerine 5-6 (right-left). Adpressed limbs are separated by two costal folds.

**Variation:** Specimens of *C. ceronorum* from Xometla are smaller and have a longer tail than those from the type locality: mean SVL 33.9 mm in males and 34.9 mm in females from Texmalaquilla vs. 31.0 mm in males and 32.0 mm in females from Xometla; and mean TL/SVL 1.0 in males and 0.97 in females from Texmalaquilla vs. 1.17 in males and 1.08 in females from Xometla.

**Coloration in life:** These notes are based on study of a series of diapositives taken by Gabriela Parra-Olea from near Xometla and by Roy W. McDiarmid from the vicinity of Santa Cruz Texmalaquilla. Colors are from *Köhler (2012)*.

The single Xometla specimen is generally dark brown and lacks a dorsal stripe or band. Dorsal and lateral coloration reddish brown (Mahogany Red, 34) anteriorly becoming brown (Brussels Brown, 33) medially and posteriorly. Lateral and ventral surfaces grayish (Smoke Gray, 266). Face and cheeks as well as limbs bright gray-brown (Smoke Gray, 267, to Light Drab, 269). Snout Ground Cinnamon (270) to True Cinnamon (28) to Vinaceous (247) at its tip. Upper eyelid Cream Yellow (82) at rim. Iris Cream Yellow (82) to bright Trogon Yellow (81) dorsally but much darker and brownish ventrally. Manus and pes bright light gray (Pale Neutral Gray, 296) but essentially colorless at the digit tips, which are transparent and show underlying reddish blood vessels.

The Texmalaquilla specimens (nine) all have dark to very dark basic ground color dorsally and laterally (venter not visible). Usually a dorsal band or stripe is present that extends from the posterior surface of the head (over the anterior extension of the epaxial muscles) to the tail tip. The band is almost uninterrupted in some specimens but is discontinuous or contains numerous spots or flecks of darker color in others. The stripe can be very bright and can be rich reddish (Pratt's Rufous, 72), orange-brown (Flesh Ocher, 57, to Orange Rufous, 56) to Salmon Color (58) and Dark Salmon Color (59). In others it is Clay Color (18, 20).

**Coloration in preservative:** The holotype is a uniform dark tannish brown dorsally, becoming paler laterally and very pale cream color ventrally. The dark tannish brown extends to the tip of the tail. Limbs are yellowish. Mental gland is beige. Nine paratypes are uniform dorsally, ranging from golden tan to very dark gray; in some, the tail is slightly paler than the dorsum. These nine paratypes have lateral surfaces paler than dorsal, and ventral surfaces are much lighter than lateral surfaces. The remaining eleven paratypes have a stripe of some sort. The stripe is always paler than immediately adjacent lateral parts, but it can be very obscure and seen mainly in the tail or it can extend all the way from the nape to the tip of the tail. The stripe is bright yellow in some individuals but typically is darker; in some specimens there is a suffusion of black in the middle of the stripe. All individuals are paler ventrally, but in some very dark animals the venter is dark gray and only the gular area is pale. The mental gland is usually pale.

**Osteology:** This account is based on examination of a µCT scan of the anterior skeleton of USNM 224212, an adult male, 36.2 mm SVL (Figs. 6 and 7; Table 3). The skull is robust in its degree of ossification, although many roofing bones are extremely thin. Paired frontals and parietals are for the most part well-articulated with one another; there is only a narrow but elongate frontoparietal fontanel, mostly along the midline. Anteriorly, the frontals articulate with the nasal and prefrontal bones, as well as with the ascending processes of the single premaxilla. The ascending processes never contact one another but gradually widen as they establish an articulation with the frontals, thereby enclosing the internasal fontanel. The palatal shelf of the premaxilla is very narrow and barely

evident. Paired septomaxillary bones are present but small. The nasal bone is triangular but very thin, and somewhat larger than the prefrontal, which is more rectangular in shape. Both bones are overlapped by the facial process of the maxilla, but where the three bones meet the foramen for the nasolacrimal duct has eroded the facial process and the prefrontal but not the adjacent nasal. The anterior, toothed portion of the maxilla comprises only around 40% of the length of the bone; the remaining 60% is edentulous and saber-shaped. In dorsal view, the posterior tip of each maxilla doesn't bow out laterally as they do in some congeners (e.g., *C. orculus*). There are five maxillary teeth on the right side and seven on the left. There is but a single, short premaxillary tooth. The orbitosphenoid, while relatively large, is only weakly articulated to the parasphenoid and frontal and mostly separated from the parietal.

The otic capsule bears a distinct crest that extends anteriorly from the midpoint of the lateral semicircular canal to about the anterior third of the anterior semicircular canal. A narrow, spine-like tab is reflected ventromedially from the posterolateral margin of the parietal, ending at about the middle of the vertical extent of the orbitosphenoid. The squamosal is robust and expanded anteroventrally. The quadrate is stout. A stubby, thick-based stylus is present on the operculum. Paired vomers are well developed but barely articulate at the midline posterior to the internasal fontanel. The preorbital process of each vomer is elongate, twisted and somewhat expanded laterally. Each side bears six vomerine teeth, which are deployed medially and do not extend onto the preorbital process. The median parasphenoid bone is triangular, but its caudal end is slightly bowed posteriorly. Paired parasphenoid tooth patches are separate at the midline; each bears approximately 60 teeth. The mandible is relatively stout. The articular bone is well ossified. The prearticular bone is well developed and bluntly rounded anteriorly, with a high coronoid process. There are 15 or 16 teeth on each dentary bone.

Digital formulae are 1-2-3-2 on each side. The terminal phalanx is barely expanded on each finger. Mesopodial cartilages are not mineralized.

**Distribution and ecology:** *Chiropterotriton ceronorum* occurs on the southern slopes of Pico de Orizaba in the states of Puebla and Veracruz at elevations that range from 2,600 to approximately 3,100 masl. Specimens have been found in arboreal bromeliads as well as under terrestrial cover objects.

**Remarks:** *Chiropterotriton ceronorum* is found in sympatry with *Pseudoeurycea gadovii*, *P. leprosa*, *Thorius spilogaster* and *T. lunaris*. Much of the natural habitat has been destroyed in recent years, making the species difficult to find. This species occurs at higher elevations than the nearby (to the NE) *Chiropterotriton chiropterus*.

**Conservation status:** *Chiropterotriton ceronorum* was very common during the 1970s, but is now very difficult to find, probably because of extensive habitat modification. On two visits to the area in 2015, no individuals of this species were seen while all the species with which it is known to co-occur were found. The remaining forest in the area where it lives is severely fragmented with ongoing degradation. We recommend that it be designated as Critically Endangered (CR) based on criterion B1ab(iii) (extent of

occurrence <100 km², severely fragmented range and continuing decline in area, extent, and quality of habitat).

**Etymology:** The species name honors members of the Ceron family of Cuautlalpan, Veracruz, who have assisted generations of herpetologists in collecting salamanders in the general region of Pico de Orizaba.

### *Chiropterotriton perotensis*, sp. nov.
Valle Alegre Salamander, Salamandra de Valle Alegre
Figures 4D–4F, 5B, 6F, 7F, 8F.

Chresonymy
*Chiropterotriton chiropterus* (part).—*Smith & Taylor, 1948*; *Wake, Papenfuss & Lynch, 1992*.
*Chiropterotriton* sp. H.—*Darda, 1994*; *Parra-Olea, 2003*; *Raffaëlli, 2007*; *Raffaëlli, 2013*; *Rovito & Parra-Olea, 2015*; *García-Castillo et al., 2017*; *García-Castillo et al., 2018*.
*Chiropterotriton* sp.—*Rovito et al., 2015a*.

**Holotype:** MVZ 200693, an adult female from 14.4 km S (by road surfaced with rocks) Las Vigas de Ramírez at Microwave Station, Valle Alegre, Veracruz, Mexico, 3,020 masl, 19.56917°N, 97.09528°W (EPE = max. error distance 1.142 km). Collected 26 August 1982 by D.M. Darda and S. Sessions.

**Paratypes:** Nineteen specimens, all from Veracruz, Mexico. Twelve males: MVZ 114356 and 114359, road from Las Vigas de Ramírez to microwave station on N Flank Cofre de Perote, 11.6 km S (by road) Las Vigas; MVZ 173428–29, Las Vigas de Ramírez, microondas road; MVZ 178661 and 178663–65, 8–15.5 km S (via microondas road) Las Vigas de Ramírez; MVZ 200681–83 and 200698, 14.4 km S (by Rock Rd.) Las Vigas de Ramírez at microwave station. Seven females: MVZ 173438–39, Las Vigas de Ramírez, microondas road; MVZ 186711, road to microwave station, 15 km S (by road) Las Vigas de Ramírez; MVZ 200691, 200694–95 and 200702, 14.4 km S (by Rock Rd.) Las Vigas de Ramírez at microwave station.

**Referred specimens:** Seventy-two specimens, all from Veracruz, Mexico. IBH 16778–82, 22384, 22391, 22395, 22568, 23062, 23066, 23072, 29853, 29857, 29863–64, 29866, 29872, 30840–41, 30844, 30847, 31032–39 and 31055–62; KU 100747–54; MVZ 114351, 114355, 114357, 114358, 173440–41, 178659–60, 178662, 178666–68, 200684–86, 200688–90, 200692, 200695–97, 200699–701 and 200703.

**Diagnosis:** This is a small but stout species of plethodontid salamander that is phylogenetically related to *Chiropterotriton lavae*, *C. ceronorum* and *C. totonacus*; mean SVL 29.7 mm in 12 adult males (range 26.5–32.8) and 31.7 mm in eight adult females (range 27.4–34.3). The head is moderately wide; HW averages 14% of SVL in both males and females (range 13–15%). The snout is short. Eyes are small and typically do not protrude laterally beyond the jaw margin in ventral view; they are less prominent than in most other species of *Chiropterotriton*. Jaw muscles caudal to the eyes are variably

developed but generally pronounced. There are few maxillary teeth in males (mean MT 7.2, range 2–17) and moderate numbers in females (mean MT 27.9, range 19–36). There are few vomerine teeth in both males (mean VT 9.0, range 7–12) and females (mean VT 11.1, range 10–13), which are arranged in a curved line that does not extend lateral to the outer margin of the internal choana. The tail is moderately sized; mean TL equals 1.03 of SVL in males (range 0.92–1.16) and 1.0 of SVL in females (range 0.79–1.11). Limbs are short; FLL + HLL averages 47% of SVL in males (range 44–50%) and 43% of SVL in females (range 41–46%). Adpressed limbs are widely separated—they never overlap—in both males (mean LI 2.5, range 1–3) and females (mean LI 3.3, range 2–4). Manus and pes are relatively small for the genus. Digital webbing ranges from absent to slight; when present, it is limited to the metatarsal region. The first digit is small and usually included within the webbing, although a small portion of it may be free at the tip. The outermost digit is less prominent than in other species; digit 5 (pes) is distinctly shorter than digits 2–4. Subterminal pads are present but not prominent. An oval-shaped mental gland is present in males but is not particularly prominent. The smallest male with a mental gland is 29.3 mm SVL. Paratoid glands are present in many individuals and prominent in some.

**Comparisons:** *Chiropterotriton perotensis* differs from *C. ceronorum* in its smaller adult body size (mean SVL 29.7 mm in male and 31.7 mm in female *C. perotensis* vs. 33.9 mm in male and 34.9 mm in female *C. ceronorum*), shorter limbs (mean LI 2.5 in male and 3.3 in female *C. perotensis* vs. 0.0 in male and 1.5 in female *C. ceronorum*), shorter head (mean HL 6.6 mm in male and 6.7 mm in female *C. perotensis* vs. 7.5 mm in male and 7.1 mm in female C. *ceronorum*), narrower head (mean HW 4.2 mm in male and 4.4 mm in female *C. perotensis* vs. 5.1 mm in both male and female *C. ceronorum*), narrower feet (mean FW 2.6 mm in both male and female *C. perotensis* vs. 3.8 mm in male and 3.5 mm in female *C. ceronorum*), fewer maxillary teeth (mean MT 7.2 in male and 27.8 in female *C. perotensis* vs. 11.0 in male and 47.7 in female *C. ceronorum*) and fewer vomerine teeth (VT 9.0 in male and 11.1 in female *C. perotensis* vs. 13.0 in male and 15.9 in female *C. ceronorum*).

*Chiropterotriton perotensis* differs from *C. totonacus* in its smaller adult body size (mean SVL 29.7 mm in male and 31.7 mm in female *C. perotensis* vs. 35.7 mm in male and 35.5 mm in female *C. totonacus*), shorter tail (mean TL/SVL 1.0 in both male and female *C. perotensis* vs. 1.16 in male and 1.20 in female *C. totonacus*), shorter limbs (mean LI 2.5 in male and 3.3 in female *C. perotensis* vs. −0.6 in male and 0.0 in female *C. totonacus*), shorter head (mean HL 6.6 mm in male and 6.7 mm in female *C. perotensis* vs. 8.5 mm in male and 7.6 mm in female *C. totonacus*), narrower head (mean HW 4.2 mm in male and 4.4 mm in female *C. perotensis* vs. 5.2 mm in both male and female *C. totonacus*), narrower feet (mean FW 2.6 mm in both male and female *C. perotensis* vs. 4.2 mm in male and 4.0 mm in female *C. totonacus*), fewer maxillary teeth (mean MT 7.2 in male and 27.9 in female *C. perotensis* vs. 32.9 in male and 52.6 in female *C. totonacus*) and fewer vomerine teeth (mean VT 9.0 in male and 11.1 in female *C. perotensis* vs. 11.6 in male and 13.7 in female *C. totonacus*).

*Chiropterotriton perotensis*, while very similar in morphological proportions to *C. melipona*, differs by its shorter limbs in females (mean LI 3.3 in *C. perotensis* vs. 1.8 in *C. melipona*), fewer maxillary teeth (mean MT 7.2 in male and 27.9 in female *C. perotensis* vs. 9.5 in male and 31.0 in female *C. melipona*) and fewer vomerine teeth (mean VT 9.0 in male and 11.1 in female *C. perotensis* vs. 11.0 in male and 13.0 in female *C. melipona*).

*Chiropterotriton perotensis* differs from *C. casasi* in its smaller adult body size (mean SVL 29.7 mm in male and 31.7 mm in female *C. perotensis* vs. 37.8 mm in male and 40.9 mm in one female *C. casasi*), shorter limbs (mean LI 2.5 in male and 3.3 in female *C. perotensis* vs. 0.80 in male and 1.0 in one female *C. casasi*), shorter head (mean HL 6.6 mm in male and 6.7 mm in female *C. perotensis* vs. 8.3 mm in male and 8.6 mm in one female *C. casasi*), narrower head (mean HW 4.2 mm in male and 4.4 mm in female *C. perotensis* vs. 5.8 mm in male and 5.9 mm in one female *C. casasi*), narrower feet (mean FW 2.6 mm in both male and female *C. perotensis* vs. 3.7 mm in both male and one female *C. casasi*), fewer maxillary teeth (mean MT 7.2 in male and 27.9 in female *C. perotensis* vs. 9.0 in male and 30.0 in one female *C. casasi*) and fewer vomerine teeth in females (mean VT 11.1 in *C. perotensis* vs.13.0 in one *C. casasi*).

*Chiropterotriton perotensis* differs from *C. chiropterus* by its smaller adult body size (mean SVL 29.7 mm in male and 31.7 mm in female *C. perotensis* vs. 37.5 mm in male and 33.5 mm in female *C. chiropterus*), shorter tail (mean TL/SVL 1.0 in both male and female *C. perotensis* vs. 1.25 in male and 1.19 in female *C. chiropterus*), shorter limbs (mean LI 2.5 in male and 3.3 in female *C. perotensis* vs. 0.30 in male and 2.0 in female *C. chiropterus*), shorter head (mean HL 6.6 mm in male and 6.7 mm in female *C. perotensis* vs. 8.1 mm in male and 7.3 mm in female *C. chiropterus*), narrower head (mean HW 4.2 mm in male and 4.4 mm in female *C. perotensis* vs. 5.6 mm in male and 4.8 mm in female *C. chiropterus*), narrower feet (mean FW 2.6 mm in both male and female *C. perotensis* vs. 3.7 mm in male and 3.1 mm in female *C. chiropterus*), fewer maxillary teeth (mean MT 7.2 in male and 27.9 in female *C. perotensis* vs. 12.6 in male and 48.0 in female *C. chiropterus*) and fewer vomerine teeth (mean VT 9.0 in male and 11.1 in female *C. perotensis* vs. 10.6 in male and 12.5 in female *C. chiropterus*).

*Chiropterotriton perotensis* differs from *C. orculus* in its smaller adult body size (mean SVL 29.7 mm in male and 31.7 mm in female *C. perotensis* vs. 35.9 mm in male and 39.0 mm in female *C. orculus*), slightly shorter limbs (mean LI 2.5 in male and 3.3 in female *C. perotensis* vs. 1.9 in male and 2.9 in female *C. orculus*), shorter head (mean HL 6.6 mm in male and 6.7 mm in female *C. perotensis* vs. 7.4 mm in male and 8.0 mm in female *C. orculus*), narrower head (mean HW 4.2 mm in male and 4.4 mm in female *C. perotensis* vs. 5.0 mm in male and 5.2 mm in female *C. orculus*), narrower feet (mean FW 2.6 mm in both male and female *C. perotensis* vs. 3.2 mm in male and 3.4 mm in female *C. orculus*) and fewer maxillary teeth (mean MT 7.2 in male and 27.9 in female *C. perotensis* vs. 8.2 in male and 28.8 in female *C. orculus*).

*Chiropterotriton perotensis* differs from *C. lavae* in having a smaller adult body size in males (mean SVL 29.7 mm in *C. perotensis* vs. 32.4 mm in *C. lavae*), shorter limbs (mean LI 2.5 in male and 3.3 in female *C. perotensis* vs. −0.6 in male and 0.6 in female *C. lavae*), a slightly narrower head (mean HW 4.2 mm in male and 4.4 mm in female

*C. perotensis* vs. 4.9 mm in male and 4.7 mm in female *C. lavae*), a shorter head (mean HL 6.6 mm in male and 6.7 mm in female *C. perotensis* vs. 7.5 mm in male and 7.0 mm in female *C. lavae*), narrower feet (FW 2.6 mm in both male and female *C. perotensis* vs. 3.7 mm in male and 3.3 mm in female *C. lavae*) and more maxillary teeth in females (mean MT 27.9 in *C. perotensis* vs. 20.8 in *C. lavae*).

*Chiropterotriton perotensis* differs from *C. aureus* in its smaller adult body size (mean SVL 29.7 mm in male and 31.7 mm in female *C. perotensis* vs. 28.5 mm in male and 26.8 mm in female *C. aureus*), shorter tail in males (mean TL/SVL 1.0 in both male and female *C. perotensis* vs. 1.28 in male and 1.16 in female *C. aureus*), broader head (mean HW 4.2 mm in male and 4.4 mm in female *C. perotensis* vs. 4.0 mm in male and 3.6 mm in female *C. aureus*), broader feet in females (mean FW 2.6 mm in female *C. perotensis* vs. 1.8 mm in female *C. aureus*), fewer maxillary teeth in females (mean MT 27.9 in female *C. perotensis* vs. 38.3 in female *C. aureus*) and fewer vomerine teeth in males (mean VT 9.0 in male *C. perotensis* vs. 15.0 in male *C. aureus*).

*Chiropterotriton perotensis* differs from *C. nubilus* in having a shorter tail (mean TL/SVL 1.0 in both male and female *C. perotensis* vs. 1.37 in male and 1.12 in female *C. nubilus*), shorter limbs (mean LI 2.5 in male and 3.3 in female *C. perotensis* vs. 2.0 in male and 1.5 in female *C. nubilus*), and fewer maxillary teeth (mean MT 7.2 in male and 27.9 in female *C. perotensis* vs. 13.0 in male and 41.5 in female *C. nubilus*).

**Description of holotype:** SVL 31.1 mm, TL 30.7 mm, AX 16.4 mm, SW 3.1 mm, HL 6.8 mm, HW 4.2 mm, HD 2.0 mm, projection of snout beyond mandible 0.4 mm, distance from anterior rim of orbit to snout 1.7 mm, interorbital distance 1.8 mm, eyelid length 2.2 mm, eyelid width 0.8 mm, horizontal orbit diameter 1.4 mm, FLL 6.5 mm, HLL 6.7 mm, snout-to-forelimb length 8.8 mm, snout to anterior angle of vent 29.5 mm, tail width at base 2.1 mm, tail depth at base 2.6 mm, FW 2.5 mm, length of fifth toe 0.5 mm, length of third (longest) toe 1.2 mm. Numbers of teeth: premaxillary 6, maxillary 15-16 (right-left) and vomerine 7-6 (right-left). Adpressed limbs are separated by 4 costal folds.

**Coloration in life:** Color notes in life are not available for specimens in the type series, but notes were recorded for the following referred specimens. IBH 29853, 29857, 29863, 29864, 29866 and 29872, 15 km S of Las Vigas on road to Valle Alegre: General coloration dark with a dark reddish brown dorsal stripe in some and obscure brown to grayish brown stripe in others. The reddish stripe is brightest laterally with darker pigment medially. Small guanophores are abundantly distributed over the mainly very dark pigment dorsally. The iris is golden brown to dark brown. The venter is dark to very dark. In one adult there is a complete melanophore network; in another, dense punctuations. Some white guanophores are prominent in the darker individual. IBH 22384, 22395, 23062, 23066 and 23072, 15.9 km on microondas road, Las Vigas: Adults are very dark dorsally—almost black—with a fine speckling of obscure white overlying the ground color. Fine background mottling of dark brown on black. Limbs are black with some paler highlights medially, but become brown distally. The iris is dark brownish black. The venter is dark, dense

mainly punctate melanophores, with a very fine superficial sprinkling of white ventrolaterally. The gular area is slightly paler. Juveniles have an indistinct brown stripe, which is less apparent in larger animals.

**Coloration in preservative:** The holotype is a uniform dark brown dorsally and laterally, becoming blackish brown on the tail. The venter is much paler than the dorsum, becoming dark brown under the tail. Limbs are dark brown. There is no other distinguishing color. Two of the paratypes have a hint of a dorsal stripe, which is slightly paler than surrounding areas. The manus and pes are paler, but in general are brown to blackish brown.

**Osteology:** This account is based on examination of a µCT scan of the anterior skeleton of MVZ 200693, an adult female, 31.1 mm SVL (Figs. 6–8; Table 3). The skull is compact. Individual cranial roofing bones are for the most part well developed, although there is a marked frontoparietal fontanel that begins at the frontal-parietal border and extends posteriorly along the midline. The frontal is fairly robust. Anteriorly, it is solidly articulated with the ascending processes of the single premaxilla, which arise separately and remain distinct along their entire length. The processes expand laterally where they articulate with the frontal bones. The premaxilla lacks a palatal shelf and there are no septomaxillary bones. The nasal bone is triangular but very thin. It is considerably larger than the rectangular prefrontal, which is distinct but small. A foramen for the nasolacrimal duct has eroded the anteroventral margin of the prefrontal, the posteroventral margin of the nasal, and the dorsal edge of the facial process of the maxilla. The anterior, toothed portion of the maxilla comprises approximately 75–80% of the length of the bone; the remaining edentulous portion is thinner and cleaver-like. The facial process of the maxilla extends rostrally. There are 16 maxillary teeth on the left side and 17 on the right. There are seven premaxillary teeth. The orbitosphenoid is moderately well developed and relatively large, but it is only weakly articulated to the parasphenoid and frontal and separated from the parietal.

The otic capsule bears a modest dorsal crest above the anterior semicircular canal but there is no distinct otic process. A well-developed tab extends ventromedially from the posterolateral surface of the parietal. It is relatively long and spine-like and extends through about two-thirds of the vertical extent of the orbitosphenoid. The squamosal bone is relatively stout, roughly triangular, and abuts the otic capsule along a broad front that subtends the lateral semicircular canal. The quadrate bone is relatively small and inconspicuous. The columella bears a distinct stylus. Bodies of the vomer are well ossified but also well separated at the midline. Each preorbital process is short, ending at the lateral edge of the internal naris. There are nine vomerine teeth on the right side and six on the left; a few are deployed on the preorbital process. The parasphenoid is fairly broad anteriorly; its posterior border is straighter (less rounded) than in some other species. Paired parasphenoid tooth patches meet at the midline both anteriorly and posteriorly, but not in between. There are approximately 105 fully developed teeth on each side and smaller, less-developed teeth along each lateral margin. The mandible is robust.

The articular is only partly ossified. The prearticular is relatively small and has a low coronoid process. Teeth are small and very numerous on each dentary bone, but a reliable count cannot be made from the CT scan.

Digital formulae are 1-2-3-2 on each side. The distal tip of the terminal phalanx is slightly expanded on each finger. Mesopodial cartilages are not mineralized.

**Distribution and ecology:** *Chiropterotriton perotensis* is found on Cofre de Perote, Veracruz, Mexico, both in pine-and-fir forest and from the tree line to the summit. Elevations range from 2,950 to 4,015 m. Specimens have been found under terrestrial objects and active on road banks and boulders at night. The species occurs in sympatry with *Aquiloeurycea cephalica*, *Isthmura naucampatepetl*, *Pseudoeurycea leprosa* and *P. melanomolga*.

**Remarks:** Allozymes of this species were studied by Darda (his unnamed species H) (1994), who also reported a sympatric species (his species D). These two were separated by four fixed differences (out of 17 proteins studied). *Parra-Olea (2003)* was unable to obtain mtDNA sequence from his remaining (ground and degraded) tissue samples and did not find additional specimens. We consider the dissected carcasses to be inadequate for preparation of a formal description, but we note the presence of a likely additional species of *Chiropterotriton* at the Las Lajas locality. Like *C. perotensis*, this unnamed species is small, but apparently more slender and lighter in coloration. The two are not sister-taxa.

We think that the specimens reported as *Chiropterotriton chiropterus* from 11,000 feet on Cofre de Perote by *Smith & Taylor (1948)* belong to *C. perotensis*.

**Conservation status:** We recommend that the species be designated as Endangered based on criterion B1ab(iii) (extent of occurrence <5,000 km$^2$, habitat severely fragmented with continuing decline in area, extent, and quality of habitat; *IUCN, 2012*).

**Etymology:** The species name is a noun in the genitive case. It refers to the Cofre de Perote volcano, where the species is found.

***Chiropterotriton totonacus*, sp. nov.**
Cruz Blanca Salamander, Salamandra de Cruz Blanca
Figures 4G–4I, 5C, 6G, 7G, 8G.

Chresonymy
*Chiropterotriton* sp. E.—*Darda, 1994*.
*Chiropterotriton chiropterus* (part).—*Taylor & Smith, 1945*; *Smith & Taylor, 1948*; *Wake, Papenfuss & Lynch, 1992*.

**Holotype:** MVZ 163945, an adult female from 6 km W Las Vigas de Ramírez, Veracruz, Mexico, 2,420 masl, 19.635° N, 97.159166° W (EPE = max. error distance 5.71 km). Collected 25 July 1979 by D.B. Wake.

**Paratypes:** Nineteen specimens, all from Veracruz, Mexico. Ten males: MVZ 163947–49, 163989–90, 163993, 171903, 171905, 171907 and 171909, 6 km W Las Vigas de Ramírez.

Nine females: MVZ 136981–82, 136986, pine forest along Mexican Hwy. 140, 4 km W Las Vigas de Ramírez; MVZ 138703–04, 138716 and 138765, Mexican Hwy. 140, 4.5 km W (by road) Las Vigas de Ramírez; MVZ 163943 and 171910, 6 km W Las Vigas de Ramírez.

**Referred specimens:** Fifty-three specimens, all from Veracruz, Mexico. IBH0122, 30998 and 31030–31031; MVZ 136983–85, 137029, 138702, 138705–15, 138717–19, 163942, 163944, 163946, 163991–92, 163994, 171904, 171906, 171908 and 171911–31.

**Diagnosis:** This medium-sized species of plethodontid salamander is phylogenetically close to *Chiropterotriton lavae*, *C. perotensis* and *C. ceronorum*; mean SVL 35.7 mm in ten adult males (range 32.0–38.6) and 35.5 mm in ten adult females (range 31.8–38.3). The head is moderately wide; HW averages 15% of SVL in both sexes (range 14–16). Jaw muscles are prominent in both sexes. Adult males have a broad, blunt snout with pronounced nasolabial protuberances that extend below the lip. Eyes are large and prominent and extend laterally beyond the jaw margin in ventral view. There are numerous maxillary teeth in males (mean MT 32.9, range 18–48) and even more teeth in females (mean MT 52.6, range 45–60). There are few vomerine teeth in both males (mean VT 11.6, range 10–15) and females (mean VT 13.7, range 9–17), which are arranged in a curved line that does not extend past the lateral margin of the internal choana. The tail is long and slender and typically exceeds SVL; mean TL equals 1.16 of SVL in males (range 0.92–1.24) and 1.20 in females (range 1.06–1.38). Limbs are moderately long; FLL + HLL averages 59% of SVL in males (range 55–64%) and 57% in females (range 53–62%). Adpressed limbs closely approach or overlap in males (mean LI −0.6, range −1 to 1) and females (mean LI 0.0, range −1 to 1). The manus and pes are relatively wide; digital tips are somewhat expanded and there are distinct subterminal pads. Digital webbing extends to the base of the terminal phalanx. The first (innermost) digit, while distinct, is included in the web except at its tip. Mental glands are large, oval-shaped and relatively prominent in males. The smallest male with a mental gland is 32.0 mm SVL. Parotoid glands are well marked in some individuals but less evident in others.

**Comparisons:** *Chiropterotriton totonacus* differs from *C. ceronorum* in its larger adult body size (mean SVL 35.7 mm in male and 35.5 mm in female *C. totonacus* vs. 33.9 mm in male and 34.9 mm in female *C. ceronorum*), longer tail (mean TL/SVL 1.16 in male and 1.20 in female *C. totonacus* vs. 1.0 in male and 0.97 in female *C. ceronorum*), longer limbs (mean LI −0.6 in male and 0.0 in female *C. totonacus* vs. 0.0 in male and 1.5 in female *C. ceronorum*), longer head (mean HL 8.5 mm in male and 7.6 mm in female *C. totonacus* vs. 7.5 mm in male and 7.1 mm in female *C. ceronorum*), slightly larger feet (mean FW 4.2 mm in male and 4.0 mm in female *C. totonacus* vs. 3.8 mm in male and 3.5 mm in female *C. ceronorum*), more maxillary teeth (mean MT 32.9 in male and 52.6 in female *C. totonacus* vs. 11.0 in male and 47.7 in female *C. ceronorum*) and fewer vomerine teeth (mean VT 11.6 in male and 13.7 in female *C. totonacus* vs. 13.0 in male and 15.9 in female *C. ceronorum*).

*Chiropterotriton totonacus* differs from *C. perotensis* in its larger adult body size (mean SVL 35.7 mm in male and 35.5 mm in female *C. totonacus* vs. 29.7 mm in male and 31.7 mm in female *C. perotensis*), longer tail (mean TL/SVL 1.16 in male and 1.20 in female *C. totonacus* vs. 1.0 in both male and female *C. perotensis*), longer limbs (mean LI −0.60 in male and 0.0 in female *C. totonacus* vs. 2.5 in male and 3.3 in female *C. perotensis*), longer head (mean HL 8.5 mm in male and 7.6 mm in female C. *totonacus* vs. 6.6 mm in male and 6.7 mm in female *C. perotensis*), broader head (mean HW 5.2 mm in both male and female *C. totonacus* vs. 4.2 mm in male and 4.4 mm in female *C. perotensis*), larger feet (mean FW 4.2 mm in male and 4.0 mm in female *C. totonacus* vs. 2.6 mm in both male and female *C. perotensis*), more maxillary teeth (mean MT 32.9 in male and 52.6 in female *C. totonacus* vs. 7.2 in male and 27.9 in female *C. perotensis*) and more vomerine teeth (mean VT 11.6 in male and 13.7 in female *C. totonacus* vs. 9.0 in male and 11.1 in female *C. perotensis*).

*Chiropterotriton totonacus* differs from *C. melipona* in its larger adult body size (mean SVL 35.7 mm in male and 35.5 mm in female *C. totonacus* vs. 29.2 mm in male and 28.5 mm in female *C. melipona*), longer tail in females (mean TL/SVL 1.20 in *C. totonacus* vs. 1.11 in *C. melipona*), longer limbs (mean LI −0.60 in male and 0.0 in female *C. totonacus* vs. 2.3 in male and 1.8 in female *C. melipona*), longer head (mean HL 8.5 mm in male and 7.6 mm in female C. *totonacus* vs. 6.3 mm in male and 6.4 mm in female *C. melipona*), broader head (mean HW 5.2 mm in both male and female *C. totonacus* vs. 4.3 mm in male and 4.2 mm in female *C. melipona*), larger feet (mean FW 4.2 mm in male and 4.0 mm in female *C. totonacus* vs. 2.4 mm in male and 2.6 mm in female *C. melipona*) and more maxillary teeth (mean MT 32.9 in male and 52.6 in female *C. totonacus* vs. 9.5 in male and 31.0 in female *C. melipona*).

*Chiropterotriton totonacus* differs from *C. casasi* in its smaller adult body size (mean SVL 35.7 mm in male and 35.5 mm in female *C. totonacus* vs. 37.8 mm in male and 40.9 mm in one female *C. casasi*), longer limbs (mean LI −0.6 in male and 0.0 in female *C. totonacus* vs. 0.80 in male and 1.0 in one female *C. casasi*), narrower head (mean HW 5.2 mm in both male and female *C. totonacus* vs. 5.8 mm in male and 5.9 mm in one female *C. casasi*), larger feet (mean FW 4.2 mm in male and 4.0 mm in female *C. totonacus* vs. 3.7 mm in both male and one female *C. casasi*) and fewer maxillary teeth (mean MT 32.9 in male and 52.6 in female *C. totonacus* vs. 9.0 in male and 30 in one female *C. casasi*).

*Chiropterotriton totonacus* differs from *C. chiropterus* in its smaller adult body size in males (mean SVL 35.7 mm in *C. totonacus* vs. 37.5 mm in *C. chiropterus*), shorter tail (mean TL/SVL 1.16 in male and 1.20 in female *C. totonacus* vs. 1.25 in male and 1.19 in female *C. chiropterus*), longer limbs (mean LI −0.60 in male and 0.0 in female *C. totonacus* vs. 0.3 in male and 2.0 in female *C. chiropterus*), longer head (mean HL 8.5 mm in male and 7.6 mm in female C. *totonacus* vs. 8.1 mm in male and 7.3 mm in female *C. chiropterus*), larger feet in males (mean FW 4.2 mm in *C. totonacus* vs. 3.7 mm in *C. chiropterus*), more maxillary teeth (mean MT 32.9 in male and 52.6 in female *C. totonacus* vs. 12.6 in male and 48.0 in female *C. chiropterus*) and more vomerine teeth (mean VT 11.6 in male and 13.7 in female *C. totonacus* vs. 10.6 in male and 12.5 in female *C. chiropterus*).

*Chiropterotriton totonacus* differs from *C. orculus* in its smaller adult body size in females (mean SVL 35.5 mm in *C. totonacus* vs. 39.0 mm in *C. orculus*), longer tail (mean TL/SVL 1.16 in male and 1.20 in female *C. totonacus* vs. 1.0 in both male and female *C. orculus*), longer limbs (mean LI −0.60 in male and 0.0 in female *C. totonacus* vs. 1.9 in male and 2.9 in female *C. orculus*), longer head in males (mean HL 8.5 mm in C. *totonacus* vs. 7.4 mm in *C. orculus*), larger feet (mean FW 4.2 mm in male and 4.0 mm in female *C. totonacus* vs. 3.2 mm in male and 3.4 mm in female *C. orculus*), more maxillary teeth (mean MT 32.9 in male and 52.6 in female *C. totonacus* vs. 8.2 in male and 28.8 in female *C. orculus*) and more vomerine teeth (mean VT 11.6 in male and 13.7 in female *C. totonacus* vs. 8.6 in male and 12.0 in female *C. orculus*).

*Chiropterotriton totonacus* differs from *C. lavae* in its larger adult body size (mean SVL 35.7 mm in male and 35.5 mm in female *C. totonacus* vs. 32.4 mm in male and 31.6 mm in female *C. lavae*), longer tail in females (mean TL/SVL 1.20 in *C. totonacus* vs. 1.02 in *C. lavae*), longer limbs in females (mean LI 0.0 in *C. totonacus* vs. 0.6 in *C. lavae*), longer head (mean HL 8.5 mm in male and 7.6 mm in female C. *totonacus* vs. 7.5 mm in male and 7.0 mm in female *C. lavae*), slightly broader head (mean HW 5.2 mm in both male and female *C. totonacus* vs. 4.9 mm in male and 4.7 mm in female *C. lavae*), larger feet (mean FW 4.2 mm in male and 4.0 mm in female *C. totonacus* vs. 3.7 mm in male and 3.3 mm in female *C. lavae*), more maxillary teeth (mean MT 32.9 in male and 52.6 in female *C. totonacus* vs. 7.0 in male and 20.8 in female *C. lavae*) and more vomerine teeth (mean VT 11.6 in male and 13.7 in female *C. totonacus* vs. 8.9 in male and 11.4 in female *C. lavae*).

*Chiropterotriton totonacus* differs from *C. aureus* in its larger adult body size (mean SVL 35.7 mm in male and 35.5 mm in female *C. totonacus* vs. 28.5 mm in one male and 26.8 mm in female *C. aureus*), longer limbs (mean LI −0.6 in male and 0.0 in female *C. totonacus* vs. 2.0 in one male and 2.3 in female *C. aureus*), longer head (mean HL 8.5 mm in male and 7.6 mm in female *C. totonacus* vs. 6.4 mm in one male and 6.0 mm in female *C. aureus*), larger feet (mean FW 4.2 mm in male and 4.0 mm in female *C. totonacus* vs. 2.4 mm in one male and 1.8 mm in female *C. aureus*), more maxillary teeth (mean MT 32.9 in male and 52.6 in female *C. totonacus* vs. 10.0 in one male and 38.3 in female *C. aureus*) and fewer vomerine teeth (mean VT 11.6 in male and 13.7 in female *C. totonacus* vs. 15.0 in one male and 12.3 in female *C. aureus*).

*Chiropterotriton totonacus* differs from *C. nubilus* in its larger adult body size (mean SVL 35.7 mm in male and 35.5 mm in female *C. totonacus* vs. 29.4 mm in one male and 30.5 mm in female *C. nubilus*), longer limbs (mean LI −0.6 in male and 0.0 in female *C. totonacus* vs. 2.0 in one male and 1.5 in female *C. nubilus*), larger feet (mean FW 4.2 mm in male and 4.0 mm in female *C. totonacus* vs. 2.6 mm in one male and 2.3 mm in female *C. nubilus*), and more maxillary teeth (mean MT 32.9 in male and 52.6 in female *C. totonacus* vs. 13.0 in one male and 41.5 in female *C. nubilus*).

**Description of holotype:** SVL 35.8 mm, TL 49.2 mm, AX 18.3 mm, SW 3.7 mm, HL 7.7 mm, HW 5.3 mm, HD 2.4 mm, projection of snout beyond mandible 0.7 mm, distance from anterior rim of orbit to snout 2.2 mm, interorbital distance 2.0 mm, eyelid length

2.2 mm, eyelid width 1.2 mm, nostril diameter 0.2 mm, FLL 9.9 mm, HLL 11.5 mm, snout-to-forelimb length 12.4 mm, snout to anterior angle of vent 33.5 mm, tail width at base 3.0 mm, tail depth at base 2.7 mm, FW 4.6 mm, length of fifth toe 0.8 mm, length of third (longest) toe 1.8 mm. Numbers of teeth: premaxillary 6, maxillary 27-23 (right-left) and vomerine 7-7 (right-left). Tips of adpressed limbs meet.

**Coloration in life:** No color information is available for the type series in life; this description is based on photos of three recently collected specimens (IBH 31030, 31031, IBH 30998). Dorsal background very dark brownish gray. Broad, reddish-brown dorsal band with background color showing only along midline (IBH 31030), broken and irregular (IBH 30998), or completely absent (IBH 31031). Small, pale gray specks present in some specimens. Dorsal surface of tail similar to dorsal coloration on body. IBH 30998 has two orangish-brown blotches at base of tail. Head dark gray with brown blotches or gray specks, similar to dorsal coloration. Paratoid region brownish in specimens with a regular or irregular dorsal band present, gray in IBH 31031 Flanks and upper surface of limbs medium gray with small pale gray and brown flecks, numerous in some specimens while nearly absent in others; toe tips reddish. Gular region pale gray; ventral surface of body, tail, and limbs medium gray. Iris dark golden-brown.

**Coloration in preservative:** The holotype is medium brown with an obscure dorsal stripe, darker brown along the margin and more reddish brown on the stipe with a narrow darker median line. The head is medium brown with a light bar extending between the eyes and snout mottled with dark cream and brown. Limbs mottled with light brown upper limbs especially near the body, darker lower limbs with light tan digits. The venter is mainly pale with some mottled darker brown. The gular region is mottled with dark cream and brown. Undersides of the tail are paler than its lateral surfaces. One specimen (MVZ 163943) has a distinct yellowish stripe bordered laterally by a very dark band of pigment, with the stripe extending to the tip of the tail. Most others are uniformly pale brown to tan dorsally with some darker brown. One individual (MVZ 163947) is generally paler gray brown.

**Osteology** This account is based on examination of a μCT scan of the anterior skeleton of MVZ 163945, an adult female, 35.8 mm SVL (Figs. 6–8; Table 3). The skull is relatively broad and somewhat ovoid in dorsal and ventral views. Many of the dermal investing bones are thin and weakly ossified, especially anteriorly. Paired frontal bones extend anterolaterally, but they are largely eroded anteromedially except for a pair of anteriorly directed spikes along the midline (one per side). Each frontal has a posterolateral tab that overlaps the adjacent parietal, but otherwise these bones only weakly articulate with one another, leaving a moderately sized frontoparietal fontanel. The single premaxilla is delicate and lacks a palatal shelf. Ascending processes initially approach one another but then diverge posterodorsally until they articulate with the weak anterior end of the frontal bone. They enclose a huge internasal fontanel, but unlike in many congeners they are not expanded posteriorly. There are no septomaxillary bones. The nasal bone is triangular but irregular in outline. It barely articulates with the facial process of the maxilla and with

the frontal but is separate from the prefrontal, which is relatively small—smaller than the nasal. The foramen of the nasolacrimal duct has eroded the anteroventral margin of the prefrontal, the posterior margin of the nasal and the dorsal margin of the facial process of the maxilla. Teeth are deployed along nearly the entire length of the maxillary bone, leaving only a small edentulous portion at its posterior tip. There are 21 maxillary teeth on each side and six premaxillary teeth. The orbitosphenoid is shortened anteroposteriorly and rather thin. It is only weakly articulated to the parasphenoid and is mostly separated from both the frontal and the parietal.

Otic capsules lack crests except for a slight projection along the anterolateral margin of each lateral semicircular canal. However, the anteromedial edge of each capsule is overlapped by a bony shelf that extends from the posterolateral portion of the adjacent parietal bone. A relatively large, triangular tab descends from the posterolateral margin of the parietal. The tab is sharply reflected ventromedially and ends in a rounded point at about the midpoint of the vertical extent of the orbitosphenoid. The roughly triangular squamosal articulates with the otic capsule dorsally. The quadrate bone is relatively small and incompletely ossified. The columella bears a pronounced stylus. Paired vomers are relatively large, but the body of each bone is very weakly ossified anteriorly. They do not articulate at the midline. Preorbital processes are very long. There are six teeth on the left side and five on the right; one or two are deployed at the base of each preorbital process. The parasphenoid bone is triangular. Paired parasphenoid tooth patches progressively broaden posteriorly and then round off caudally. There are 80–85 teeth in each patch. The mandible is relatively weak. The articular bone is poorly ossified. The prearticular bone is small, with a relatively low coronoid process. Each dentary bone bears 24 teeth.

Digital formulae are 1-2-3-2 right and 1-2-2-2 left. The distal tip of the terminal phalanx is slightly expanded on each finger. Mesopodial cartilages are not mineralized.

**Distribution and ecology:** *Chiropterotriton totonacus* is known from Veracruz, Mexico, along the ridge between Cruz Blanca and Las Vigas at elevations between 2,200 and 2,450 masl, and from La Joya at 2,000 masl. It occurs in mossy pine forest and is terrestrial. Recently collected specimens were found under logs in disturbed pine forest.

**Remarks:** This species occurs in sympatry at the upper end of its range above Las Vigas with *P. leprosa* and *Thorius munificus*, and at the lower end of its range near La Joya with *Chiropterotriton lavae, Pseudoeurycea lynchi, Thorius minydemus,* and *Isthmura gigantea*, and throughout is range with *Aquiloeurycea cephalica*. We think this is *Darda's (1994)* species E (his population 7), which he assigned to *C. chiropterus*. It differs from *C. lavae* by two fixed allozymic differences and a Nei D value of 0.148, but we have no samples of a second species (in addition to *C. lavae*) from La Joya so our assignment of Darda's material must be viewed as tentative. He had no specimens from the area west of Las Vigas or Cruz Blanca. If we assume that Darda's species E is assignable to *C. totonacus*, it is surprising that it is so distinct from *C. perotensis* (seven fixed differences, Nei *D* = 0.725). It is closer to Darda's species C from Puerto del Aire (3 fixed differences)

and I from regions to the south of Pico de Orizaba (5 fixed differences), the latter here named *C. ceronorum*. We are not yet prepared to deal with species C at this time.

Chiropterotriton totonacus has long been known from the Las Vigas-Cruz Blanca area, and from Toxtlacoaya, above La Joya (*Taylor & Smith, 1945*; *Smith & Taylor, 1948*). The species was reported to occur under clumps of dead grass, under and in rotten logs, under loose bark, and in stump holes that had filled with pine needles and loose earth.

**Conservation status:** Most of the pine forest around Las Vigas de Ramírez has been cut down or fragmented into very small patches. Recently, we found three specimens (one in 2016 and two in 2017) in a secondary pine forest near the type locality at Cruz Blanca. This secondary forest, which is highly disturbed and has few logs or cover objects where salamanders could be found, is the only place where the species is currently known to occur given that nearly all forest from the type locality has been logged. The largest extent of remaining forest in the area is in the "Bosque Estatal San Juan del Monte", but *C. totonacus* has not been found there despite survey efforts. Based on its scarcity and very limited geographic range, we recommend that this species be designated as Critically Endangered under IUCN Red List criterion B1ab(iii) (extent of occurrence <100 km$^2$, distribution severely fragmented with continuing decline in area, extent, and quality of habitat; *IUCN, 2012*).

**Etymology:** The specific epithet refers to the native Totonac culture of the central region of Veracruz where *Chiropterotriton totonacus* is found.

### *Chiropterotriton melipona*, sp. nov.
Xicotepec Salamander, Salamandra de Xicotepec
Figures 4J–4L, 5D, 6D, 7D, 8D.

Chresonymy
*Chiropterotriton* sp. F.—*Darda, 1994*; *Parra-Olea, 2003*; *Raffaëlli, 2007*; *Raffaëlli, 2013*; *Rovito & Parra-Olea, 2015*; *García-Castillo et al., 2017*; *García-Castillo et al., 2018*.

**Holotype:** MVZ 200726, an adult male from Xicotepec de Juárez, 3.3 km S of Hotel Mi Ranchito on Mexican Hwy. 130, 2.1 km E on road to La Unión, Puebla, México, 1,080 masl, 20.227755° N, 97.953269° W (EPE = max. error distance 1.0 km). Collected 8 December 1983 by D.M. Darda and P.A. Garvey.

**Paratypes:** Seven specimens, all from Puebla, Mexico. Four males: MVZ 178706 and 178708, 3.9 km S of Xicotepec de Juárez on Mexican Hwy. 130; MVZ 200723–24, Xicotepec de Juárez, Mexican Hwy. 130, 21 km E on road to La Unión. Three females: MVZ 178707, 3.9 km S of Xicotepec de Juarez on Mexican Hwy. 130; MVZ 185972, 2.2 km on road to Patla from junction with Mexican Hwy. 130 SW out of Xicotepec de Juárez; MVZ 200725, Xicotepec de Juárez, Mexican Hwy. 130, 21 km E on road to La Unión.

**Referred specimens:** Two specimens: IBH 30112 and MVZ 133019, Cuetzalan, Puebla, Mexico.

**Diagnosis:** This is a small species of plethodontid salamander phylogenetically related to *Chiropterotriton chiropterus*; mean SVL 29.2 mm in four adult males (range 26.4–31.4) and 28.5 mm in three adult females (range 27.1–29.8). The head is moderately wide; HW averages 15% of SVL in both males and females (range 14–15%). Adults have a broad, bluntly rounded snout and adult males have moderately developed nasolabial protuberances. Eyes are large and prominent and extend laterally beyond the jaw margin in ventral view. There are few maxillary teeth in males (mean MT 9.5, range 7–12) and moderate numbers of teeth in females (mean MT 31.0, range 25–34). There are few vomerine teeth in both males (mean VT 11.0, range 8–15) and females (mean VT 13.0, range 9–19), which are arranged in a row that does not extend lateral to the outer margin of the internal choana. The tail is long and slender and exceeds SVL in all adults with complete tails; mean TL/SVL 1.16 in males (range 1.10–1.22) and 1.11 in females (range 1.03–1.18). Limbs are short; FLL + HLL averages 46% of SVL in males (range 39–50) and 49% in females (range 46–52). Adpressed limbs are widely separated and never overlap in males (mean LI 2.3, range 2–2.5) and females (mean LI 1.8, range 1.0–2.5). Manus and pes are relatively small; digits are slender and their tips only slightly expanded. Digital webbing ranges from slight to absent and is limited to the metatarsal region. The first digit is distinct but largely included in the webbing. Subterminal pads are small but well developed. A relatively small, rounded to oval-shaped mental gland present in most adult males. The smallest adult male (pigmented testes) is 26.4 mm SVL; the smallest male with a mental gland is 28.5 mm SVL. Parotoid glands are not evident.

**Comparisons:** *Chiropterotriton melipona* differs from *C. ceronorum* in its smaller adult body size (mean SVL 29.2 mm in male and 28.5 mm in female *C. melipona* vs. 33.9 mm in male and 34.9 mm in female *C. ceronorum*), shorter tail (mean TL/SVL 1.16 in male and 1.11 in female *C. melipona* vs. 1.0 in male and 0.97 in female *C. ceronorum*), shorter head (mean HL 6.3 mm in male and 6.4 mm in female *C. melipona* vs. 7.5 mm in male and 7.1 mm in female *C. ceronorum*), narrower head (mean HW 4.3 mm in male and 4.2 mm in female *C. melipona* vs. 5.1 mm in both male and female *C. ceronorum*), shorter limbs in males (mean LI 2.3 in *C. melipona* vs. 0.0 in *C. ceronorum*), narrower feet (mean FW 2.4 mm in male and 2.6 mm in female *C. melipona* vs. 3.8 mm in male and 3.5 mm in female *C. ceronorum*), fewer maxillary teeth (mean MT 9.5 in male and 31.0 in female *C. melipona* vs. 11.0 in male and 47.7 in female *C. ceronorum*) and fewer vomerine teeth (mean VT 11.0 in male and 13.0 in female *C. melipona* vs. 13.0 in male and 15.9 in female *C. ceronorum*).

*Chiropterotriton melipona* differs from *C. perotensis* in its slightly smaller adult body size (mean SVL 29.2 mm in male and 28.5 mm in female *C. melipona* vs. 29.7 mm in male and 31.7 mm in female *C. perotensis*), shorter tail (mean TL/SVL 1.16 in male and 1.11 in female *C. melipona* vs. 1.0 in both male and female *C. perotensis*), shorter head (mean HL 6.3 mm in male and 6.4 mm in female *C. melipona* vs. 6.6 mm in male and 6.7 mm in female *C. perotensis*), more maxillary teeth (mean MT 9.5 in male and 31.0 in female *C. melipona* vs. 7.2 in male and 27.9 in female *C. perotensis*) and fewer vomerine

teeth (mean VT 11.0 in male and 13.0 in female *C. melipona* vs. 9.0 in male and 11.1 in female *C. perotensis*).

 *Chiropterotriton melipona* differs from *C. totonacus* in its smaller adult body size (mean SVL 29.2 mm in male and 28.5 mm in female *C. melipona* vs. 35.7 mm in male and 35.5 mm in female *C. totonacus*), shorter head (mean HL 6.3 mm in male and 6.4 mm in female *C. melipona* vs. 8.5 mm in male and 7.6 mm in female *C. totonacus*), narrower head (mean HW 4.3 mm in male and 4.2 mm in female *C. melipona* vs. 5.2 mm in both male and female *C. totonacus*), shorter limbs (mean LI 2.3 in male and 1.8 in female *C. melipona* vs. −0.6 in male and 0.0 in female *C. totonacus*), narrower feet (mean FW 2.4 mm in male and 2.6 mm in female *C. melipona* vs. 4.2 mm in male and 4.0 mm in female *C. totonacus*) and more maxillary teeth (mean MT 9.5 in male and 31.0 in female *C. melipona* vs. 32.9 in male and 52.6 in female *C. totonacus*).

 *Chiropterotriton melipona* differs from *C. casasi* in its smaller adult body size (mean SVL 29.2 mm in male and 28.5 mm in female *C. melipona* vs. 37.8 mm in male and 40.9 mm in one female *C. casasi*), shorter tail in males (mean TL/SVL 1.16 in *C. melipona* vs. 1.0 in *C. casasi*), shorter head (mean HL 6.3 mm in male and 6.4 mm in female *C. melipona* vs. 8.3 mm in male and 8.6 mm in one female *C. casasi*), narrower head (mean HW 4.3 mm in male and 4.2 mm in female *C. melipona* vs. 5.8 mm in male and 5.9 mm in one female *C. casasi*), shorter limbs (mean LI 2.3 in male and 1.8 in female *C. melipona* vs. 0.8 in male and 1.0 in one female *C. casasi*) and narrower feet (mean FW 2.4 mm in male and 2.6 mm in female *C. melipona* vs. mean 3.7 mm in both male and one female *C. casasi*).

 *Chiropterotriton melipona* differs from *C. chiropterus* in its smaller adult body size (mean SVL 29.2 mm in male and 28.5 mm in female *C. melipona* vs. 37.5 mm in male and 33.5 mm in female *C. chiropterus*), shorter tail in males (mean TL/SVL 1.16 in *C. melipona* vs. 1.25 in *C. chiropterus*), shorter head (mean HL 6.3 mm in male and 6.4 mm in female *C. melipona* vs. 8.1 mm in male and 7.3 mm in female *C. chiropterus*), narrower head (mean HW 4.3 mm in male and 4.2 mm in female *C. melipona* vs. 5.6 mm in male and 4.8 mm in female *C. chiropterus*), shorter limbs in males (mean LI 2.3 in *C. melipona* vs. 0.3 in *C. chiropterus*), narrower feet (mean FW 2.4 mm in male and 2.6 mm in female *C. melipona* vs. 3.7 mm in male and 3.1 mm in female *C. chiropterus*) and fewer maxillary teeth (mean MT 9.5 in male and 31.0 in female *C. melipona* vs. 12.6 in male and 48.0 in female *C. chiropterus*).

 *Chiropterotriton melipona* differs from *C. orculus* in its smaller adult body size (mean SVL 29.2 mm in male and 28.5 mm in female *C. melipona* vs. 35.9 mm in male and 39.0 mm in female *C. orculus*), shorter tail (mean TL/SVL 1.16 in male and 1.11 in female *C. melipona* vs. 1.02 in both male and female *C. orculus*), shorter head (mean HL 6.3 mm in male and 6.4 mm in female *C. melipona* vs. 7.4 mm in male and 8.0 mm in female *C. orculus*), narrower head (mean HW 4.3 mm in male and 4.2 mm in female *C. melipona* vs. 5.0 mm in male and 5.2 mm in female *C. orculus*), shorter limbs in males (mean LI 2.3 in *C. melipona* vs. 1.9 in *C. orculus*), narrower feet (mean FW 2.4 mm in male and 2.6 mm in female *C. melipona* vs. 3.2 mm in male and 3.4 mm in female *C. orculus*), more maxillary teeth (mean MT 9.5 in male and 31.0 in female *C. melipona* vs.

8.2 in male and 28.8 in female *C. orculus*) and more vomerine teeth (mean VT 11.0 in male and 13.0 in female *C. melipona* vs. 8.6 in male and 12.0 in female *C. orculus*).

*Chiropterotriton melipona* differs from *C. lavae* in its smaller adult body size (mean SVL 29.2 mm in male and 28.5 mm in female *C. melipona* vs. 32.4 mm in male and 31.6 mm in female *C. lavae*), shorter head (mean HL 6.3 mm in male and 6.4 mm in female *C. melipona* vs. 7.5 mm in male and 7.0 mm in female *C. lavae*), narrower head (mean HW 4.3 mm in male and 4.2 mm in female *C. melipona* vs. 4.9 mm in male and 4.7 mm in female *C. lavae*), shorter limbs (mean LI 2.3 in male and 1.8 in female *C. melipona* vs. −0.6 in male and 0.6 in female *C. lavae*), narrower feet (mean FW 2.4 mm in male and 2.6 mm in female *C. melipona* vs. 3.7 mm in male and 3.3 mm in female *C. lavae*), more maxillary teeth (mean MT 9.5 in male and 31.0 in female *C. melipona* vs. 7.0 in male and 20.8 in female *C. lavae*) and more vomerine teeth (mean VT 11.0 in male and 13.0 in female *C. melipona* vs. 8.9 in male and 11.4 in female *C. lavae*).

*Chiropterotriton melipona* differs from *C. aureus* in its larger adult body size (mean SVL 29.2 mm in male and 28.5 mm in female *C. melipona* vs. 28.5 mm in one male and 26.8 mm in female *C. aureus*), shorter tail in females (mean TL/SVL 1.11 in female *C. melipona* vs. 1.16 in female *C. aureus*), wider head (mean HW 4.3 mm in male and 4.2 mm in female *C. melipona* vs. 4.0 mm in one male and 3.6 mm in female *C. aureus*), longer limbs in females (mean LI 1.8 in female *C. melipona* vs. 2.3 in female *C. aureus*), and wider feet in females (mean FW 2.6 mm in female *C. melipona* vs. 1.8 mm in female *C. aureus*).

*Chiropterotriton melipona* differs from *C. nubilus* in having a shorter head (mean HL 6.3 mm in male and 6.4 mm in female *C. melipona* vs. 6.6 mm in one male and 7.4 mm in female *C. nubilus*), and fewer maxillary teeth (mean MT 9.5 in male and 31.0 in female *C. melipona* vs. 13 in one male and 41.5 in female *C. nubilus*).

**Description of holotype:** SVL 28.5 mm, TL 31.4 mm, AX 15.5 mm, SW 3.3 mm, HL 6.3 mm, HW 4.1 mm, HD 2.1 mm, projection of snout beyond mandible 0.7 mm, distance from anterior rim of orbit to snout 1.5 mm, interorbital distance 1.4 mm, distance between corners of eyes 2.2 mm, interorbital width 1.3 mm, eyelid length 1.7 mm, eyelid width 0.9 mm, nostril diameter 0.2 mm, FLL 5.1 mm, HLL 6.1 mm, snout-to-forelimb length 8.4 mm, distance from snout to anterior angle of vent 24.4 mm, snout to gular fold distance 6.3 mm, tail depth at base 2.7 mm and FW 2.2 mm. Numbers of teeth: premaxillary 3, maxillary 4-4 (right-left) and vomerine 7-8 (right-left). Adpressed limbs are separated by 2.5 costal folds.

**Coloration in life:** Color notes in life are not available for the type series of this species, thus we describe coloration from a photo of one of the referred specimens (IBH 30112). The head is dark brown with numerous pale gray specks on the rostrum, sides of head, interocular region, and eyelids. This brown coloration with gray specks extends from behind each eye in an inverted triangle to the nuchal region. Both sides of this triangle in parotoid region are orangish-brown. Orange-brown coloration extends in a band along dorsum and along the dorsal side of tail, where it is more yellowish along midline and

orangish-brown along edges. Flanks are dark brown with numerous pale gray specks. Limbs gray-brown with some pale yellow-brown specks; manus and pes grayish. Sides of tail dark brown. Iris coppery.

**Coloration in preservative:** The holotype, while faded, is generally bright yellow to yellowish tan. The snout is pale yellow with scattered brown pigment. A broad, bright yellow dorsal stripe extends from the eyes to the tip of the tail. It is bordered by a dark stripe that arises at the eye and extends posteriorly onto the tail. This dark stripe, in turn, is bordered by a pale brown stripe that becomes paler ventrolaterally. The venter is very pale, almost pigmentless. The tail has some light brown pigment along its lateral margins. Paratypes all faded but yellowish tan with a pale yellowish tan dorsal stripe evident in all individuals to some degree. Dorsal stripe always bordered by a thin dorsal lateral light brown stripe. Venter very pale. Manus and pes are pale.

**Osteology:** This account is based on examination of a μCT scan of the anterior skeleton of MVZ 178706, an adult male, 28.5 mm SVL, which may be sexually immature and not representative of the adult condition (Figs. 6–8; Table 3). The skull is weakly developed and delicate, both in general and relative to other members of the genus such as *C. chiropterus*, and even *C. casasi*. Cranial roofing bones are very thin. Frontals are weakly articulated with each other and with the paired parietals, leaving a relatively large frontoparietal fontanel that extends both anteroposteriorly (in the midline) and transversely (at the frontal-parietal interface). Paired ascending processes of the single premaxilla begin diverging immediately dorsal to the dental process. They continue to diverge posterolaterally and ultimately articulate in grooves on the anterior part of the paired frontals, enclosing a large internasal fontanel. Unlike in many other congeners, they remain thin and are not expanded at their dorsal ends. A palatal shelf is barely evident on the premaxillary. Tiny paired septomaxillae lie approximately at the level of the articulation between premaxilla and maxilla. Nasal bones are expansive but otherwise weakly developed, with indistinct borders anteriorly and weak articulations with adjacent bones, including both the prefrontal and the maxilla. The prefrontal is well articulated with the facial process of the maxilla ventrally and overlaps the frontal dorsally. A foramen for the passage of the nasolacrimal duct is framed by the anterior margin of the prefrontal, the posterolateral margin of the nasal, and the dorsal midportion of the facial process of the maxilla. There are five large teeth on the anterior portion of each maxilla. The posterior half of the bone lacks teeth and resembles a shallow cleaver. There are three premaxillary teeth. The orbitosphenoid is fairly well developed, although not well articulated with the parietal. In general, the braincase is moderately well developed.

There is a nascent bony crest on the otic capsule above the anterior semicircular canal where it abuts a bony shelf that extends posterolaterally from the parietal. The parietal also bears a moderately developed, posterolateral tab that is sharply directed ventomedially. The tab is triangular and ends in a rounded point at a level about halfway through the vertical extent of the orbitosphenoid. The squamosal is a roughly triangular bone that articulates dorsally with the otic capsule opposite the lateral semicircular canal. In lateral

view, its ventral portion appears to buttress the otic capsule ventral to the lateral semicircular canal, but when viewed from different angles these bones can be seen to be well separated. The quadrate bone is relatively small and inconspicuous. The columellar stylus is distinct, cylindrical and long. Paired vomers are relatively robust; they barely articulate in the midline posterior to the internasal fontanel. Preorbital processes are very long. There are four-to-six vomerine teeth on each side; two or three of these are deployed at the base of each preorbital process. The parasphenoid bone is broadly triangular. Paired parasphenoid tooth patches are well separated from each other and from the vomerine teeth anteriorly. Each patch bears approximately 75 teeth. The mandible is unremarkable. The articular bone is poorly ossified. The prearticular bone has a coronoid process of moderate height. There are seven teeth on the right dentary bone and eight on the left.

Digital formula is 1-2-3-2 on each side. There is a slightly expanded knob at the tip of the terminal phalanx on the two longest fingers of each hand (digits 3 and 4). Mesopodial cartilages are not mineralized.

**Distribution and ecology:** *Chiropterotriton melipona* is known from the Sierra Norte in the northernmost part of Puebla near Cuetzalan, Xocoyolo and Xicotepec de Juarez at elevations between 690 and 1,420 masl. It likely occurs between known localities near Cuetzalan and Xicotepec. This range includes the lowest elevational record of any known species of the genus. The species is arboreal and has been collected from banana plants and bromeliads and has been found in sympatry with *Aquiloeurycea quetzalanensis*.

**Remarks:** This species was included in *Darda's (1994)* electrophoretic study as population 19, new species F. It was most similar to populations 12 (*C. lavae*; three fixed differences, Nei $D = 0.22$) and 19 (new species F, sympatric with *C. lavae*; three fixed differences, Nei $D = 0.23$). The lowland population from near Cuetzalan is discussed and illustrated by *Raffaëlli (2013)*. We think the specimens he describes are assignable to *C. melipona*. He reports them at 780 masl, in the outer leaves of bananas.

**Conservation status:** Most mature forest at known localities for this species has been cut down, and the species has recently been found in small patches of forest and secondary vegetation, as well as cafetales. Because of the highly fragmented nature and decreasing quality of forest habitat within its range, we recommend that the species be designated as Endangered based on IUCN criterion B1ab(iii) (extent of occurrence <5,000 km$^2$, distribution severely fragmented continuing decline in extent, and quality of habitat; *IUCN, 2012*).

**Etymology:** Xicotepec, the name of the type locality, comes from the Nahuatl language and means "place of the jicotes." Jicotes are stingless bees of the genus *Melipona*. The name used for this species is a noun in apposition referring to the genus *Melipona*.

**Chiropterotriton casasi, sp. nov.**
Tlapacoyan Salamander, Salamandra de Tlapacoyan
Figures 2, 4M–4O, 5E, 6A, 7A, 8A.

**Holotype:** MVZ 92874, an adult male from 13 mi SW Tlapacoyan, Veracruz, Mexico, 19.868483° N, 97.301500° W (EPE = max. error distance 2 km). The elevation is between 1,450 and 1,550 masl. Collected 26 December 1969 by R. Altig.

**Paratypes:** Four males, MVZ 92875 and 92877–79, and one female, MVZ 92876, all from the type locality.

**Diagnosis:** This is a relatively large species of *Chiropterotriton* that stands out from other species considered here in being relatively stout and long legged, and being morphologically distinct; mean SVL 37.8 mm in four adult males (range 34.5–42.0). Only one female has been collected, SVL 40.9 mm. The head is moderately wide; HW averages 16% of SVL in males (range 13–17%) and 14% in the female. In males, the snout is broad and truncated. Jaw muscles are pronounced and visible as a bulging mass immediately caudal to the eyes. Eyes are moderately protuberant and extend laterally beyond the jaw margin in ventral view. There are few maxillary teeth in males (mean MT 9.0, range 6–13) but they are more numerous in the female (MT 30). There are few vomerine teeth in males (mean VT 9.0, range 8–11) and the female (VT 13), which are arranged in a row that extends to, or just lateral to, the inner margin of the internal choana. The tail is moderately long; mean TL equals 1.0 of SVL in males (range 0.90–1.15). Limbs are short and slender; FLL + HLL averages 57% of SVL in males (range 55–60) and 55% in the female. Adpressed limbs approach closely in males (mean LI 0.8, range 0.0–1) and are separated by one costal fold in the female. Digits are long and slender with blunt tips, distinct subterminal pads, and moderate webbing that extends onto the penultimate phalanx of the third toe. Digits II–V are discrete, while digit I is very short and does not extend beyond the webbing. The outermost toes are particularly well developed. The mental gland is oval-shaped in adult males. The smallest male with a mental gland is 37.2 mm SVL. Parotoid glands are not evident.

**Comparisons:** *Chiropterotriton casasi* differs from *C. ceronorum* in its larger adult body size (mean SVL 37.8 mm in male and 40.9 mm in one female *C. casasi* vs. 33.9 mm in male and 34.9 mm in female *C. ceronorum*), longer head (mean HL 8.3 mm in male and 8.6 mm in one female *C. casasi* vs. 7.5 mm in male and 7.1 mm in female *C. ceronorum*), broader head (mean HW 5.8 mm in male and 5.9 mm in one female *C. casasi* vs. 5.1 mm in both male and female *C. ceronorum*) and shorter limbs in males (mean LI 0.8 in *C. casasi* vs. 0.0 in *C. ceronorum*).

*Chiropterotriton casasi* differs from *C. perotensis* in its larger adult body size (mean SVL 37.8 mm in male and 40.9 mm in one female *C. casasi* vs. 29.7 mm in male and 31.7 mm in female *C. perotensis*), longer head (mean HL 8.3 mm in male and 8.6 mm in one female *C. casasi* vs. 6.6 mm in male and 6.7 mm in female *C. perotensis*), broader head (mean HW 5.8 mm in male and 5.9 mm in one female *C. casasi* vs. 4.2 mm in male and

4.4 mm in female *C. perotensis*), longer limbs (mean LI 0.8 in male and 1.0 in one female *C. casasi* vs. 2.5 in male and 3.3 in female *C. perotensis*), and larger feet (mean FW 3.7 mm in both male and one female *C. casasi* vs. 2.6 mm in both male and female *C. perotensis*).

*Chiropterotriton casasi* differs from *C. totonacus* in its larger adult body size (mean SVL 37.8 mm in male and 40.9 mm in one female *C. casasi* vs. 35.7 mm in male and 35.5 mm in female *C. totonacus*), shorter tail (mean TL/SVL 1.0 in male *C. casasi* vs. 1.16 in male *C. totonacus*; the only female specimen of *C. casasi* has a broken tail), longer head in females (mean HL 8.6 mm in one *C. casasi* vs. 7.6 mm in *C. totonacus*), broader head in females (mean HW 5.9 mm in one *C. casasi* vs. 5.2 mm in *C. totonacus*), shorter limbs (mean LI 0.8 in male and 1.0 in one female *C. casasi* vs. −0.6 in male and 0.0 in female *C. totonacus*), narrower feet (mean FW 3.7 mm in both male and one female *C. casasi* vs. 4.2 mm in male and 4.0 mm in female *C. totonacus*) and fewer maxillary teeth (mean MT 9.0 in male and 30 in one female *C. casasi* vs. 32.9 in male and 52.6 in female *C. totonacus*).

*Chiropterotriton casasi* differs from *C. melipona* in its larger adult body size (mean SVL 37.8 mm in male and 40.9 mm in one female *C. casasi* vs. 29.2 mm in male and 28.5 mm in female *C. melipona*), shorter tail (mean TL/SVL 1.04 in male *C. casasi* vs. 1.16 in male *C. melipona*; the only female specimen of *C. casasi* has a broken tail), longer head (mean HL 8.3 mm in male and 8.6 mm in one female *C. casasi* vs. 6.3 mm in male and 6.4 mm in female *C. melipona*), broader head (mean HW 5.8 mm in male and 5.9 mm in one female *C. casasi* vs. 4.3 mm in male and 4.2 mm in female *C. melipona*), longer limbs (mean LI 0.8 in male and 1.0 in one female *C. casasi* vs. 2.3 in male and 1.8 in female *C. melipona*) and broader feet (mean FW 3.7 mm in both male and one female *C. casasi* vs. 2.4 mm in male and 2.6 mm in female *C. melipona*).

*Chiropterotriton casasi* differs from *C. chiropterus* in its larger adult body size in females (mean SVL 40.9 mm in one *C. casasi* vs. 33.5 mm in *C. chiropterus*), shorter tail (mean TL/SVL 1.04 in male *C. casasi* vs. 1.25 in male *C. chiropterus*; the only female specimen of *C. casasi* has a broken tail), longer head (mean HL 8.3 mm in male and 8.6 mm in one female *C. casasi* vs. 8.1 mm in male and 7.3 mm in female *C. chiropterus*), narrower head (mean HW 5.8 mm in male and 5.9 mm in one female *C. casasi* vs. 5.6 mm in male and 4.8 mm in female *C. chiropterus*), shorter limbs in males (mean LI 0.8 in *C. casasi* vs. 0.3 in *C. chiropterus*) and fewer maxillary teeth (mean MT 9.0 in male and 30 in one female *C. casasi* vs. 12.6 in male and 48.0 in female *C. chiropterus*).

*Chiropterotriton casasi* differs from *C. orculus* in its larger adult body size (mean SVL 37.8 mm in male and 40.9 mm in one female *C. casasi* vs. 35.9 mm in male and 39.0 mm in female *C. orculus*), longer head (mean HL 8.3 mm in male and 8.6 mm in one female *C. casasi* vs. 7.4 mm in male and 8.0 mm in female *C. orculus*), broader head (mean HW 5.8 mm in male and 5.9 mm in one female *C. casasi* vs. 5.0 mm in male and 5.2 mm in female *C. orculus*) and longer limbs (mean LI 0.8 in male and 1.0 in one female *C. casasi* vs. 1.9 in male and 2.9 in female *C. orculus*).

*Chiropterotriton casasi* differs from *C. lavae* in its larger adult body size (mean SVL 37.8 mm in male and 40.9 mm in one female *C. casasi* vs. 32.4 mm in male and 31.6 mm in

female *C. lavae*), shorter tail in males (mean TL/SVL 1.04 in *C. casasi* vs. 1.19 in *C. lavae*), longer head (mean HL 8.3 mm in male and 8.6 mm in one female *C. casasi* vs. 7.5 mm in male and 7.0 mm in female *C. lavae*), broader head (mean HW 5.8 mm in male and 5.9 mm in one female *C. casasi* vs. 4.9 mm in male and 4.7 mm in female *C. lavae*), shorter limbs (mean LI 0.8 in male and 2.0 in one female *C. casasi* vs. −0.6 in male and 0.6 in female *C. lavae*) and more maxillary teeth in females (mean MT 30 in one *C. casasi* vs. 20.8 in *C. lavae*).

*Chiropterotriton casasi* differs from *C. aureus* in its larger adult body size (mean SVL 37.8 mm in male and 40.9 mm in one female *C. casasi* vs. 28.5 mm in one male and 26.8 mm in female *C. aureus*), shorter tail (mean TL/SVL 1.0 in male *C. casasi* vs. 1.28 in one male *C. aureus;* the only female specimen of *C. casasi* has a broken tail), longer head (mean HL 8.3 mm in male and 8.6 mm in one female *C. casasi* vs. 6.4 mm in one male and 6.0 mm in female *C. aureus*), broader head (mean HW 5.8 mm in male and 5.9 mm in one female *C. casasi* vs. 4.0 mm in one male and 3.6 mm in female *C. aureus*), longer limbs (mean LI 0.8 in male and 1.0 in one female *C. casasi* vs. 2.0 in one male and 2.3 in female *C. aureus*), and wider feet (mean FW 3.7 mm in both male and one female *C. casasi* vs. 2.4 mm in one male and 1.8 mm in female *C. aureus*).

*Chiropterotriton casasi* differs from *C. nubilus* in its larger adult body size (mean SVL 37.8 mm in male and 40.9 mm in one female *C. casasi* vs. 29.4 mm in one male and 30.5 mm in female *C. nubilus*), shorter tail (mean TL/SVL 1.0 in male *C. casasi* vs. 1.37 in one male *C. nubilus;* the only female specimen of *C. casasi* has a broken tail), longer head (mean HL 8.3 mm in male and 8.6 mm in one female *C. casasi* vs. 6.6 mm in one male and 7.4 mm in female *C. nubilus*), broader head (mean HW 5.8 mm in male and 5.9 mm in one female *C. casasi* vs. 4.0 mm in one male and 4.4 mm in female *C. nubilus*), longer limbs (mean LI 0.8 in male and 1.0 in one female *C. casasi* vs. 2.0 in one male and 1.5 in female *C. nubilus*), wider feet (mean FW 3.7 mm in both male and one female *C. casasi* vs. 2.6 mm in one male and 2.3 mm in female *C. nubilus*) and fewer maxillary teeth in females (mean MT 30.0 in female *C. casasi* vs. 41.5 in female *C. nubilus*).

**Description of holotype:** SVL 42.0 mm, TL 37.6 mm, AX 20.4 mm, SW 3.8 mm, HL 8.8 mm, HW 5.6 mm, HD 2.8 mm, projection of snout beyond mandible 0.2 mm, distance from anterior rim of orbit to snout 1.7 mm, interorbital distance 2.4 mm, eyelid length 2.1 mm, eyelid width 1.1 mm, horizontal orbit diameter 2.1 mm, nostril diameter 0.4 mm, FLL 10.7 mm, HLL 12.6 mm, snout-to-forelimb length 11.4 mm, distance from snout to anterior angle of vent 36.6 mm, tail width at base 3.3 mm, tail depth at base 3.9 mm, FW 4.0 mm, length of fifth toe 0.8 mm, length of longest (third) toe 1.2 mm, mental gland length 1.3 mm, mental gland width 1.3. Numbers of teeth: premaxillary 4, maxillary 4-4 (right-left) and vomerine 4-4 (right-left). Adpressed limbs are separated by 0 costal folds.

**Coloration in life:** No data.

**Coloration in preservative:** Faded brown, dorsally and laterally. No sign of dorsal stripe. Limbs mottled. Head is uniform pale brown with some mottling on the snout.

The paratypes present some variation. The entire body of MVZ 92876 is mottled with faded pale and dark brown. A pale band extends between the anterior part of the eyes; the snout is very mottled. Posteriorly, the body is strongly mottled; the anterior part of the tail has an irregularly bordered light dorsal stripe. MVZ 92875 is less boldly mottled but has some mottling. All of them have a paler venter than dorsum. MVZ 92877 also has a pale bar that extends between the eyes.

**Osteology:** This account is based on examination of a μCT scan of the anterior skeleton of MVZ 92874, an adult male, 42.0 mm SVL (Figs. 6–8; Table 3). The skull is robust and well developed. Notable features include the complete articulation of the paired frontals and parietals—there is no frontoparietal fontanel—and a robust premaxillary bone with paired ascending processes that broaden laterally as each approaches a solid articulation with the frontal bone on the same side. A distinct, albeit narrow palatal shelf is present on the premaxilla, and the two ascending processes enclose a distinct fontanel. Tiny paired septomaxillae bones are well separated from all other bones. The triangular nasal is weakly developed anteriorly, where the bone is very thin and has an irregular edge. It is partially overlapped laterally by the facial process of the maxilla. A large prefrontal articulates anteriorly with the nasal bone, its ventral portion is overlapped by the facial process of the maxilla, and it bears an ascending, pointed tab that overlaps the frontal extensively. The anteroventral margin of the prefrontal and the adjacent portion of the facial process of the maxilla are eroded by a foramen that allows passage of the nasolacrimal duct. Otherwise, the facial process of the maxillary bone is broad and robust and solidly articulated with adjacent bones. The maxillary bone resembles that in *Aneides* (*Wake, 1963*); the toothed portion is confined to the anterior 45–50% of the bone, whereas posteriorly the bone is cleaver-shaped with an extended posterior tip. There is a distinct, relatively broad palatal shelf on the lingual side. There are few maxillary teeth—five on each side—but they are large, sharp and recurved, with highly reduced anterior cusps. There are three premaxillary teeth. They appear unicuspid and sharp but are shorter than the maxillary teeth. The well-developed orbitosphenoid is solidly articulated with neighboring bones, forming a relatively stout braincase.

A prominent bony crest overlies the anterior semicircular canal dorsally. It is derived from the posterolateral portion of the parietal and the anteromedial portion of the otic capsule. An additional, crest-like spur emerges at right angles from this crest and is directed posterolaterally. A second, smaller crest similarly overlies the posterior semicircular canal. The parietal bears a very large and well-developed posterolateral tab that is sharply reflected ventromedially, extending nearly two-thirds down the vertical extent of the orbitosphenoid. The very robust squamosal articulates dorsally with the otic capsule opposite the lateral semicircular canal. As in other species, the shape of its curved anterior margin conforms closely with, but is nevertheless separate from, the lateral face of the otic capsule. The quadrate is small and inconspicuous, but appears to be well developed. There is a short, stout stylus on the columella, which otherwise is just a

rounded ossicle. Bodies of the paired vomers articulate tightly at the midline posterior to the internasal fontanel. Preorbital processes are long. There are four vomerine teeth on the right side and five on the left; one or two teeth are deployed at the base of each preorbital process. The unpaired parasphenoid bone is robust. It is narrow anteriorly but gradually widens posteriorly until very near the caudal end where it reaches its maximum width. Paired parasphenoid tooth patches are well separated from one another medially and from the vomerine teeth rostrally. There are approximately 95–100 teeth in each patch. The mandible is robust. The articular is fully ossified and appears to be fused to the prearticular bone. The height of the large coronoid process on the prearticular exceeds that of the dentary bone ventral to it. There are six sharply recurved and somewhat enlarged teeth on each dentary bone.

Only the distal portion of each forelimb is visible in the CT scan. Digital formulae are 1-2-3-2 on each side. Mesopodial cartilages are not mineralized.

**Distribution and ecology:** *Chiropterotriton casasi* is known only from the type locality. Vegetation at this locality now consists of secondary forest and thicket, but was likely cloud forest in the past. The species could occur somewhat more widely, but little intact forest remains in the vicinity of the type locality.

**Remarks:** The phylogenetic position of *Chiropterotriton casasi* relative to congeners is unknown due to the lack of tissue samples for genetic analyses. Geographically associated species include *C. chiropterus*, *C. melipona*, *C. perotensis*, *C. totonacus*, *C. ceronorum* and *C. lavae*. We have searched repeatedly in the vicinity of the type locality and have found another, unnamed, species of *Chiropterotriton*, but not this species.

**Conservation status:** *Chiropterotriton casasi* has not been seen since the original collection in 1969, and nearly all of the primary forest at the type locality has been cut down. Efforts to find this species at the type locality in recent years have not been successful. We recommend that it be designated as Critically Endangered based on criterion B1ab(iii) (extent of occurrence <100 km$^2$, distribution severely fragmented and known from only one locality, continuing decline in extent and quality of habitat; *IUCN, 2012*). Concerted efforts should be made to locate extant populations of this species in remaining habitat patches near the type locality.

**Etymology:** The species name honors Gustavo Casas Andreu, a Mexican herpetologist who has dedicated his career to describe the biodiversity of Mexican amphibians and reptiles.

## REDESCRIPTIONS

Original descriptions of *Chiropterotriton chiropterus* (*Cope, 1863*) and *C. orculus* (*Cope, 1865*) were extremely brief and contained relatively little information about the species' morphology. We provide more detailed redescriptions of both of these species, including the designation of a neotype for each. Common names declared for these species are from *Liner & Casas-Andreu (2008)*.

*Chiropterotriton chiropterus* Cope, 1863
Common Flat-footed Salamander, Salamandra de Pie Plano Común
Figures 4P–4R, 5F, 6H, 7H, 8H.

Chresonymy
*Chiropterotriton* sp. J.—Darda, 1994 (population 23, 24)
*Chiropterotriton* sp.—Wake, 1987; Papenfuss & Wake, 1987; Lynch & Wake, 1989; Wake, Papenfuss & Lynch, 1992

**Neotype:** MVZ 85590, an adult male from 1.4 mi southwest by road southwest edge of Huatusco de Chicuellar, Veracruz, Mexico, 19.141388°N, 96.98083°W (EPE = max. error distance 1.202 mi). The estimated elevation is 1,400 masl. Collected 16 January 1969 by R. W. McDiarmid and R.D. Worthington.

**Additional specimens examined:** Twelve specimens, all from 1.4 mi southwest by road southwest edge of Huatusco de Chicuellar, Veracruz, Mexico. Eight males: MVZ 85588–89, 85591–92, 85594, 85599, 85613, and 85602; and four females: MVZ 85597–98, 85605 and 85632.

**Diagnosis:** This is a medium-sized species of plethodontid salamander phylogenetically related to *C. melipona*; mean SVL 37.5 mm in eight adult males (range 36.1–38.8) and 33.5 mm in four adult females (range 30.7–36.7). The head is moderately wide; HW averages 15% of SVL in both males and females (range 14–16). The snout is broad and bluntly rounded in males. Jaw muscles are relatively pronounced. Eyes are moderately protuberant and extend laterally beyond the jaw margin in ventral view. There are few maxillary teeth in males (mean MT 12.6, range 9–17) but many in females (mean MT 48.0, range 42–57). There are few vomerine teeth in both males (mean VT 10.6, range 9–12) and females (mean VT 12.5, 10–15), which are arranged in a row that does not reach or barely reaches the inner margin of the internal choana. The tail is long and slender and exceeds SVL by a considerable amount in nearly all specimens; mean TL equals 1.25 of SVL in males (range 1.13–1.38) and 1.19 in females (1.01–1.26). Limbs are short to moderate length; FLL + HLL averages 52% of SVL in males (range 48–54%) and 50% in females (range 47–53%). Adpressed limbs closely approach or overlap slightly in males (mean LI 0.3, range −0.5 to 1) but are more widely separated in females (mean LI 2.0, range 1.5–2.5). Manus and pes are relatively small, digits are slender. Subterminal pads are small but well developed. Digital webbing ranges from slight to absent and is limited to the metatarsal region. The first digit is distinct but largely included in the webbing. Digital tips are only slightly expanded. The mental gland is oval-shaped and not especially prominent in males. The smallest mature male (pigmented testes) is 36.1 mm SVL; the smallest male with a mental gland is 33.3 mm SVL. Parotoid glands are not evident.

**Comparisons:** *Chiropterotriton chiropterus* differs from *C. ceronorum* in its larger adult body size in males (mean SVL 37.5 mm in *C. chiropterus* vs. 33.9 mm in *C. ceronorum*), longer tail (mean TL/SVL 1.25 in male and 1.19 in female *C. chiropterus* vs. 1.0 in male and 0.97 in female *C. ceronorum*), shorter limbs (mean LI 0.3 in male and 2.0 in female

*C. chiropterus* vs. 0.0 in male and 1.5 in female *C. ceronorum*), longer head (mean HL 8.1 mm in male and 7.3 mm in female *C. chiropterus* vs. 7.5 mm in male and 7.1 mm in female *C. ceronorum*), broader head in males (mean HW 5.6 mm in *C. chiropterus* vs. 5.1 mm in *C. ceronorum*) and fewer vomerine teeth (mean VT 10.6 in male and 12.5 in female *C. chiropterus* vs. 13.0 in male and 15.9 in female *C. ceronorum*).

*Chiropterotriton chiropterus* differs from *C. perotensis* in its larger adult body size (mean SVL 37.5 mm in male and 33.5 mm in female *C. chiropterus* vs. 29.7 mm in male and 31.7 mm in female *C. perotensis*), longer tail (mean TL/SVL 1.25 in male and 1.19 in female *C. chiropterus* vs. 1.0 in both male and female *C. perotensis*), longer limbs (mean LI 0.3 in male and 2.0 in female *C. chiropterus* vs. 2.5 in male and 3.3 in female *C. perotensis*), longer head (mean HL 8.1 mm in male and 7.3 mm in female *C. chiropterus* vs. 6.6 mm in male and 6.7 mm in female *C. perotensis*), broader head (mean HW 5.6 mm in male and 4.8 mm in female *C. chiropterus* vs. 4.2 mm in male and 4.4 mm in female *C. perotensis*), broader feet (mean FW 3.7 mm in male and 3.1 mm in female *C. chiropterus* vs. 2.6 mm in both male and female *C. perotensis*), fewer maxillary teeth (mean MT 12.6 in male and 48.0 in female *C. chiropterus* vs. 7.2 in male and 27.9 in female *C. perotensis*) and more vomerine teeth (mean VT 10.6 in male and 12.5 in female *C. chiropterus* vs. 9.0 in male and 11.1 in female *C. perotensis*).

*Chiropterotriton chiropterus* differs from *C. totonacus* in its larger adult body size in males (mean SVL 37.5 mm in *C. chiropterus* vs. 35.7 mm in *C. totonacus*), longer tail (mean TL/SVL 1.25 in male and 1.19 in female *C. chiropterus* vs. 1.16 in male and 1.20 in female *C. totonacus*), shorter limbs (mean LI 0.3 in male and 2.0 in female *C. chiropterus* vs. −0.60 in male and 0.0 in female *C. totonacus*), shorter head (mean HL 8.1 mm in male and 7.3 mm in female *C. chiropterus* vs. 8.5 mm in male and 7.6 mm in female C. *totonacus*), narrower feet in males (mean FW 3.7 mm in *C. chiropterus* vs. 4.2 mm in *C. totonacus*), fewer maxillary teeth (mean MT 12.6 in male and 48.0 in female *C. chiropterus* vs. 32.9 in male and 52.6 in female *C. totonacus* ) and fewer vomerine teeth (mean VT 10.6 in male and 12.5 in female *C. chiropterus* vs. 11.6 in male and 13.7 in female *C. totonacus*).

*Chiropterotriton chiropterus* differs from *C. melipona* in its larger adult body size (mean SVL 37.5 mm in male and 33.5 mm in female *C. chiropterus* vs. 29.2 mm in male and 28.5 mm in female *C. melipona*), longer tail (mean TL/SVL 1.25 in male and 1.19 in female *C. chiropterus* vs. 1.16 in male and 1.11 in female *C. melipona*), longer head (mean HL 8.1 mm in male and 7.3 mm in female *C. chiropterus* vs. 6.3 mm in male and 6.4 mm in female *C. melipona*), wider head (mean HW 5.6 mm in male and 4.8 mm in female *C. chiropterus* vs. 4.3 mm in male and 4.2 mm in female *C. melipona*), longer limbs in males (mean LI 0.3 in *C. chiropterus* vs. 2.3 in *C. melipona*), wider feet (mean FW 3.7 mm in male and 3.1 mm in female *C. chiropterus* vs. 2.4 mm in male and 2.6 mm in female *C. melipona*) and more maxillary teeth (mean MT 12.6 in male and 48.0 in female *C. chiropterus* vs. 9.5 in male and 31.0 in female *C. melipona*).

*Chiropterotriton chiropterus* differs from *C. casasi* in its smaller adult body size in females (mean SVL 33.5 mm in *C. chiropterus* vs. 40.9 mm in one *C. casasi*), longer tail in males (mean TL/SVL 1.25 in *C. chiropterus* vs. 1.04 in *C. casasi*), shorter head (mean HL

8.1 mm in male and 7.3 mm in female *C. chiropterus* vs. 8.3 mm in male and 8.6 mm in one female *C. casasi*), broader head (mean HW 5.6 mm in male and 4.8 mm in female *C. chiropterus* vs. 5.8 mm in male and 5.9 mm in one female *C. casasi*), longer limbs in males (mean LI 0.3 in *C. chiropterus* vs. 0.8 in *C. casasi*) and more maxillary teeth (mean MT 12.6 in male and 48.0 in female *C. chiropterus* vs. 9.0 in male and 30 in one female *C. casasi*).

*Chiropterotriton chiropterus* differs from *C. orculus* in its longer tail (mean TL/SVL 1.25 in male and 1.19 in female *C. chiropterus* vs. 1.02 in both male and female *C. orculus*), longer head in males (mean HL 8.1 mm in *C. chiropterus* vs. 7.4 mm in *C. orculus*), wider head in males (mean HW 5.6 mm in *C. chiropterus* vs. 5.0 mm in *C. orculus*), longer limbs (mean LI 0.3 in male and 2.0 in female *C. chiropterus* vs. 1.9 in male and 2.9 in female *C. orculus*), wider feet in males (mean FW 3.7 mm in *C. chiropterus* vs. 3.2 mm in *C. orculus*) and more maxillary teeth (mean MT 12.6 in male and 48.0 in female *C. chiropterus* vs. 8.2 in male and 28.8 in female *C. orculus*).

*Chiropterotriton chiropterus* differs from *C. lavae* in its larger adult body size (mean SVL 37.5 mm in male and 33.5 mm in female *C. chiropterus* vs. 32.4 mm in male and 31.6 mm in female *C. lavae*), longer tail (mean TL/SVL 1.25 in male and 1.19 in female *C. chiropterus* vs. 1.19 in male and 1.02 in female *C. lavae*), shorter limbs (mean LI 0.3 in male and 2.0 in female *C. chiropterus* vs. −0.6 in male and 0.6 in female *C. lavae*), longer head (mean HL 8.1 mm in male and 7.3 mm in female *C. chiropterus* vs. 7.5 mm in male and 7.0 mm in female *C. lavae*), broader head (mean HW 5.6 mm in male and 4.8 mm in female *C. chiropterus* vs. 4.9 mm in male and 4.7 mm in female *C. lavae*), more maxillary teeth (mean MT 12.6 in male and 48.0 in female *C. chiropterus* vs. 7.0 in male and 20.8 in female *C. lavae*) and more vomerine teeth (mean VT 10.6 in male and 12.5 in female *C. chiropterus* vs. 8.9 in male and 11.4 in female *C. lavae*).

*Chiropterotriton chiropterus* differs from *C. aureus* in its larger adult body size (mean SVL 37.5 mm in male and 33.5 mm in female *C. chiropterus* vs. 28.5 mm in one male, mean 26.8 mm in females *C. aureus*), relatively longer limbs in males (mean LI 0.3 in male *C. chiropterus* vs. 2.0 in one male *C. aureus*), longer head (mean HL 8.1 mm in male and 7.3 mm in female *C. chiropterus* vs. 6.4 mm in one male, mean 6.0 mm in female *C. aureus*), broader head (mean HW 5.6 mm in male and 4.8 mm in female *C. chiropterus* vs. 4.0 mm in one male, 3.6 mm in female *C. aureus*), and larger feet (mean FW 3.7 mm in male and 3.1 mm in female *C. chiropterus* vs. 2.4 mm in one male, 1.8 mm in females of *C. aureus*).

*Chiropterotriton chiropterus* differs from *C. nubilus* in its larger adult body size (mean SVL 37.5 mm in male and 33.5 mm in female *C. chiropterus* vs. 29.4 mm in one male, mean 30.5 mm in females *C. nubilus*), relatively longer limbs in males (mean LI 0.3 in male *C. chiropterus* vs. 2.0 in one male *C. nubilus*), longer head in males (mean HL 8.1 mm in *C. chiropterus* vs. 6.6 mm in one male *C. nubilus*), broader head (mean HW 5.6 mm in male and 4.8 mm in female *C. chiropterus* vs. 4.0 mm in one male, 4.4 mm in female *C. nubilus*), and larger feet (mean FW 3.7 mm in male and 3.1 mm in female *C. chiropterus* vs. 2.6 mm in one male and 2.3 mm in females of *C. nubilus*).

**Description of neotype:** SVL 38.8 mm, TL 46.0 mm, AX 20.8 mm, SW 4.1 mm, HL 8.0 mm, HW 5.4 mm, HD 2.8 mm, projection of snout beyond mandible 0.4 mm, distance from anterior rim of orbit to snout 1.8 mm, interorbital distance 2.4 mm, eyelid length 2.7 mm, eyelid width 1.2 mm, horizontal orbit diameter 1.7 mm, nostril diameter 0.4 mm, FLL 9.5 mm, HLL 10.8 mm, snout-to-forelimb length 10.2 mm, distance from snout to anterior angle of vent 36.7 mm, tail width at base 2.8 mm, tail depth at base 2.7 mm, FW 3.6 mm, length of fifth toe 0.9 mm, length of longest (third) toe 1.5 mm, mental gland length 1.3 mm, mental gland width 1.3. Numbers of teeth: premaxillary 5, maxillary 6-10 (right-left) and vomerine 5-5 (right-left). Adpressed limbs are separated by 0 costal folds.

**Coloration in life:** Data have been derived from diapositives of seven specimens from Huatusco taken by Roy W. McDiarmid. This is a generally brightly colored species in which yellowish colors predominate. It is generally pale laterally and ventrally. A dorsal light band is generally present that extends onto the tail, sometimes to the tip, but there are some darker specimens that lack an obvious stripe. Coloration varies extensively from one specimen to the next with respect to the nature of the dorsal band and its coloration. In one large adult, the color is a relatively intense Orange Rufous (5) at the origin of the band, behind the eyes, but it becomes lighter and yellower posteriorly and on the sides of the head and neck, from Tawny Olive (17) to Pale Horn Color (11), then Yellow Ochre (14). Over the shoulder and more posteriorly, yellowish-to-cream spots (Light Buff, 2) form in a dorsolateral ragged line, with the dorsomedial stripe becoming Light Neutral Gray (297) grading into Pale Neutral Gray (296) and extending onto the tail as Pale Mauve (204) with speckles of Cinnamon Drab (50). The limbs are yellowish (Chamois, 84). The iris is dark ventrally but has a yellow-gold highlight. The dorsal eyelid is pale and colorless. A faint light cream bar extends between the eyes.

Another specimen is more colorful dorsally. The head is complexly colored with a bright snout (Salmon Color, 82) to the midpoint between the eyes. A dark bar extends between the eyes, beginning on the eyelid, and an inverted triangular dark area extends posteriorly to the anterior boundary of the epaxial muscles. The temporal region of the head back over the shoulders is light in coloration (Chamois, 84) and there is a lateral excursion of the color over the shoulder region. The base of the tail becomes brighter and rich reddish brown (Carmine, 64). The limbs are a bright mottling of gray and yellow (Cream Yellow, 82, to Chamois 84).

Some animals are darker than the above but most have a light, bright dorsal coloration in the tan-to-yellow range with some brighter orange on the snout. In some the dorsal coloration is pale to very pale. There is usually a bar between the eyes and a ventrolateral excursion of the dorsal band in front of and over the shoulders.

**Coloration in preservative:** The dorsum is a relatively pale brown, either uniform or with an indistinct, broad brown dorsal stripe bordered by thin, darker-brown dorsolateral lines that extend from the nape to the base of the tail. The dorsal surface of the tail is a relatively pale brown with some darker mottling; the head sometimes has a small amount

of darker mottling. The venter and gular region are a uniform pale tan; the ventral side of the tail is a uniform, slightly darker brown.

**Osteology:** This account is based primarily on examination of a μCT scan of the skull of MVZ 85602, an adult male, 38.9 mm SVL (Figs. 6–8; Table 3). In addition, four cleared-and-stained specimens were scored for osteological characters evaluated by *Darda & Wake (2015)*. The skull is well developed. The cranial roof is complete: paired frontals and parietals articulate across the midline—there is no frontoparietal fontanel— although tabs that extend posteriorly from the frontals to overlap the parietals, which are present in some congeners, are absent. Rostral bones articulate firmly with one another, including many overlapping articulations, such as the prefrontal and nasal by the maxilla. Ascending processes of the single premaxilla are separate along their entire length and broaden laterally as they approach their articulation with the frontals. A very small septomaxilla is present on each side. The nasal is large, including an anteromedial protrusion that forms a medial wall to the external naris and nearly contacts the premaxilla at its rostral articulation with the maxilla. The prefrontal is robust; dorsally, it overlaps the frontal bone whereas ventrally it is overlapped by the facial process of the maxilla. The foramen for the nasolacrimal duct has eroded abutting portions of the facial process of the maxilla, the nasal and the prefrontal. The five teeth on the left maxilla and six on the right are confined to the anterior 50% of each bone. The remaining (edentulous) portion of each maxilla is cleaver-like. There are four premaxillary teeth. The orbitosphenoid is fully articulated with the frontal and parietal dorsally and the parasphenoid ventrally, thus forming a solid braincase.

There are two large (elevated) crests on each otic capsule. One arises dorsal to the anterior semicircular canal. The other emerges at right angles from the midpoint of the first crest and extends posterolaterally towards the lateral semicircular canal. A moderately sized tab emerges from the posterolateral edge of the parietal and is sharply reflected ventromedially, extending at least halfway down the vertical extent of the orbitosphenoid. The squamosal, while typical for *Chiropterotriton*, bears a distinctive longitudinal ridge on its lateral face. The quadrate, while robust, is nevertheless small and inconspicuous. The columellar stylus is well developed for *Chiropterotriton*; it comprises a short but distinct rod that is directed towards but does not contact the squamosal. Paired vomers articulate medially both anteriorly and posteriorly, partially obliterating the internasal fontanel in ventral view. Preorbital processes of the vomer are spine-like—elongate and pointed—and completely lack teeth. There are five vomerine teeth on the right side and six on the left. The parasphenoid bone is relatively narrow posteriorly. Paired parasphenoid tooth patches are separated across midline; each bears 45–50 teeth. The mandible is robust. The articular bone is robust and solidly articulated with the prearticular and the dentary. The prearticular is well developed; the coronoid process is very high. There are 10 teeth on the right dentary bone and 11 on the left.

Digital formulae are 1-2-3-2 on each side. The distal tip of the terminal phalanx is greatly expanded on each finger except the first. Mesopodial cartilages are not mineralized.

**Distribution and ecology:** *Chiropterotriton chiropterus* is found from the vicinity of the type locality near Huatusco, Veracruz, south to the Sierra de Juárez, Oaxaca. Geographically associated species include *C. orculus*, *C. perotensis*, *C. ceronorum* and *C. lavae*. The species occurs at higher elevations in Oaxaca than in Veracruz and the overall elevational range is from 1,400 to 2,170 masl.

**Remarks:** Populations from the Sierra de Juárez, Oaxaca, were previously considered to represent an undescribed species (*Chiropterotriton* sp. J) based on allozyme data (*Darda, 1994*), but that study lacked specimens of topotypic *C. chiropterus*. Mitochondrial DNA sequenced data showed that *Chiropterotriton* sp. J is most closely related to *C. chiropterus*. Based on examination of a series of specimens from the north slope of Cerro Pelón, Oaxaca, we are unable to find any discrete morphological differences between these populations that would support the recognition of *C.* sp. J as a distinct species. We therefore assign populations from Oaxaca previously referred to *Chiropterotriton* sp. J to *C. chiropterus*. With the assignment of the Oaxacan populations to this species, it now has by far the widest geographic range of any species of the genus, approximately 200 km in direct linear distance.

**Conservation status:** *Chiropterotrition chiropterus* is designated as Critically Endangered by the most recent IUCN Red List of Threatened Species (*Parra-Olea, Wake & Hanken, 2008*).

*Chiropterotriton orculus Cope, 1865*
Cope's Flat-footed Salamander, Salamandra de Pie Plano de Cope
Figures 4S–4U, 5G, 6E, 7E, 8E.

Chresonymy
***Spelerpes orculus***—Cope, 1865: 196. Syntypes: USNM or ANSP, not now present in either collection. Type locality: "Mexican Table Land" (*Frost, 2019*).
***Spelerpes chiropterus*** (part)—*Cope, 1869*: 106; *Taylor & Smith, 1945*; *Smith & Taylor, 1948*.
***Chiropterotriton orculus***—*Darda, 1994*; *Raffaëlli, 2007*; *Raffaëlli, 2013*.

**Neotype:** MVZ 138783, an adult male from the ridge between Popocatepetl and Iztaccihuatl, along Mexican Hwy. 196, 16.2 km by road east jct Mexican Hwy. 115, Mexico, Mexico, 3,300 masl, 19.0973°N, 98.6829°W. Collected 26 July 1976 by J.F. Lynch, D.B. Wake and M.E. Feder.

**Additional specimens examined:** Nineteen specimens, all from the ridge between Popocatepetl and Iztaccihuatl, México, Mexico. Nine males: MVZ 76161, 138694, 138696–97, 138700, 138778, 138784, 138804 and 200630; and ten females: MVZ 138686, 138688, 138776–77, 138779, 138781, 138793, 138796–97 and 200629.

**Diagnosis:** This is a medium-sized species of *Chiropterotriton*; mean SVL 35.9 mm in ten adult males (range 33.6–38.9) and 39.0 mm in ten adult females (range 34.9–43.0). The head is moderately wide; HW averages 14% of SVL in males (range 13–15) and 13% in

females (range 12–14). Jaw muscles are prominent in both males and females. Adult males have a broad, bluntly rounded snout with broad and moderately developed nasolabial protuberances. Eyes are large and relatively prominent and extend slightly beyond the jaw margin in ventral view. There are few maxillary teeth in males (mean MT 8.2, range 5–11) and moderate numbers in females (mean MT 28.8, range 23–35). There are few vomerine teeth in both males (mean VT 8.6, 5–11) and females (mean VT 12.0, range 9–15), which are arranged in a curved row that does not extend lateral to the outer margin of the internal choana. The tail is moderately long and slightly exceeds snout-vent length in most specimens; mean TL/SVL equals 1.02 in both males (range 0.86–1.15) and females (range 0.87–1.12). Limbs are short to moderately long in both females and males; FLL + HLL averages 51% of SVL in males (range 43–56) and 47% in females (range 44–50). Adpressed limbs approach closely in males (mean LI 1.9, range 0.0–3.0) but are widely separated in females (mean LI 2.9, range 2.0–3.0). The manus and pes are relatively small, digits are broad. Subterminal pads are well developed. Digital webbing ranges from slight to moderate, extending to the base of the penultimate phalanx on the third toe. The first digit is distinct but barely emerges from the webbing. Digital tips are only slightly expanded. The mental gland is prominent, relatively large and oval (nearly round) in males. The smallest mature male is 33.6 mm SVL.

**Comparisons:** *Chiropterotriton orculus* differs from *C. ceronorum* in its larger adult body size (mean SVL 35.9 mm in male and 39.0 mm in female *C. orculus* vs. 33.9 mm in male and 34.9 mm in female *C. ceronorum*), shorter limbs (mean LI 1.9 in male and 2.9 in female *C. orculus* vs. 0.0 in male and 1.5 in female *C. ceronorum*), fewer maxillary teeth (mean MT 8.2 in male and 28.8 in female *C. orculus* vs. 11.0 in male and 47.7 in female *C. ceronorum*) and fewer vomerine teeth (mean VT 8.6 in male and 12.0 in female *C. orculus* vs. 13.0 in male and 15.9 in female *C. ceronorum*).

*Chiropterotriton orculus* differs from *C. perotensis* in its larger adult body size (mean SVL 35.9 mm in male and 39.0 mm in female *C. orculus* vs. 29.7 mm in male and 31.7 mm in female *C. perotensis*), slightly longer limbs (mean LI 1.9 in male and 2.9 in female *C. orculus* vs. 2.5 in male and 3.3 in female *C. perotensis*), longer head (mean HL 7.4 mm in male and 8.0 mm in female *C. orculus* vs. 6.6 mm in male and 6.7 mm in female *C. perotensis*), broader head (mean HW 5.0 mm in male and 5.2 mm in female *C. orculus* vs. 4.2 mm in male and 4.4 mm in female *C. perotensis*), larger feet (mean FW 3.2 mm in male and 3.4 mm in female *C. orculus* vs. 2.6 mm in both male and female *C. perotensis*) and more maxillary teeth (mean MT 8.2 in male and 28.8 in female *C. orculus* vs. 7.2 in male and 27.9 in female *C. perotensis*).

*Chiropterotriton orculus* differs from *C. totonacus* in its larger adult body size in females (mean SVL 39.0 mm in *C. orculus* vs. 35.5 mm in *C. totonacus*), shorter tail (mean TL/SVL 1.02 in both male and female *C. orculus* vs. 1.16 in male and 1.20 in female *C. totonacus*), shorter limbs (mean LI 1.9 in male and 2.9 in female *C. orculus* vs. −0.60 in male and 0.0 in female *C. totonacus*), shorter head in males (mean HL 7.4 mm in *C. orculus* vs. 8.5 mm in C. *totonacus*), narrower feet (mean FW 3.2 mm in male and 3.4 mm in female *C. orculus* vs. 4.2 mm in male and 4.0 mm in female *C. totonacus*), fewer maxillary

teeth (mean MT 8.2 in male and 28.8 in female *C. orculus* vs. 32.9 in male and 52.6 in female *C. totonacus*) and fewer vomerine teeth (mean VT 8.6 in male and 12.0 in female *C. orculus* vs. 11.6 in male and 13.7 in female *C. totonacus*).

*Chiropterotriton orculus* differs from *C. melipona* in its larger adult body size (mean SVL 35.9 mm in male and 39.0 mm in female *C. orculus* vs. 29.2 mm in male and 28.5 mm in female *C. melipona*), shorter tail (mean TL/SVL 1.02 in both male and female *C. orculus* vs. 1.16 in male and 1.11 in female *C. melipona*), longer head (mean HL 7.4 mm in male and 8.0 mm in female *C. orculus* vs. 6.3 mm in male and 6.4 mm in female *C. melipona*), broader head (mean HW 5.0 mm in male and 5.2 mm in female *C. orculus* vs. 4.3 mm in male and 4.2 mm in female *C. melipona*) and broader feet (mean FW 3.2 mm in male and 3.4 mm in female *C. orculus* vs. 2.4 mm in male and 2.6 mm in female *C. melipona*).

*Chiropterotriton orculus* differs from *C. casasi* in its smaller adult body size (mean SVL 35.9 mm in male and 39.0 mm in female *C. orculus* vs. 37.8 mm in male and 40.9 mm in one female *C. casasi*), shorter head (mean HL 7.4 mm in male and 8.0 mm in female *C. orculus* vs. 8.3 mm in male and 8.6 mm in one female *C. casasi*), narrower head (mean HW 5.0 mm in male and 5.2 mm in female *C. orculus* vs. 5.8 mm in male and 5.9 mm in one female *C. casasi*) and shorter limbs (mean LI 1.9 in male and 2.9 in female *C. orculus* vs. 0.8 in male and 1.0 in one female *C. casasi*).

*Chiropterotriton orculus* differs from *C. chiropterus* in its shorter tail (mean TL/SVL 1.02 in both male and female *C. orculus* vs. 1.25 in male and 1.19 in female *C. chiropterus*), shorter head in males (mean HL 7.4 mm in *C. orculus* vs. 8.1 mm in *C. chiropterus*), narrower head in males (mean HW 5.0 mm in *C. orculus* vs. 5.6 mm in *C. chiropterus*), shorter limbs (mean LI 1.9 in male and 2.9 in female *C. orculus* vs. 0.3 in male and 2.0 in female *C. chiropterus*), narrower feet in males (mean FW 3.2 mm in *C. orculus* vs. 3.7 mm in *C. chiropterus*) and fewer maxillary teeth (mean MT 8.2 in male and 28.8 in female *C. orculus* vs. 12.6 in male and 48.0 in female *C. chiropterus*).

*Chiropterotriton orculus* differs from *C. lavae* in its larger adult body size (mean SVL 35.9 mm in male and 39.0 mm in female *C. orculus* vs. 32.4 mm in male and 31.6 mm in female *C. lavae*), shorter tail in males (mean TL/SVL 1.02 in *C. orculus* vs. 1.19 in *C. lavae*), shorter limbs (mean LI 1.9 in male and 2.9 in female *C. orculus* vs. −0.60 in male and 0.6 in female *C. lavae*) and more maxillary teeth (mean MT 8.2 in male and 28.8 in female *C. orculus* vs. 7.0 in male and 20.8 in female *C. lavae*).

*Chiropterotriton orculus* differs from *C. dimidiatus* in its larger adult body size (mean SVL 35.9 mm in male and 39.0 mm in female *C. orculus* vs. 24.6 mm in male and 25.8 mm in female *C. dimidiatus*), longer tail (mean TL/SVL 1.02 in both male and female *C. orculus* vs. 0.89 in male and 0.87 in female *C. dimidiatus*), longer limbs (mean LI 1.90 in male and 2.90 in female *C. orculus* vs. 3.8 in male and 4.9 in female *C. dimidiatus*), longer head (mean HL 7.4 mm in male and 8.0 mm in female *C. orculus* vs. 5.2 mm in male and 5.0 mm in female *C. dimidiatus*), broader head (mean HW 5.0 mm in male and 5.2 mm in female *C. orculus* vs. 3.4 mm in male and 3.5 mm in female *C. dimidiatus*), broader feet (mean FW 3.2 mm in male and 3.4 mm in female *C. orculus* vs. 1.7 mm in both male and female *C. dimidiatus*), more maxillary teeth (mean MT 8.2 in male and 28.8

in female *C. orculus* vs. 3.8 in male and 17.0 in female *C. dimidiatus*) and more vomerine teeth (mean VT 8.6 in male and 12.0 in female *C. orculus* vs. 5.6 in male and 8.3 in female *C. dimidiatus*).

*Chiropterotriton orculus* differs from *C. chico* in its smaller adult body size in males (mean SVL 35.9 mm in *C. orculus* vs. 38.4 mm in *C. chico*), shorter tail (mean TL/SVL 1.02 in both male and female *C. orculus* vs. 1.18 in male and 1.12 in female *C. chico*), shorter limbs (mean LI 1.90 in male and 2.90 in female *C. orculus* vs. 0.6 in male and 2.1 in female *C. chico*), shorter head (mean HL 7.4 mm in male and 8.0 mm in female *C. orculus* vs. 8.8 mm in male and 8.7 mm in female *C. chico*), narrower head (mean HW 5.0 mm in male and 5.2 mm in female *C. orculus* vs. 5.6 mm in male and 5.7 mm in female *C. chico*), narrower feet (mean FW 3.2 mm in male and 3.4 mm in female *C. orculus* vs. 4.1 mm in male and 4.2 mm in female *C. chico*) and fewer vomerine teeth (mean VT 8.6 in male and 12.0 in female *C. orculus* vs. 13.6 in male and 15.6 in female *C. chico*).

*Chiropterotriton orculus* differs from *C. arboreus* in its larger adult body size (mean SVL 35.9 mm in male and 39.0 mm in female *C. orculus* vs. 33.4 mm in male and 32.2 mm in female *C. arboreus*), longer tail (mean TL/SVL 1.02 in both male and female *C. orculus* vs. 0.83 in male and 0.87 in female *C. arboreus*) and shorter limbs (mean LI 1.90 in male and 2.90 in female *C. orculus* vs. 0.20 in male and 1.0 in female *C. arboreus*).

*Chiropterotriton orculus* differs from *C. terrestris* in its larger adult body size (mean SVL 35.9 mm in male and 39.0 mm in female *C. orculus* vs. 24.2 mm in male and 23.0 mm in female *C. terrestris*), longer head (mean HL 7.4 mm in male and 8.0 mm in female *C. orculus* vs. 5.7 mm in male and 5.2 mm in female *C. terrestris*), broader head (mean HW 5.0 mm in male and 5.2 mm in female *C. orculus* vs. 3.5 mm in male and 3.3 mm in female *C. terrestris*) and broader feet (mean FW 3.2 mm in male and 3.4 mm in female *C. orculus* vs. 1.9 mm in male and 1.7 mm in female *C. terrestris*).

*Chiropterotriton orculus* differs from *C. aureus* by being larger (mean SVL 35.9 mm in male and 39.0 mm in female *C. orculus* vs. 28.5 mm in one male, mean 26.8 mm in female *C. aureus*), with a shorter tail (mean TL/SVL 1.02 in both male and female *C. orculus* vs. 1.28 in one male, mean 1.16 in female *C. aureus*), relatively shorter limbs in females (mean LI 2.9 in female *C. orculus* vs. 2.3 in female *C. aureus*), larger head (mean HL 7.4 mm in male and 8.0 mm in female *C. orculus* vs. 6.4 mm in one male, mean 6.0 mm in female *C. aureus*), broader head (mean HW 5.0 mm in male and 5.2 mm in female *C. orculus* vs. 4.0 mm in one male, 3.6 mm in female *C. aureus*), and broader feet (mean FW 3.2 mm in male and 3.4 mm in female *C. orculus* vs. 2.4 mm in one male, mean 1.8 mm in female *C. aureus*).

*Chiropterotriton orculus* differs from *C. nubilus* in being larger (mean SVL 35.9 mm in male and 39.0 mm in female *C. orculus* vs. 29.4 mm in one male, mean 30.5 mm in female *C. nubilus*), with a shorter tail (mean TL/SVL 1.02 in both male and female *C. orculus* vs. 1.37 in one male, mean 1.12 in female *C. nubilus*), relatively shorter limbs in females (mean LI 2.9 in female *C. orculus* vs. 1.5 in female *C. nubilus*), longer head (mean HL 7.4 mm in male and 8.0 mm in female *C. orculus* vs. 6.6 mm in one male, mean 7.4 mm in female *C. nubilus*), broader head (mean HW 5.0 mm in male and 5.2 mm in female *C. orculus* vs. 4.0 mm in one male, mean 4.4 mm in female *C. nubilus*), and broader feet

(mean FW 3.2 mm in male and 3.4 mm in female *C. orculus* vs. 2.6 mm in one male, mean 2.3 mm in female *C. nubilus*).

**Description of neotype:** SVL 38.9 mm, TL 33.6 mm, AX 20.5 mm, SW 4.0 mm, HL 8.1 mm, HW 5.5 mm, HD 2.4 mm, projection of snout beyond mandible 0.6 mm, distance from anterior rim of orbit to snout 2.3 mm, interorbital distance 2.3 mm, eyelid length 3.5 mm, eyelid width 1.6 mm, horizontal orbit diameter 1.8 mm, nostril diameter 0.3 mm, FLL 9.3 mm, HLL 9.6 mm, snout-to-forelimb length 9.5 mm, distance from snout to anterior angle of vent 33.8 mm, tail width at base 3.1 mm, tail depth at base 3.2 mm, FW 3.5 mm, length of fifth toe 0.5 mm, length of longest (third) toe 1.2 mm, mental gland length 1.3 mm, mental gland width 1.3. Numbers of teeth: premaxillary 4, maxillary 4-5 (right-left) and vomerine 5-4 (right-left). Adpressed limbs are separated by 2 costal folds.

**Coloration in life:** No information is available for the neotype or topotypic individuals; this description is based on photos of specimens from Lagunas de Zempoala. The background dorsal color is very dark gray. A broad dorsal band is typically present, varying in color from reddish brown to tan or nearly golden brown; the background color is visible only along midline. This coloration continues onto the tail, although the band is less regular and somewhat broken up in many individuals. The head is very dark brown, with splotches of brown similar in coloration to those on the dorsum. Small, pale-gray specks often present on both head and tail. The dorsal band is bordered by very dark gray. Some individuals lack a dorsal band and are very dark-brownish-grey dorsally with pale flecks throughout. Flanks are dark gray with pale gray specks, which are numerous on the body with some on the sides of the head and few to none on the sides of the tail. Upper side of limbs either similar in coloration to flanks or slightly paler. The iris is coppery.

**Coloration in preservative:** The dorsum, head and tail are a uniform medium brown. The upper side of the limbs and feet are paler brown. The venter, gular region and underside of the forelimbs are tan to pale brown; the underside of the hind limbs and tail are slightly darker brown.

**Osteology:** This account is based on examination of a μCT scan of the anterior skeleton of the neotype: MVZ 138783, an adult male, 38.9 mm SL (Figs. 6–8; Table 3). The skull is compact and robust, especially anteriorly. The snout is blunt in lateral view. Cranial roofing bones are moderately well ossified. Paired frontals articulate across the midline anteriorly for about two thirds of their length but then separate to participate in a relatively large frontoparietal fontanel, which includes about three fourths of the length of the parietals. Posteriorly extending tabs of the frontals overlap the parietals anteriorly. Ascending processes of the single premaxilla approach one another medially but remain separate for their entire length. They twist and broaden greatly as they ascend before establishing a firm articulation with the frontal. The dental process of the premaxilla is deep (high) but no palatal shelf is evident. Septomaxillae are present on both sides; they are

very small but nevertheless well developed for *Chiropterotriton*. The nasal bone is broadly triangular, but also thin and less well-developed anteromedially. It barely abuts the premaxilla medially and the maxilla laterally; is separated from the prefrontal posterolaterally; and slightly overlaps the frontal posteriorly. The prefrontal is broad, compact and almost quadrangular. The foramen for the nasolacrimal duct has eroded the ventral margin of the prefrontal and the dorsal margin of the facial process of the maxilla, but the nasal is not involved. The maxilla is edentulous posteriorly for about 55% of its length. Its posterior tip flares laterally beyond the margin of the lower jaw in dorsal view. There are five large maxillary teeth on each side and four premaxillary teeth. The orbitosphenoid, while moderately well-developed, is articulated solidly to the parasphenoid, weakly to the frontal, and not at all to the parietal. The oculomotor foramen is absent on the right side.

There are no prominent crests on the dorsal surface of either otic capsule. The posterolateral tab of the parietal is well-developed but relatively short and triangular; it is reflected ventromedially and ends in a rounded point about halfway down the vertical extent of the orbitosphenoid. The squamosal bone is more elongate and less triangular than in other *Chiropterotriton*; its dorsal tip articulates with a small portion of the otic capsule opposite the lateral semicircular canal. The quadrate is small and inconspicuous. The stylus on the columella is short and stout. Paired bodies of the vomer are reasonably well developed but they barely articulate medially posterior to the internasal fontanel. Postorbital processes of are long, thin and slightly curved. There are six vomerine teeth on the right side and six on the left; one or two teeth are deployed at the base of each preorbital process. The parasphenoid expands posteriorly but truncates abruptly at its caudal border. Each lateral edge is sculpted by a shallow notch opposite the jaw articulation, and by an erosion of bone (and teeth) opposite the ventromedially directed parietal tab. It has an unusual shape along the lateral margin. Paired parasphenoid tooth patches are separated across the midline; each bears 50–52 fully developed teeth, but there are many additional developing teeth along the lateral margin. The mandible is solid. The articular bone is well developed and may be at least partly fused to the pre-articular on each side. The prearticular has a relatively high coronoid process. There are approximately 12 teeth on each dentary bone.

Digital formulae are 1-2-3-2 on each side. There is a tiny expanded knob at the tip of each terminal phalanx. Mesopodial cartilages are not mineralized.

**Distribution and ecology:** *Chiropterotriton orculus* is restricted to the central and eastern portion of the Trans-Mexican Volcanic Belt (La Marquesa, Desierto de los Leones, Ajusco, Lagunas de Zempoala, Iztaccihuatl, Popocatépetl, Rio Frio and La Malinche). It occurs in pine and fir forest and is terrestrial; it is typically found under the bark of logs or inside rotting logs. This widely distributed species ranges between 2,500 and 3,500 masl.

**Remarks:** This species was raised from synonymy with *C. chiropterus* by Darda (his species G, population 20). While it is relatively widespread, we are unsure of its northeastern limits. Population G is from near Chignahuapan, Puebla. We also include Darda's sp.

A (Desierto de Los Leones, DF) and B (Rio Frio, Mexico) in our current understanding of this taxon.

**Conservation status:** *Chiropterotriton orculus* is designated as Vulnerable by the most recent IUCN Red List of Threatened Species (*Parra-Olea & Wake, 2008*). The species remains relatively common near Lagunas de Zempoala.

## OTHER SPECIES OF *CHIROPTEROTRITON* FROM CENTRAL VERACRUZ

In addition to the recently described *C. aureus* and *C. nubilus*, *C. lavae* also occurs in the mountains of central Veracruz. While *Taylor's (1942)* original description of this species was relatively thorough, we provide a brief overview of this species for comparative purposes using additional specimens collected since the type series. We also examined the holotype and several paratypes of this species to provide additional information not contained in Taylor's description.

*Chiropterotriton lavae* (*Taylor, 1942*)

Chresonymy
*Bolitoglossa lavae*—Taylor, 1942. *Holotype:* EHT-HMS 28937, now FMNH 100118. *Type locality:* "2 miles west of La Joya-Veracruz", Mexico.
Pigmy Flat-footed Salamander, Salamandra de pie plano pigmea
Figures 4V–4X, 5H, 6C, 7C, 8C.

**Specimens examined:** Nineteen specimens, all from La Joya, Veracruz, Mexico. Ten males: MVZ 163912–13, 163915, 171873–74, 173394–95, 173398, 178685 and 192789; and nine females: MVZ 106537, 106548, 171876, 171881, 171885, 171901, 192788, 197788 and 200638.

**Diagnosis:** This is a medium-sized species of plethodontid salamander phylogenetically related to *Chiropterotriton totonacus, C. perotensis* and *C. ceronorum*; mean SVL 32.4 mm in ten adult males (range 31.1–33.8) and 31.6 mm in nine adult females (range 27.9–34.9). The head is moderately wide; HW averages 15% of SVL in males (range 14–17) and 15% in females (range 14–16). Jaw muscles are prominent in both males and females. Adult males and females have a bluntly rounded snout with moderately developed nasolabial protuberances. Eyes are large and prominent and extend laterally well beyond the jaw margin in ventral view. There are few maxillary teeth in males (mean MT 7.0, range 1–10) and moderate numbers in females (mean MT 20.8, range 13–36). There are few vomerine teeth in both males (mean VT 8.9, 7–10) and females (mean VT 11.4, range 8–15), which are arranged in a short row that does not reach or barely reaches the inner margin of the internal choana. The tail is moderately long and slightly exceeds SVL in most specimens; mean TL/SVL equals 1.19 in males (range 1.11–1.27) and 1.02 in females (range 0.85–1.15). Limbs are moderately to very long in both females and males; FLL + HLL averages 59% of SVL in males (range 53–65) and 54% in females (range 50–59). Adpressed limbs closely approach or overlap in males (mean LI −0.60, range −1.0 to 0.0) but are more separated in females (mean LI 0.6, range 0.0–2.0). The manus and pes

are moderate in size. Subterminal pads are well developed. Digital webbing is modest, reaching only to the base of the penultimate phalanx on the third toe. The first digit is included entirely in webbing. Digital tips are slightly expanded. The mental gland is oval (nearly round), somewhat prominent and moderately sized in males. The smallest male with a mental gland is 31.2 mm SVL.

**Comparisons:** *Chiropterotriton lavae* differs from *C. ceronorum* in its slightly smaller adult body size (mean SVL 32.4 mm in male and 31.6 mm in female *C. lavae* vs. 33.9 mm in male and 34.9 mm in female *C. ceronorum*), longer tail (mean TL/SVL 1.19 in male and 1.02 in female *C. lavae* vs. 1.0 in male and 0.97 in female *C. ceronorum*), longer limbs (mean LI −0.6 in male and 0.6 in female *C. lavae* vs. 0.0 in male and 1.5 in female *C. ceronorum*), fewer maxillary teeth (mean MT 7.0 in male and 20.8 in female *C. lavae* vs. 11.0 in male and 47.7 in female *C. ceronorum*) and fewer vomerine teeth (mean VT 8.9 in male and 11.4 in female *C. lavae* vs. 13.0 in male and 15.9 in female *C. ceronorum*).

 *Chiropterotriton lavae* differs from *C. perotensis* in its larger adult body size in males (mean SVL 32.4 mm in *C. lavae* vs. 29.7 mm in *C. perotensis*), longer limbs (mean LI −0.6 in male and 0.6 in female *C. lavae* vs. 2.5 in male and 3.3 in female *C. perotensis*), slightly wider head (mean HW 4.9 mm in male and 4.7 mm in female *C. lavae* vs. 4.2 mm in male and 4.4 mm in female *C. perotensis*), longer head (mean HL 7.5 mm in male and 7.0 mm in female *C. lavae* vs. 6.6 mm in male and 6.7 mm in female *C. perotensis*), wider feet (FW 3.7 mm in male and 3.3 mm in female *C. lavae* vs. 2.6 mm in both male and female *C. perotensis*) and slightly fewer maxillary teeth in females (mean MT 20.8 in *C. lavae* vs. 27.9 in *C. perotensis*).

 *Chiropterotriton lavae* differs from *C. totonacus* in its smaller adult body size (mean SVL 32.4 mm in male and 31.6 mm in female *C. lavae* vs. 35.7 mm in male and 35.5 mm in female *C. totonacus*), shorter tail in females (mean TL/SVL 1.02 in *C. lavae* vs. 1.20 in *C. totonacus*), shorter limbs in females (mean LI 0.6 in *C. lavae* vs. 0.0 in *C. totonacus*), shorter head (mean HL 7.5 mm in male and 7.0 mm in female *C. lavae* vs. 8.5 mm in male and 7.6 mm in female *C. totonacus*), slightly narrower head (mean HW 4.9 mm in male and 4.7 mm in female *C. lavae* vs. 5.2 mm in both male and female *C. totonacus*), narrower feet (mean FW 3.7 mm in male and 3.3 mm in female *C. lavae* vs. 4.2 mm in male and 4.0 mm in female *C. totonacus*), fewer maxillary teeth (mean MT 7.0 in male and 20.8 in female *C. lavae* vs. 32.9 in male and 52.6 in female *C. totonacus*) and fewer vomerine teeth (mean VT 8.9 in male and 11.4 in female *C. lavae* vs. 11.6 in male and 13.7 in female *C. totonacus*).

 *Chiropterotriton lavae* differs from *C. melipona* in its larger adult body size (mean SVL 32.4 mm in male and 31.6 mm in female *C. lavae* vs. 29.2 mm in male and 28.5 mm in female *C. melipona*), longer head (mean HL 7.5 mm in male and 7.0 mm in female *C. lavae* vs. 6.3 mm in male and 6.4 mm in female *C. melipona*), broader head (mean HW 4.9 mm in male and 4.7 mm in female *C. lavae* vs. 4.3 mm in male and 4.2 mm in female *C. melipona*), longer limbs (mean LI −0.6 in male and 0.6 in female *C. lavae* vs. 2.3 in male and 1.8 in female *C. melipona*), broader feet (mean FW 3.7 mm in male and 3.3 mm in female *C. lavae* vs. 2.4 mm in male and 2.6 mm in female *C. melipona*), fewer

maxillary teeth (mean MT 7.0 in male and 20.8 in female *C. lavae* vs. 9.5 in male and 31.0 in female *C. melipona*) and fewer vomerine teeth (mean VT 8.9 in male and 11.4 in female *C. lavae* vs. 11.0 in male and 13.0 in female *C. melipona*).

*Chiropterotriton lavae* differs from *C. casasi* in its smaller adult body size (mean SVL 32.4 mm in male and 31.6 mm in female *C. lavae* vs. 37.8 mm in male and 40.9 mm in one female *C. casasi*), longer tail in males (mean TL/SVL 1.19 in *C. lavae* vs. 1.04 in *C. casasi*), shorter head (mean HL 7.5 mm in male and 7.0 mm in female *C. lavae* vs. 8.3 mm in male and 8.6 mm in one female *C. casasi*), narrower head (mean HW 4.9 mm in male and 4.7 mm in female *C. lavae* vs. 5.8 mm in male and 5.9 mm in one female *C. casasi*), longer limbs (mean LI −0.6 in male and 0.6 in female *C. lavae* vs. 0.8 in male and 1.0 in one female *C. casasi*) and fewer maxillary teeth in females (mean MT 20.8 in *C. lavae* vs. 30 in *C. casasi*).

*Chiropterotriton lavae* differs from *C. chiropterus* in its smaller adult body size (mean SVL 32.4 mm in male and 31.6 mm in female *C. lavae* vs. 37.5 mm in male and 33.5 mm in female *C. chiropterus*), shorter tail (mean TL/SVL 1.19 in male and 1.02 in female *C. lavae* vs. 1.25 in male and 1.19 in female *C. chiropterus*), longer limbs (mean LI −0.6 in male and 0.6 in female *C. lavae* vs. 0.3 in male and 2.0 in female *C. chiropterus*), shorter head (mean HL 7.5 mm in male and 7.0 mm in female *C. lavae* vs. 8.1 mm in male and 7.3 mm in female *C. chiropterus*), narrower head (mean HW 4.9 mm in male and 4.7 mm in female *C. lavae* vs. 5.6 mm in male and 4.8 mm in female *C. chiropterus*), fewer maxillary teeth (mean MT 7.0 in male and 20.8 in female *C. lavae* vs. 12.6 in male and 48.0 in female *C. chiropterus*) and fewer vomerine teeth (mean VT 8.9 in male and 11.4 in female *C. lavae* vs. 10.6 in male and 12.5 in female *C. chiropterus*).

*Chiropterotriton lavae* differs from *C. orculus* in its smaller adult body size (mean SVL 32.4 mm in male and 31.6 mm in female *C. lavae* vs. 35.9 mm in male and 39.0 mm in female *C. orculus*), longer tail in males (mean TL/SVL 1.19 in *C. lavae* vs. 1.02 in *C. orculus*) and longer limbs (mean LI −0.6 in male and 0.6 in female *C. lavae* vs. 1.9 in male and 2.9 in female *C. orculus*).

*Chiropterotriton lavae* differs from *C. aureus* in its larger adult body size (mean SVL 32.4 mm in male and 31.6 mm in female *C. lavae* vs. 28.5 mm in one male, mean 26.8 mm in female *C. aureus*), larger head (mean HL 7.5 mm in male and 7.0 mm in female *C. lavae* vs. 6.4 mm in one male, mean 6.0 mm in female *C. aureus*), broader head (mean HW 4.9 mm in male and 4.7 mm in female *C. lavae* vs. 4.0 mm in one male, 3.6 mm in female *C. aureus*), longer limbs (mean LI −0.6 in male and 0.6 in female *C. lavae* vs. 2.0 in one male, mean 2.3 in female *C. aureus*), and broader feet (mean FW 3.7 mm in male and 3.3 mm in female *C. lavae* vs. 2.4 mm in one male, mean 1.8 mm in female *C. aureus*).

*Chiropterotriton lavae* differs from *C. nubilus* in its larger adult body size in males (mean SVL 32.4 mm in *lavae* vs. 29.4 mm in one male *C. nubilus*), shorter tail (mean TL/SVL 1.19 in male and 1.02 in female *C. lavae* vs. 1.37 in one male, mean 1.12 in female *C. nubilus*), broader head (mean HW 4.9 mm in male and 4.7 mm in female *C. lavae* vs. 4.0 mm in one male, mean 4.4 mm in female *C. nubilus*), relatively longer limbs (mean LI −0.6 in male and 0.6 in female *C. lavae* vs. 2.0 in one male, mean 1.5 in female

*C. nubilus*), and broader feet (mean FW 3.7 mm in male and 3.3 mm in female *C. lavae* vs. 2.6 mm in one male, mean 2.3 mm in female *C. nubilus*).

**Measurements of holotype:** Adult female, SVL 33.5 mm, TL 40.7 mm, AX 18.1 mm, SW 4.8 mm, HL 7.7 mm, HW 5.6 mm, HD 2.9 mm, interorbital distance 2.1 mm, eyelid length 1.3 mm, FLL 9.2 mm, HLL 9.7 mm, snout-to-forelimb length 10.2 mm, snout to anterior angle of vent 33 mm, length of fifth toe 0.9 mm, distance from eye to nostril 1.2, internarial distance 2.0, FW 4.0, length of longest (third) toe 1.6 mm. Numbers of teeth: premaxillary 6, maxillary 16-14 (right-left) and vomerine 6-6 (right-left). *Taylor (1942)* listed 28 maxillary and premaxillary teeth on each side but counted missing teeth, while we count only ankylosed teeth that are present. Adpressed limbs touch.

**Coloration in life:** Dorsal coloration highly variable. Background dorsal color dark brown; some individuals have a broad, continuous dorsal band of yellow, reddish-brown or orangish-brown to pale brown stretching from posterior portion of head to tip of tail, while in other individuals this dorsal band is either irregular, reduced to paler brown or golden-brown blotches or streaks, or absent. Head dark brown, often with golden-brown specks, especially between eyes and snout. Flanks, sides of tail, and dorsal side of limbs and feet dark brown, typically uniform along dorsal edge but often with paler brown or golden-brown flecks or tan streaks below; toe tips reddish. Venter dark gray to paler gray, with some white speckling in some individuals. Iris golden-brown.

**Coloration in preservative:** The dorsum, tail and head are relatively pale to dark brown, often with a paler, broad dorsal band that is bordered by darker brown coloration. The paler background color is often faintly mottled with darker brown. The venter is a uniform tan to pale brown; the underside of the tail and limbs are a slightly darker brown. The gular region is tan to pale brown, sometimes with a small amount of mottling.

**Osteology:** This account is based on examination of a μCT scan of the anterior skeleton of MVZ 163912, an adult male, 33.8 mm SVL (Figs. 6–8). The skull is well developed. The cranial roof is for the most part complete and solidly articulated. There is no frontoparietal fontanel, although there are slight gaps medially between the paired frontals and paired parietals. Ascending processes of the single premaxillary bone remain separate along their entire length; each broadens laterally as it approaches its dorsal articulation with the adjacent frontal. A very narrow palatal shelf is present on each side of the premaxilla but absent medially. There are no septomaxillary bones. The nasal bone is large and triangular, but also very thin and poorly ossified. The prefrontal bone is rectangular and robust; its ventral portion is overlapped extensively by the facial process of the maxilla. The foramen of the nasolacrimal duct has eroded the prefrontal along its anteroventral margin and the dorsal margin of the facial process of the maxilla; the nasal abuts the foramen but is eroded minimally, if at all. The maxillary bone is saber-like in lateral view, not cleaver-like as in many other *Chiropterotriton*. Its posterior, edentulous portion comprises about 60% of the length of the bone. There are four maxillary teeth on

each side and two premaxillary teeth. The teeth are thin and poorly developed. The orbitosphenoid is very thin and delicate. It is solidly articulated to the parasphenoid but weakly articulated to the frontal and parietal.

A prominent bony ridge overlies the anterior semicircular canal dorsally. It is derived from the posterolateral portion of the parietal bone and the anteromedial portion of the otic capsule. An additional, crest-like spur emerges at right angles from this crest and is directed posterolaterally. A second ridge similarly overlies the posterior semicircular canal. The squamosal bone is robust and roughly triangular. A well-developed, spine-like tab on the ventrolateral margin of each parietal is sharply reflected ventromedially and extends nearly the full vertical extent of the orbitosphenoid. The quadrate is small and inconspicuous and incompletely ossified. There is a stubby, stout stylus on the columella, with a limited free portion. Paired vomers are weakly ossified; they approach one another across the midline posterior to the internasal fontanel but do not articulate. Preorbital processes are needlelike—thin and elongate. There are four vomerine teeth on each side; one tooth is deployed at the base of the preorbital process, but only on the left side. The parasphenoid bone is relatively wide anteriorly. Each lateral edge is sculpted by a deep notch opposite the jaw articulation. Paired parasphenoid tooth patches are widely separated across the midline; each contains approximately 50 teeth. The mandible is stout. The articular is well ossified. The prearticular is very thin in its central portion but has a moderately high coronoid process. There are eight teeth on each dentary bone. The posterior teeth are sharply recurved and needlelike.

Digital formulae are 1-2-3-2 on each side. Phalanges appear to be slightly thinner than in other *Chiropterotriton*. There is a slightly expanded knob at the tip of each terminal phalanx of digits 2–4. Mesopodial cartilages are not mineralized.

**Distribution and ecology:** *Chiropterotriton lavae* is known only from forested areas between the towns of Toxtlacoaya and La Joya, along the road from Perote to Xalapa, Veracruz, Mexico. It occurs in bromeliads in the cloud forest and has been found in somewhat disturbed habitat in and around La Joya. This narrowly distributed species is known only between 2,000 and 2,200 masl.

**Remarks:** As part of the redescription of this species, we examined the holotype and part of the series of paratypes at the Field Museum of Natural History. The portion of the type series examined corresponds closely in morphology to the specimens that we examined.

There has long been a suspicion that two species of *Chiropterotriton* occur in the vicinity of La Joya. For example, *Smith & Taylor (1948)* report *Chiropterotriton chiropterus* (almost certainly not that species) from Toxtlacoaya, and they also report *C. lavae* from that site. This small village is at the western edge of La Joya. *Darda (1994)* also reports two species from La Joya, *C. lavae* and his new species E (which we tentatively assign to *C. totonacus* in this paper). We have only found one species in the La Joya region.

**Conservation Status:** *Chiropterotriton lavae* is designated as Critically Endangered by the most recent IUCN Red List of Threatened Species (*IUCN SSC Amphibian Specialist Group, 2016*). Much of the habitat where it occurs is highly disturbed or has been converted to

pasture, but this species remains relatively common even in disturbed forest where there are bromeliads.

## DISCUSSION

Since its initial designation by *Taylor (1944)*, *Chiropterotriton* has proven to be a problematic taxon. As originally conceived, the genus contained small montane species of tropical salamanders with broad hands and feet and the outermost digit relatively well developed. Species ranged from terrestrial to arboreal and occurred at relatively high elevations (9,000 to 11,000 feet, or roughly 2,750 to 3,350 m). With a largely Mexican distribution, the initial ten species nevertheless extended geographically to Honduras. Later, species from as far south as Costa Rica were added to the genus. Today, the taxon is restricted to Mexico, north and west of the Isthmus of Tehuantepec but mainly in eastern Mexico (as far west as southeastern Coahuila, central San Luis Potosi and Queretaro, and western Distrito Federal and Morelos). The known elevational range is both lower (to about 690 m below Xicotepec de Juarez, Veracruz) and higher (to at least 4,015 m on Cofre de Perote) than was known when Taylor worked. Species further to the south once considered congeneric are now assigned to the distantly related genera *Cryptotriton, Dendrotriton* and *Nototriton*. While most species are small, *C. magnipes* reaches about 60 mm SVL. *Darda's (1994)* southern group is the most taxonomically difficult group in the genus, and even after our description of four new members of it herein (*C. casasi* stands out as morphologically unique among the taxa named, and we cannot determine at this time to which group it belongs) there is still taxonomic work remaining. Moreover, opportunities exist for additional research investigations, especially cytological. Chromosomal heteromorphism is reported for a few species of *Chiropterotriton*, including potential sex chromosomes and supernumerary chromosomes (*Sessions & Kezer, 1991*). Similarly, genome size has been studied in only four species. Known values are at the smaller end of the size range for the tropical salamander radiation; average C-value per species ranges from 24.7 to 28.5 pg DNA (*Sessions & Kezer, 1991*).

Despite the passage of nearly 50 years between the description of *C. magnipes* and *C. miquihuanus*, it has long been known based on both morphological and molecular evidence that a great deal of additional diversity exists within the genus (*Rabb, 1958*; *Darda, 1994*; *Parra-Olea, 2003*). The recent descriptions of three species identified as distinct in previous morphological or molecular analyses (*C. chico, C. cieloensis* and *C. infernalis*) went some way towards formalizing the known but undescribed diversity of *Chiropterotriton*, while the descriptions of three more species not included in previous analyses (*C. aureus, C. miquihuanus*, and *C. nubilus*) showed that there is still previously undocumented diversity left to discover. Of the five species we describe here, four were previously identified as distinct, while the fifth (*C. casasi*) has not been included in any previous analysis. These five species add to the already high diversity of the eastern portion of the Trans-Mexican Volcanic Belt (TMVB).

Using allozyme data, *Darda (1994)* provided the first in-depth taxonomic study of the genus *Chiropterotriton* that included molecular data. Darda's *C. chiropterus* complex (the southern clade) was formed by *C. chiropterus* (represented in his study by sp. E from

La Joya, Veracruz) and *C. orculus* (represented by sp. G from Chignahuapan, Puebla), plus nine undescribed taxa: *C.* sp A, *C.* sp B, and *C.* sp F, from Puebla; *C.* sp C, *C.* sp D, *C.* sp H, and *C.* sp I from Veracruz; and *C.* sp. J and *C.* sp. K from Oaxaca. Once sequences of mitochondrial genes became available, *Parra-Olea (2003)* defined the type localities for *C. chiropterus* and *C. orculus*. *Parra-Olea (2003)* assigned the name *C. chiropterus* to populations from Huatusco, Veracruz, leaving Darda's sp. E as an undescribed species. She also assigned the name *C. orculus* to populations from the central region of the Trans-Mexican Volcanic Belt (TMVB) including Darda's sp. A and sp. B, indicating that sp. G from Chignahuapan might represent an undescribed taxon. No further taxonomic work was performed on this complex until now. Based on our analyses, including molecular and morphological data, here we describe four of these taxa: *C. totonacus* (sp. E from La Joya Veracruz), *C. melipona* (sp. F from Xicotepec, Veracruz), *C. perotensis* (sp. H and sp. D from Las Vigas, Veracruz) and *C. ceronorum* (sp. I from Santa Cruz Texmalaquilla, Puebla). We assign *C.* sp. J from La Esperanza, Oaxaca, as part of *C. chiropterus*.

Phylogenetic evidence, based first on allozyme data (*Darda, 1994*) and continuing with mtDNA data from the work of *Parra-Olea (2003)* to the present study has been indispensable to working out species limits within the genus. One of the most problematic taxonomic issues with the genus *Chiroperotriton* was the status of *C. chiropterus*. The fact that the original description contained little morphological information, combined with an imprecise type locality and lost holotype, made assignment of populations to this species difficult. At different times, this name has been applied to populations ranging from Tamaulipas south through San Luis Potosí, Querétaro, Hidalgo, and Veracruz. Furthermore, the species is relatively generalized in morphology, resembling a number of other small to medium-sized members of the genus. Our designation of a neotype formalized the assignment of the name *C. chiropterus* for populations from the region of Huatusco, Veracruz, following *Parra-Olea (2003)*. Inclusion of samples from Huatusco in both phylogenetic and morphological analyses allowed us to distinguish several of the new species from the eastern edge of the TMVB. Furthermore, while *Parra-Olea (2003)* restricted *C. chiropterus* to the vicinity of Huatusco, we now understand that it ranges south to northern Oaxaca. Rather than being microendemic, it now has one of the largest ranges of any *Chiropterotriton*. Similarly, while *Darda (1994)* restricted *C. orculus* to a single population based on allozyme data, our results support the status of *C. orculus* as a more widely ranging species throughout the eastern TMVB.

Of the species identified as undescribed in previous analyses (*Darda, 1994*; *Parra-Olea, 2003*), only *Chiropterotriton* sp. C, sp. G, and sp. K have not been either described or assigned to an existing species. We believe that *C.* sp. C (from Puerto del Aire, Veracruz) likely represents a distinct species but currently lack sufficient material to describe it. Major declines in salamander abundance have occurred at this site (*Rovito et al., 2009*) and no *Chiropterotriton* have been found in recent years. *Chiropterotriton* sp. G is similar to *C. orculus* in external morphology and was assigned to that species by *Darda (1994)*, but *Parra-Olea (2003)* reversed this decision and applied the name *C. orculus* to populations around Mexico City. Additional morphological analyses are necessary to determine if

*C.* sp. G represents a distinct species or can be assigned to the wider-ranging *C. orculus*. The case of *C.* sp. K, however, is more difficult. This species, collected only once in 1980, has not been seen over the course of many visits to Cerro San Felipe, Oaxaca. While it is possible that the locality is in error, many other species at this site have undergone catastrophic declines (*Parra-Olea, García-París & Wake, 1999*; *Rovito et al., 2009*). *Chiropterotriton* sp. K may be present on Cerro San Felipe at greatly diminished abundance, or it may simply exist on a part of the mountain that has not been checked on subsequent visits; the locality of the known specimens is not specific enough to determine exactly where they were collected. Concerted field efforts covering different parts of Cerro San Felipe are needed to confirm that *C.* sp. K does indeed exist at the locality. While the descriptions of *C. perotensis* (sp. D and sp. H), *C. totonacus* (sp. E), *C. ceronorum* (sp. I) and *C. melipona* (sp. F), together with the assignment of *C.* sp. J to *C. chiropterus*, nearly deal with all the identified but undescribed diversity within the genus, we continue to discover populations that likely represent additional, undescribed species of *Chiropterotriton* from eastern portions of the TMVB. A final issue is the status of a population from the Sierra Madre del Sur of Oaxaca, known from a single long-preserved specimen in the American Museum of Natural History (*Darda, 1994*).

These five new species increase the content of *Chiropterotriton* from 18 to 23. This represents a considerable increase in the somewhat slow but steady rise in species descriptions trajectory that began in the 1980s when molecular data became readily available. With the use of protein electrophoresis data, 19 new species of salamanders were described from Mexico (*Hanken & Wake, 1994*, *1998*, *2001*; *Hanken, Wake & Freeman, 1999*) and with the use of mitochondrial markers 31 new species have been described from Mexico since 2001 (*Parra-Olea, Papenfuss & Wake, 2001*; *Parra-Olea, García-París & Wake, 2002*; *Parra-Olea, Canseco-Márquez & García-París, 2004*; *Parra-Olea et al., 2004*, *2005a*, *2005b*, *2010*, *2016*; *Brodie, Mendelson & Campbell, 2002*; *Canseco-Marquez & Parra-Olea, 2003*; *Canseco-Márquez & Gutiérrez-Mayén, 2005*; *Rovito et al., 2012*, *2015b*; *Rovito & Parra-Olea, 2015*; *García-Castillo et al., 2017*, *2018*; *Sandoval-Comte et al., 2017*). Thus, almost 40% of Mexican bolitoglossines have been described using molecular characters in combination with morphological and ecological traits. The number of described species in *Chiropterotriton* alone has nearly doubled over the course of 5 years, and we expect that additional fieldwork in the TMVB and Sierra Madre Oriental will reveal additional species.

## CONCLUSIONS

The genus *Chiropterotriton*, an endemic group of Mexican salamanders, has been a taxonomic challenge to researchers for many years. Previously published molecular data indicated that a number of undescribed species were present, but lack of a thorough morphological analysis had stalled the advances in the description of the diversity of this group. This article is a big step towards this goal. Herein we describe five new species of *Chiropterotriton* and redescribed two more, based on molecular and morphological data, increasing considerably the known diversity of the genus. However, more work is still needed for the description of several more taxa when additional data are available.

# APPENDIX

**Appendix 1** Specimens examined for morphological comparisons.

*Chiropterotriton casasi* **sp.nov.**: Mexico, Veracruz: MVZ 92874–78, 13 mi SW Tlapacoyan

*Chiropterotriton ceronorum* **sp.nov.**: Mexico, Puebla: USNM 224202, 224207–08, 224211–12, 224218–20, 224230, 224236, 224240–41, 224247, 224250, 224252–53, 224257, 224259, 224275–76, Santa Cruz Texmalaquilla (4.7 mi by road NE of Atzitzintla), ca. 1 km NE of, on south slope of Pico de Orizaba

*Chiropterotriton chiropterus*: Mexico, Veracruz: MVZ 85588–92, 85594, 85597–99, 85605, 85613, 85632, 1.4 mi SW (by road) SW edge of Huatusco de Chicuellar

*Chiropterotriton lavae*: Mexico, Veracruz: MVZ 106537, 106548, W edge of La Joya, along Hwy. 140; MVZ 163912–13, 163915, 171873–74, 171876, 171881, 171885, 171901, 173394–95, 173398, 192788–89, 197788, La Joya; 178685, La Joya, Mexico Hwy. 140; MVZ 200638 forest W of La Joya

*Chiropterotriton melipona* **sp.nov.**: Mexico, Puebla: MVZ 178706–08, 3.9 km S Xicotepec de Juárez on Hwy. 130; MVZ 185972, 2.2 km on road to Patla from junction with Hwy. 120 SW out of Xicotepec de Juárez; MVZ 200724–26, 3.3 km S of Hotel M Ranchito on Mexico Hwy. 130, 2.1 km E on road to La Unión, Xicotepec de Juárez

*Chiropterotriton orculus*: Mexico, Estado de México: MVZ 76161, 138686, 138688, 138694, 138696–97, 138700, 138776–79, 138781, 138783–84, 138793, 138796–97, 138804, 200629–30, ridge between Volcanoes Popocatepetl and Iztaccihuatl, along Mexico Hwy. 196, 16.2 km E (by road) Hwy. 115

*Chiropterotriton perotensis* **sp.nov.**: Mexico, Veracruz: MVZ 114356, 114359, road from Las Vigas de Ramírez to Microwave Station on N Flank Cofre de Perote, 11.6 km S (by road) Las Vigas; MVZ 173428–29, 173438–39, Las Vigas de Ramírez, Microondas road; MVZ 178661, 178663–65, 8–15.5 km S (via Microondas Rd.) Las Vigas de Ramírez; MVZ 186711, road to Microwave Station, 15 km S (by road) Las Vigas de Ramírez; MVZ 200681–83, 200691, 200693–95, 200698, 200702 14.4 km S (by Rock Rd.) Las Vigas de Ramírez at Microwave Station

*Chiropterotriton totonacus* **sp.nov.**: MVZ 136981–82, 136986, pine forest along Mexico Hwy. 140, 4 km W Las Vigas de Ramírez; MVZ 138703–04, 138716, 138765, Mexico Hwy. 140, 4.5 km W (by road) Las Vigas de Ramírez; MVZ 163943, 163945, 163947–49, 163989–90, 163993, 171903, 171905, 171907, 171909–10, 6 km W Las Vigas de Ramírez

## ACKNOWLEDGEMENTS

We thank Ángel Soto-Pozos, Maria Delia Basanta and Omar Becerra-Soria for valuable help during field work of 2016–2018; Laura Marquez Valdelamar and Andrea Jimenez Marin for their help in laboratory work; Carol Spencer for help provided with specimen photographs, and Eduardo Pineda for tissues and photographs of *C. chiropterus* specimens from Colección de Referencia de Anfibios y Reptiles del Instituto de Ecología, A.C (CARIE).

### Funding

Research was supported by grants from Programa de Apoyo a Proyectos de Investigación e Innnovación Tecnológica (PAPIIT-UNAM) IN203617 to Gabriela Parra Olea; CONACyT Ciencia Básica Grant #221614 (CB-2013-01), Langebio-Cinvestav; a UC Mexus-CONACyT postdoctoral fellowship to Sean M. Rovito; National Science Foundation: EF-0334846 and DBI-1702263 to James Hanken; and National Science Foundation: DEB-0613802 to David B Wake. CONACyT provided a scholarship grant (CVU/Becario 413761/262662) to Mirna Garcia Castillo. Putnam Expeditionary Fund of the Museum of Comparative Zoology and the David Rockefeller Center for Latin American Studies, Harvard University, supported James Hanken. This publication was supported by a grant from the Wetmore-Colles Fund of the Museum of Comparative Zoology to James Hanken.

The funders had no role in study design, data collection and analysis, decision to publish, or preparation of the manuscript.

## Grant Disclosures

The following grant information was disclosed by the authors:
Programa de Apoyo a Proyectos de Investigación e Innnovación Tecnológica
(PAPIIT-UNAM): IN203617.
CONACyT Ciencia Básica: #221614 (CB-2013-01).
UC Mexus-CONACyT.
National Science Foundation: EF-0334846 and DBI-1702263.
National Science Foundation: DEB-0613802.
CONACyT: CVU/Becario 413761/262662.
Putnam Expeditionary Fund of the Museum of Comparative Zoology and the David
Rockefeller Center for Latin American Studies, Harvard University.
Wetmore-Colles Fund.

## Competing Interests

Gabriela Parra Olea is an Academic Editor for PeerJ.

## Author Contributions

- Gabriela Parra Olea conceived and designed the experiments, performed the experiments, analyzed the data, prepared figures and/or tables, authored or reviewed drafts of the paper, and approved the final draft.
- Mirna G. Garcia-Castillo conceived and designed the experiments, performed the experiments, analyzed the data, prepared figures and/or tables, authored or reviewed drafts of the paper, and approved the final draft.
- Sean M. Rovito conceived and designed the experiments, performed the experiments, analyzed the data, prepared figures and/or tables, authored or reviewed drafts of the paper, and approved the final draft.
- Jessica A. Maisano conceived and designed the experiments, performed the experiments, analyzed the data, prepared figures and/or tables, authored or reviewed drafts of the paper, and approved the final draft.
- James Hanken conceived and designed the experiments, performed the experiments, analyzed the data, prepared figures and/or tables, authored or reviewed drafts of the paper, and approved the final draft.
- David B. Wake conceived and designed the experiments, performed the experiments, analyzed the data, prepared figures and/or tables, authored or reviewed drafts of the paper, and approved the final draft.

## Animal Ethics

The following information was supplied relating to ethical approvals (i.e., approving body and any reference numbers):

Animal use was approved by the University of California, Berkeley, IACUC protocol #R093-0205 to DBW.

## Field Study Permissions

The following information was supplied relating to field study approvals (i.e., approving body and any reference numbers):

Collection permits were provided by the Secretaría del Medio Ambiente y Recursos Naturales (SEMARNAT): SGPA/DGVS/00947/16, SGPA/DGVS/03038/17 and FAUT-0303, issued to Gabriela Parra-Olea.

## DNA Deposition

The following information was supplied regarding the deposition of DNA sequences:

The sequences of *Chiropterotriton* are available at GenBank: MN914712–MN914747 and MN920423–MN920429. A fasta file is also available as a Supplemental File.

## Data Availability

Raw data is available as a Supplemental File.

The CT data are available at MorphoSource:

*Chiropterotriton chiropterus*: https://www.morphosource.org/Detail/SpecimenDetail/Show/specimen_id/27352.

*C. orculus*: https://www.morphosource.org/Detail/SpecimenDetail/Show/specimen_id/27372.

*C. lavae*: https://www.morphosource.org/Detail/SpecimenDetail/Show/specimen_id/27387.

*C. totonacus*: https://www.morphosource.org/Detail/SpecimenDetail/Show/specimen_id/27391.

*C. melipona*: https://www.morphosource.org/Detail/SpecimenDetail/Show/specimen_id/27394.

*C. perotensis*: https://www.morphosource.org/Detail/SpecimenDetail/Show/specimen_id/27399.

*C. casasi*: https://www.morphosource.org/Detail/SpecimenDetail/Show/specimen_id/27354.

*C. ceronorum*: https://www.morphosource.org/Detail/SpecimenDetail/Show/specimen_id/27400.

## New Species Registration

The following information was supplied regarding the registration of a newly described species:

Publication LSID:

urn:lsid:zoobank.org:pub:9B4B9DFF-E12B-430D-A541-BA0EBB9B90E6

*Chiropterotriton casasi* sp. nov. LSID:

urn:lsid:zoobank.org:act:248D1A23-66B7-4672-8AA3-44C4058D4F4F

*Chiropterotriton ceronorum* sp. nov. LSID:

urn:lsid:zoobank.org:act:5BE9F6D2-CACD-41F7-8E1C-09C5E0FE140A

*Chiropterotriton melipona* sp. nov. LSID:

urn:lsid:zoobank.org:act:ED19C47F-B804-4FFB-A004-A258625E3E25
*Chiropterotriton perotensis* sp. nov. LSID:
urn:lsid:zoobank.org:act:54AB015C-5CCD-46C7-B260-8BACA8D02C68
*Chiropterotriton totonacus* sp. nov. LSID:
urn:lsid:zoobank.org:act:831CB0EF-5D91-4DEC-A4B1-76714D9C21AD.

## Supplemental Information

Supplemental information for this article can be found online at http://dx.doi.org/10.7717/peerj.8800#supplemental-information.

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
