# Peer review of "Descriptions of five new species of the salamander genus Chiropterotriton (Caudata: Plethodontidae) from eastern Mexico and the status of three currently recognized taxa"

_PeerJ, doi:10.7717/peerj.8800_

## Round 0.1 · original submission · Minor Revisions

This paper is an important contribution to the knowledge on Mexican salamanders. The manuscript is extremely well-written and all data presented are solid. Quality of descriptions and illustrations is outstanding. Two reviewers provided their quite positive opinions. I also carefully read the manuscript and noted several minor points, mostly dealing with manuscript consistency, formatting of the manuscript according to PeerJ rules, and some stylistic issues. All my comments are presented in the attached reviewed PDF file.

Please note that according to PeerJ regulations, the GenBank Accession Numbers and the links to raw data depository must be provided at the submission of the manuscript.

I am quite sure that the authors will easily address all the raised questions and resubmit the manuscript very soon.

·

Basic reporting

No comment

Experimental design

No comments

Validity of the findings

No comment

Additional comments

Brief review of: Descriptions of five new species of the salamander genus Chiropterotriton (Caudata" Plethodontidae) from eastern Mexico and the status of three currently recognized taxa

I have carefully read this paper and think it is an extremely well-written and reasoned contribution. It makes a huge step in understanding the diversity of one of the most difficult plethodontid genera that has been a challenge to researchers or many years. These small salamanders are speciose and often morphologically conservative. The authors are well suited to tackle this problem and are among the foremost systematists of Neotropical salamanders.

Information available previously showed that a number of undescribed species were present in the mountains of eastern Mexico. Therefore, this is a welcome follow-up to previous work that demonstrated that the genus contained a number of cryptic species. This paper summarizes previously known information and adds considerable additional molecular and morphological (i.e., CT scans) data.

Most of these new species of salamanders, along with previously known species, are microendemics. This paper elucidates the composition of the genus and provides distributional data that will be useful to a wide audience, including biogeographers, evolutionary biologists, and conservationists.

A helpful overview of the history of research is given. Solutions are provided to several long-standing taxonomic problems of the group, such as the identities of Chiropterotriton orculus and C. chiropterus, by designating neotypes for each. A more complete description of C. lavae is also given

Descriptions are thorough and follow previous formats making comparison among species easier. Especially welcome are the detailed diagnoses (i.e., "Comparisons") of these new species with congeners. Redescription of previously described species and their comparison with other species is necessary in view of the current complexity of the genus. This kind of systematic review is labor-intensive and necessary understanding the diversity of this neglected and previously poorly understood group.

It might be noted that, despite the admirable effort expended by the authors in this research, they indicate that the last word on Chiropterotriton diversity has not been said, and they identify several other yet-to-be-described populations that undoubtedly will be part of subsequent papers when additional data become available.

The figures are well executed. I strongly endorse publication.

Reviewer 2 ·

Basic reporting

Please see attached

Experimental design

Please see attached

Validity of the findings

Please see attached

Additional comments

Please see attached

Annotated reviews are not available for download in order to protect the identity of reviewers who chose to remain anonymous.

---

## Round 0.2 · accepted · Accept

Thank you for taking the time to revise and resubmit your manuscript. I have now read through your paper as well as your letter in response to the reviews. I think that you have successfully addressed all of the concerns raised very well, and would like to accept your manuscript for publication in PeerJ. Congratulations!

Thank you for all the hard work you have put in to this. Your paper makes a strong contribution to the literature and I look forward to seeing it published.

---

## Author Rebuttal · Round 0.2

Dr Nikolay Poyarkov
Academic Editor
Peer J

Dear Dr. Poyarkov

We received revisions from Dr. J. Campbell, an anonymous reviewer and yourself. First of all, we appreciate your time and effort invested in this manuscript. We revised the manuscript considering all of the useful suggestions and modified the manuscript as follows:

Reviewer # 1 Dr. Jonathan Campbell strongly endorsed publication of the manuscript because he found it to be very useful for the understanding of salamander systematics and taxonomy.

Reviewer #2. Anonymous. Requested that we include or at least mention something about karyological data in the ms as stated in his revision: *The manuscript presents a convincing analysis of this group and a thorough diagnosis of the newly identified species.* ***My one and only concern is that this is one of the few groups of plethodontid salamanders that have interesting features in their karyotypes, and yet chromosome diversity is not mentioned in this manuscript at all!***

We followed his advice and included the following paragraph at the beginning of the discussion: Since its initial designation by Taylor (1944), *Chiropterotriton* has proven to be a problematic taxon. As originally conceived, the genus contained small montane species of tropical salamanders with broad hands and feet and the outermost digit relatively well developed. Species ranged from terrestrial to arboreal and occurred at relatively high elevations (9,000 to 11,000 feet, or roughly 2750 to 3350 m). With a largely Mexican distribution, the initial ten species nevertheless extended geographically to Honduras. Later, species from as far south as Costa Rica were added to the genus. Today, the taxon is restricted to Mexico, north and west of the Isthmus of Tehuantepec but mainly in eastern Mexico (as far west as southeastern Coahuila, central San Luis Potosi and Queretaro, and western Distrito Federal and Morelos). The known elevational range is both lower (to about 690 m below Xicotepec de Juarez, Veracruz) and higher (to at least 4015 m on Cofre de Perote) than was known when Taylor worked. Species further to the south once considered congeneric are now assigned to the distantly related genera *Cryptotriton, Dendrotriton* and *Nototriton*. While most species are small, *C. magnipes* reaches about 60 mm SVL. Darda's (1994) southern group is the most taxonomically difficult group in the genus, and even after our description of four new members of it herein (*C. casasi* stands out as morphologically unique among the taxa named, and we can't determine at this time to which group it belongs) there is still taxonomic work remaining. **Moreover, opportunities exist for additional research investigations, especially cytological. Chromosomal heteromorphism is reported for a few species of *Chiropterotriton*, including potential sex chromosomes and supernumerary chromosomes (Sessions and Kezer 1991). Similarly, genome size has been studied in only four species. Known values are at the smaller end of the size range for the tropical salamander radiation; average C-value per species ranges from 24.7 to 28.5 pg DNA (Sessions and Kezer 1991).**

Editor revision. Finally, from your revision and annotated manuscript we added "sp. nov." where the new species are mentioned in all the figures and tables and dealt with all the minor things marked in the ms.

I hope you find this revision acceptable.

Sincerly,

Dra. Gabriela Parra Olea